# Introspective inference counteracts perceptual distortion

Andra Mihali [1,2] ✉, Marianne Broeker [1,2,3,4], Florian D. M. Ragalmuto[1,2,5,6] & Guillermo Horga[1,2] ✉

Introspective agents can recognize the extent to which their internal perceptual experiences deviate from the actual states of the external world. This ability, also known as insight, is critically required for reality testing and is impaired in psychosis, yet little is known about its cognitive underpinnings. We develop a Bayesian modeling framework and a psychophysics paradigm to quantitatively characterize this type of insight while people experience a motion after-effect illusion. People can incorporate knowledge about the illusion into their decisions when judging the actual direction of a motion stimulus, compensating for the illusion (and often overcompensating). Furthermore, confidence, reaction-time, and pupil-dilation data all show signatures consistent with inferential adjustments in the Bayesian insight model. Our results suggest that people can question the veracity of what they see by making insightful inferences that incorporate introspective knowledge about internal distortions.

Successfully navigating the modern world requires questioning the validity of sensory information—testing whether it conforms to reality rather than simply taking it at face value. In other words, one cannot always believe what one sees. This is apparent in considering "deep-fakes", where objective sensory information is deliberately manufactured in such lifelike ways that viewers sometimes believe it is real. This is also the case for hallucinations, where internal distortions drive seemingly veridical percepts that are often falsely believed to be real. The corollary is that to avoid harboring false beliefs, and acting erroneously on their basis, the presence of potentially deceiving sensory information or experiences needs to be met with a healthy degree of skepticism in one's own senses. The ability to perform such "reality testing"—e.g., testing whether subjective, internal sensory experiences reflect actual states of the external world—is characteristically impaired in psychosis[1,2]. Although perceptual distortions, including hallucinations, are common in a variety of neuropsychiatric conditions as well as in some non-clinical groups[3], psychotic disorders such as schizophrenia are classically distinguished by an inability to question the reality of distorted percepts, a concept referred to as impaired

insight[4]. Put more simply, among the people who experience perceptual distortions, some can recognize them as such and others cannot. The latter case of impaired insight leads to false beliefs and can drive seemingly erratic behaviors, including poor treatment adherence[5,6]. Alterations in reality testing also have important legal ramifications given their impact on determinations of criminal responsibility. Despite the broad societal implications of reality testing, the cognitive mechanisms underlying this ability are insufficiently understood[4].

Progress in this area has been hindered by the lack of a formal explanatory framework and experimental paradigms suitable to study reality testing. Previous work related to reality testing has broadly used two main approaches: (1) source (self versus other) memory tasks in the context of semantic association and sentence completion[7,8]; and, more recently, (2) imagery tasks in the context of perceptual decision-making[9–13]. The first approach relies on episodic memory and does not capture in-the-moment reality testing; the second approach captures in-the-moment processes and leverages signal-detection theory but relies on individuals' imagery ability, thus limiting experimental control and translatability. Furthermore, while imagery is relevant to some

[1]New York State Psychiatric Institute, New York, NY, USA. [2]Columbia University, Department of Psychiatry, New York, NY, USA. [3]Columbia University, Teachers College, New York, NY, USA. [4]University of Oxford, Department of Experimental Psychology, Oxford, UK. [5]Vrije Universiteit, Faculty of Behavioral and Movement Science, Amsterdam, the Netherlands. [6]Berliner FortbildungsAkademie, Berlin, DE, Germany. ✉e-mail: andra.mihali@nyspi.columbia.edu; horgag@nyspi.columbia.edu

forms of reality testing, reality testing is a broader construct[11,14] and its impairments need not involve altered imagery. Here, we focus on a form of reality testing that we refer to as 'perceptual insight': the in-the-moment process of incorporating introspective knowledge about distortions in internal percepts to effectively infer the actual state of the external world. We propose a formal model of perceptual insight building on Bayesian theory and present a perceptual-insight paradigm that minimizes memory confounds and enhances experimental control over previous designs.

In this work, we reasoned that probing perceptual insight would first require experimentally inducing a sufficiently strong perceptual distortion—a discrepancy between an objective stimulus feature and its subjective perceptual experience or estimate—and then probing beliefs about the true state of varying objective stimuli. To do this, we used the motion after-effect (MAE), a well-characterized class of illusions that includes strong illusory percepts of (seemingly veridical) complex motion induced by prolonged viewing of an adaptor motion stimulus. We deemed the complex MAE particularly suitable for studying perceptual insight because explicit knowledge about the illusion seemed to allow accurate inferences on the actual state of objective stimuli while their subjective experience was distorted. Augmenting classic MAE methods with psychophysics tools, computational modeling, and pupillometry, we thus set out to validate an explanatory framework for human perceptual insight.

Using these approaches, here we show that healthy participants can compensate for the MAE illusion when they report their beliefs about actual direction of motion, relative to when they report perceived motion. We show that this MAE compensation is best explained by a Bayesian model that captures insight through adjustments at an intermediate inferential stage. This conclusion is further supported by pupillometry data and drift-diffusion modeling. In sum, we show that people can make insightful inferences that incorporate knowledge about their internal perceptual distortions to counteract these distortions.

## Results

### Formal model of perceptual insight

We focus on the concept of perceptual insight, defined as the incorporation of introspective knowledge about internal distortions to effectively infer the actual states of the external world. Although related to other notions of introspection[15–18], this construct specifically implies an ability to make judgments incorporating knowledge that the internal representation of external states may not accurately reflect these states—i.e., that internal experiences may not match reality. In particular, it implies incorporating knowledge that internal representations may be systematically influenced by factors other than external states (e.g., knowing that a voice one hears or a motion pattern one sees is "in one's head" and does not correspond to a speaker or a moving object in the outside world).

To formulate this construct, we first consider a standard model of perceptual decision-making[19,20]. Broadly, this model captures how an agent observing a stimulus $s$ makes a decision about the stimulus, for instance the category $C$ it belongs to. This process involves an early sensory-encoding stage, in which the external stimulus $s$ is encoded into a neural measurement or internal sensory representation $x$ (Fig. 1A, left). Based on $x$, which is corrupted by noise, at an intermediate stage the agent infers aspects of the external stimulus such as its category $C$. Finally, at a late stage, the agent makes a choice $\hat{C}$ with a certain level of confidence $q$. A Bayesian observer formally solves this problem by first inferring the posterior probability of $C$ at the intermediate stage, combining its likelihood and prior probability as $p(C|x) \propto p(x|C)p(C)$, and computing a perceptual-decision variable $d$ consisting of the log-posterior ratio of the two possible options ($C = 1$ or $C = -1$). The observer then chooses the category with the highest posterior probability at the late stage by comparing $d$ to a threshold $k_{\text{choice}}$, with a value of 0 in the

unbiased case. The observer further generates a confidence response $q$ by comparing the posterior probability of the chosen response $p(C = \hat{C}|x)$ to a threshold $k_{\text{confidence}}$[21–23] (see Methods, "Computational models", "Standard perceptual decision model").

Now we consider a scenario directly relevant to perceptual insight where the internal representation $x$ is additionally influenced by a distortion factor $A$ and is no longer solely a function of the external stimulus $s$. Factor $A$ (e.g., a psychotomimetic substance or a sensory adaptor) can distort the internal representation $x$, for instance off-setting $x$ from $s$ (Fig. 1A, center). Similar to the standard case, an insightful Bayesian observer can optimally infer the posterior probability $p(C|x)$ but will require incorporating knowledge about the distortion factor $A$, specifically about the influence of $A$ on $x$ and the possible values of $A$, as $p(C|x) \propto \int p(x|C,A)p(C)p(A)dA$ (see Methods, "Computational models"). This optimal agent has perceptual insight because it knows its internal representation $x$ can be influenced by factors other than external states (Fig. 1B, right). By optimally incorporating this knowledge at an intermediate inferential stage and appropriately shifting the perceptual-decision variable $d$, the insightful agent can compensate for factor $A$ and make accurate decisions about category $C$ using an unbiased response rule—i.e., choosing the option with the highest posterior probability by comparing $d$ to a $k_{\text{choice}}$ of 0 (Fig. 1C, right).

We must distinguish this insightful optimal agent (Fig. 1C, right) from one that behaves similarly under factor $A$ but may lack perceptual insight (Fig. 1C, center). Without explicitly incorporating knowledge about factor $A$ at the intermediate inferential stage, this alternative agent may still compensate for factor $A$ at a late stage by adjusting its response rule, i.e., via a response bias (as if it knew its decisions were inaccurate but not the underlying cause). This agent would infer $p(C|x)$ as $p(x|C)p(C)$—that is, not incorporating knowledge about factor $A$ into the perceptual-decision variable $d$—and would only change its responses by using a $k_{\text{choice}}$ different than 0 (Fig. 1C, center). This is reminiscent of patients with impaired insight who nonetheless learn to report skepticism about their distorted percepts (such as hallucinations) to derive some benefit. While an insightful agent could possibly incorporate knowledge about factor $A$ suboptimally by adjusting its response rule, observing a response bias does not guarantee an insightful strategy (as it is also consistent with the absence of insight). In contrast, observing the abovementioned shift in $d$ incorporating knowledge about factor $A$ (as in Fig. 1B, right) does imply the use of an insightful strategy.

Here, we set out to test whether human observers experiencing internal distortions in perception demonstrate perceptual insight. Because its hallmark is a distinct shift in the perceptual-decision variable $d$ under perceptual insight (Fig. 1D, center versus right), through which the insightful agent incorporates introspective knowledge about the internal distortion into its posterior beliefs, we placed particular emphasis on behavioral and physiology measures shown to reflect this variable $d$. More generally, our study is distinct from previous extensive work showing that observers integrate external stimuli[24–28] and external feedback[20,29,30] into perceptual decisions in line with Bayesian theory, in that it is, to our knowledge, the first to examine integration of evidence from external stimuli with introspective knowledge about an internal distortion in the absence of external feedback. Furthermore, previous work has shown that perceptual-decision changes at different stages can manifest as distinct patterns of choice, reaction times (RT), confidence reports[31,32], and pupil dilation[33], which together can help differentiate shifts in the perceptual-decision variable $d$ such as those we specifically hypothesize under perceptual insight.

### Experimental paradigm of perceptual insight

To produce a strong perceptual distortion, we induced an MAE illusion of complex motion using a counterclockwise-rotating Archimedean

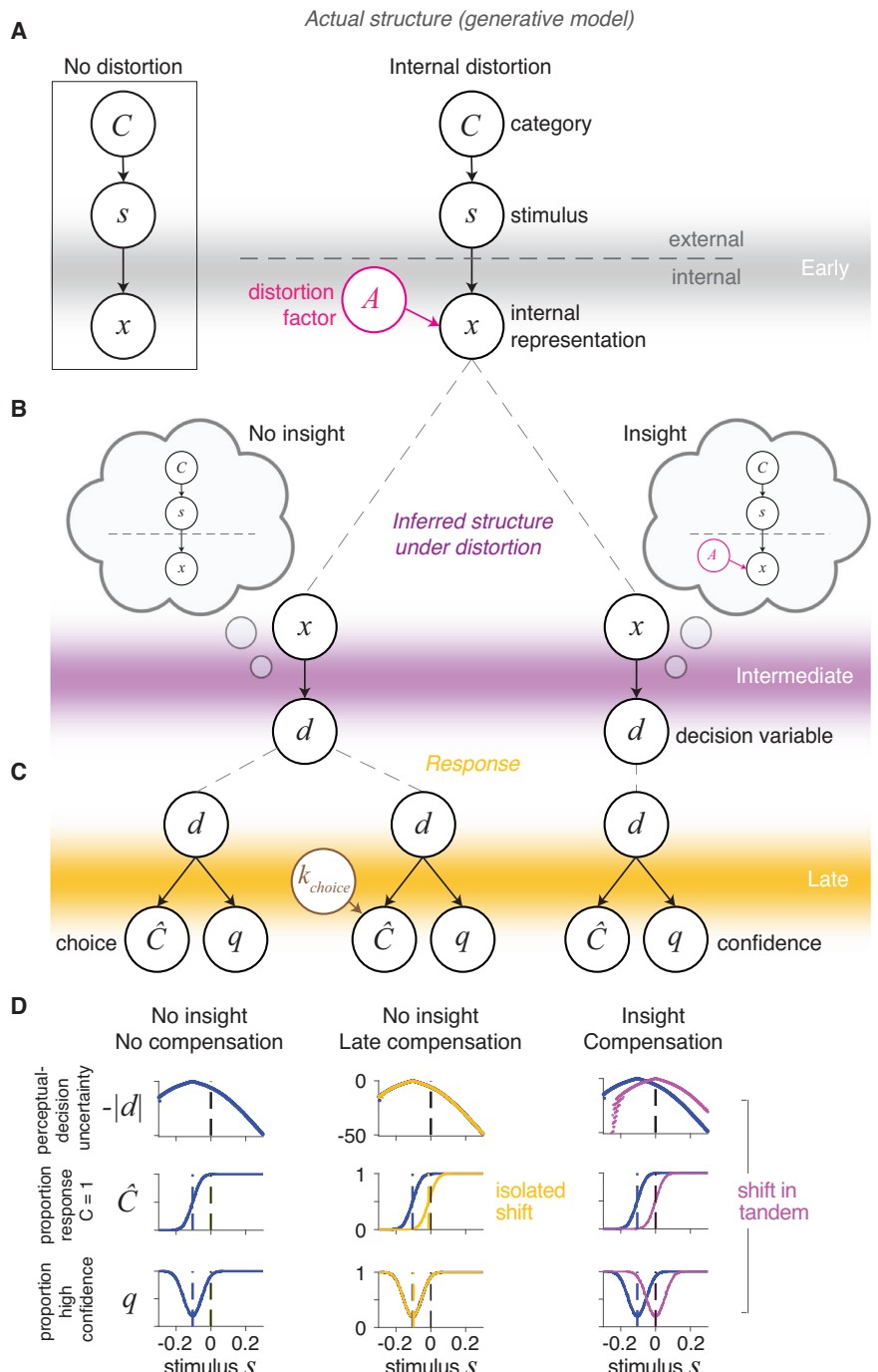

**Fig. 1 | Perceptual-decision framework for perceptual insight. A** Generative model of the early encoding stage with internal distortion due to factor $A$ (center) and with no distortion (left box). **B** At the intermediate stage, an observer's inference under distortion can either take into account the actual generative model and thus demonstrate insight (right cloud) or wrongly assume the generative model without distortion and thus lack insight (left cloud). **C** Response–choice ($\hat{C}$) and confidence ($q$)–generation from the decision variable $d$ at the late stage. **D** Choice and confidence patterns with insight (right) and without insight (center and left) are distinct. Compensation for the distortion through perceptual insight (right) leads to shifts in tandem of the choice and confidence curves. Late compensation with no insight (center) leads to isolated shifts in the choice curves.

spiral, based on the ability of complex motion (e.g., rotational) to induce stronger adaptation than simple motion (e.g., translational)[34,35]. We measured MAE strength via the nulling method[36] as the shift in the bias of the psychometric curve, i.e., the bias induced by a first moving spiral (the adaptor) on the observer's judgment of a second spiral moving at different speeds and directions. The overall paradigm (Fig. 2) had a two-by-two design, with two types of conditions, 'Adapt' (rotating first spiral acting as adaptor) and 'No-Adapt' (static first spiral

acting as control), and two types of responses required upon observing the second spiral (test stimulus), 'See' and 'Believe'. 'See' responses required participants to report the perceived direction of motion of the test stimulus (i.e., whether they saw the second spiral moving counterclockwise [left] or clockwise [right]); 'Believe' responses instead required them to report the inferred true direction of motion of the test stimulus (i.e., whether participants thought that the second spiral was actually moving counterclockwise or clockwise).

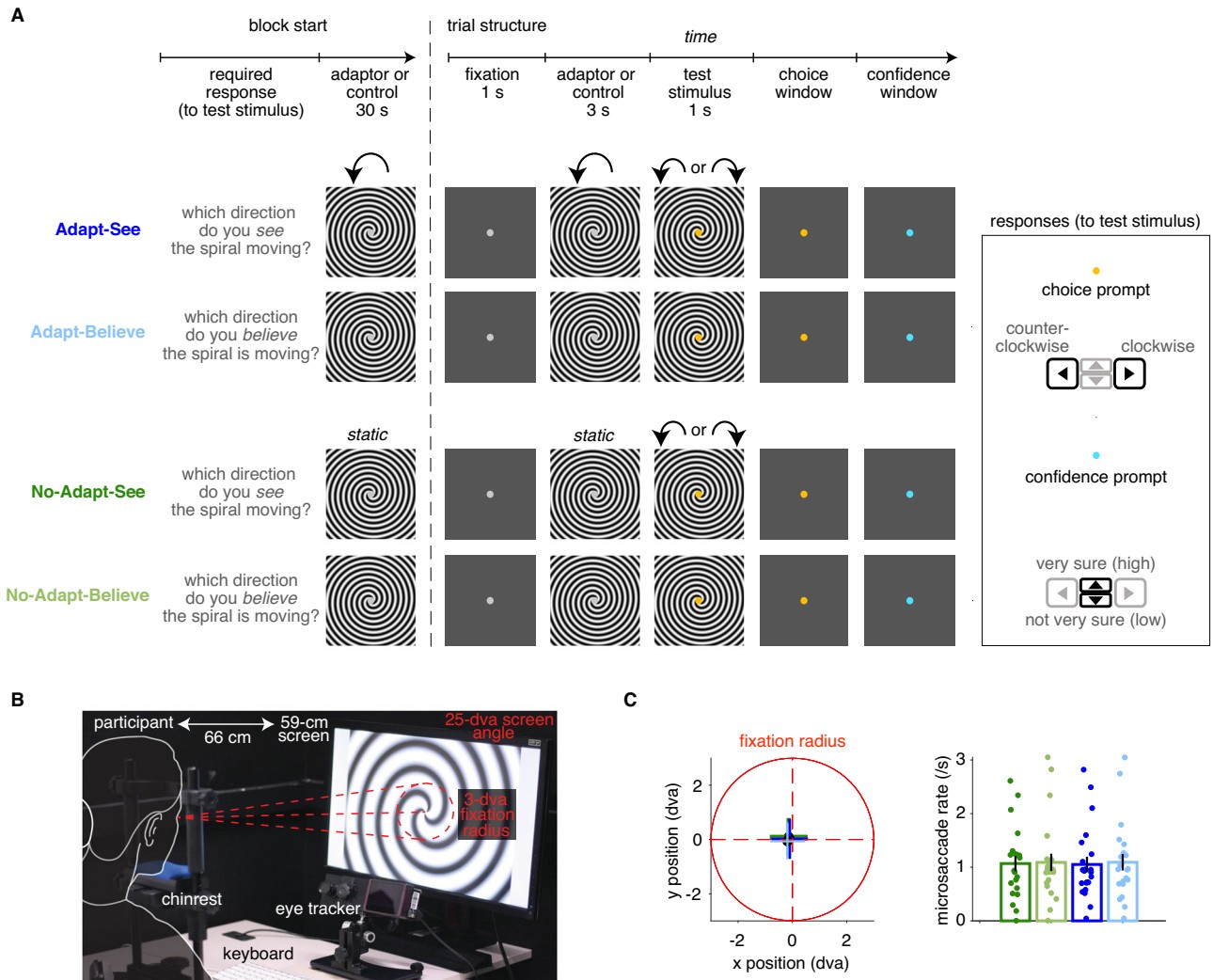

**Fig. 2 | Perceptual-insight task and fixation stability. A** The four task conditions differing on the presence of a rotating spiral adaptor or static control spiral (Adapt, top blue, versus No-Adapt, bottom green) and required responses (See, dark colors, versus Believe, light colors) are depicted. Details (e.g., prompt colors and response-window duration) correspond to Experiment 2 (see Methods "Experiment 1" for minor differences in Experiment 1). Blocks start with a reminder of required responses followed by a 30-s spiral rotating counterclockwise at constant speed or a static control spiral. Trials start with a fixation screen followed by the first spiral (rotating adaptor at constant speed counterclockwise or static control) and then a second spiral (test stimulus) of variable speed. A binary left/right choice is then prompted about motion direction (for clockwise or counterclockwise motion in the test stimulus, respectively, as seen or believed depending on the required response), followed by an up/down confidence response, provided to the question

"how sure are you of your response?". **B** Visualization of the experimental setup and gaze fixation control (see Methods "Trial structure with gaze fixation control"). **C** Left: Fixation positions. There were no statistically significant differences across the average fixation positions (averaged across trials per condition for each one of the $N = 22$ participants) in the four task conditions (planned post-hoc two-sided $t$-tests, all $p > 0.216$). Data represent mean across $N = 22$ participants and SEM across participants. Right: Microsaccade rates. Bars and error bars representing mean and SEM across $N = 22$ participants from Experiment 2. Microsaccade rates could reflect differences in fixation stability or affect MAE strength[145]. There were no significant differences in the microsaccade rates across conditions (planned post-hoc two-sided $t$-tests, all $p > 0.355$). (Microsaccades were measured following[146] and using parameters minimum velocity threshold multiplier $\lambda = 6$, minimum amplitude threshold of 1 dva and minimum duration of 6 ms).

Adapt blocks started with an extended exposure to an adaptor spiral (as in ref. 37) followed by trial-by-trial 3-second exposures to the adaptor spiral before each stimulus, and adaptor spirals always rotated counterclockwise at a constant speed. After each See or Believe response, participants also reported their confidence or "how sure they were about their responses" (high or low confidence). Participants received no feedback on their responses during the task. Critically, participants had explicit knowledge of the MAE and its illusory nature, acquired through detailed instructions and practice and individually demonstrated in MAE strength estimates (see Methods "Experiment 1" and "Experiment 2", "Instructions, demonstration, training" and Fig. S1). We measured MAE compensation as a relative correction of the Adapt-See bias (i.e., the MAE illusion) in the Adapt-Believe

condition and thus a candidate measure for perceptual insight. This corrective shift may in principle reflect participants' knowledge of the direction and strength of the illusion since both conditions had matched stimuli and only differed in their required responses.

**Participants compensate for distorted perception**

In a first experiment, participants experiencing the MAE illusion showed evidence of compensation (Fig. 3, top). Observers' responses tracked stimulus strength derived from a staircase procedure (see Methods "Experiment 1", "Test stimulus generation"; Fig. 3A, top) and were well described by psychometric curves (Gaussian cumulative density functions) with condition-specific bias $\mu$ and noise $\sigma$, and a shared lapse parameter $\lambda$ (Fig. 3B, see Methods "Psychometric

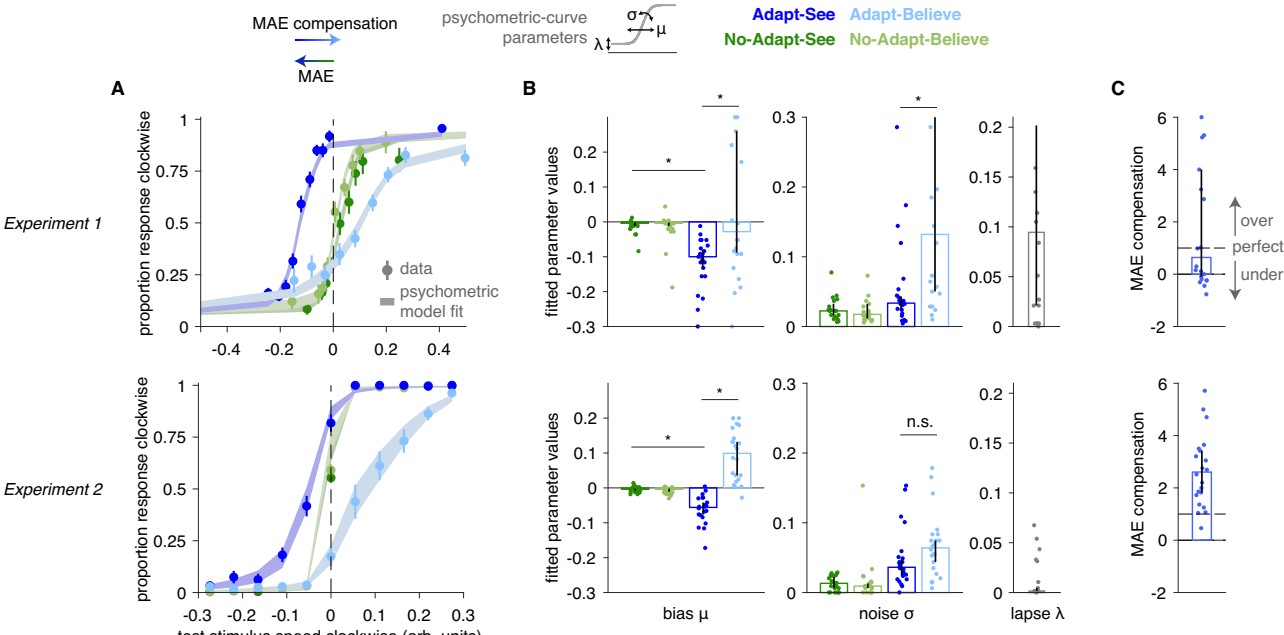

**Fig. 3 | Changes in psychometric curves demonstrate MAE compensation.** Data shown from Experiment 1 (top) and Experiment 2 (bottom). **A** Psychometric curves showing mean data (±SEM) across $N = 22$ participants and averaged psychometric-curve fits. **B** Fitted psychometric-curve parameter values (bars and error bars representing median ± bootstrapped 95% confidence intervals [CI]), showing a negative $\mu$ bias for Adapt-See (MAE) and $\mu$ close to 0 or positive for Adapt-Believe (indicating MAE compensation). * indicates $p < 0.01$ for planned two-sided Wilcoxon signed-rank tests. Note that psychometric curves are means ± SEM and that, to account for outliers, the fitted parameter plots use medians (and 95% CI), more consistent with the corresponding non-parametric tests presented in detail in text. We reiterate that we find differences between the bias $\mu$ parameter between No-Adapt-See and Adapt-See in Experiment 1 ($p < 0.001$) and Experiment 2 ($p < 0.001$), as well as between Adapt-See and Adapt-Believe in Experiment 1 ($p = 0.008$) and Experiment 2 ($p < 0.001$). **C** MAE compensation index normalized by each participant's respective $|\mu_{\text{Adapt–See}}|$ (bar plots and error bars are median ± 95% CI across $N = 22$ participants).

curves"). The MAE appeared as a leftward shift in the psychometric curves in Adapt-See relative to the control No-Adapt-See condition, indicating a bias towards perceiving clockwise motion (difference in $\mu$ bias parameter: $z = 4.108$, $p < 0.001$, bootstrapped 95% confidence interval [CI] [4.108, 4.108], two-sided Wilcoxon signed-rank test, effect size $r = 0.619$). Critically, relative to Adapt-See, the psychometric curves for Adapt-Believe showed a rightward corrective shift, indicating a compensation for the MAE ($z = 2.642$, $p = 0.008$, 95% CI [0.823, 3.602], effect size $r = 0.398$; Fig. 3C, bottom; note that throughout, this MAE compensation index is normalized by $|\mu_{\text{Adapt–See}}|$).

Despite evidence for MAE compensation in Experiment 1, we noted substantial interindividual variability and could not rule out the possibility that participants solved the Adapt-Believe condition by actively eluding the adaptor to minimize its influence (e.g., subtly looking away despite the instructions). To control for this, Experiment 2 used eye tracking to enforce fixation and prevent blinks for a 8 s time window encompassing the fixation, adaptor and stimulus periods and 3 s post-stimulus (see Methods "Trial structure with gaze fixation control"). We also sampled stimuli from a uniform distribution (informed by Experiment 1) to improve interpretability and facilitate modeling[38]. All results from Experiment 1 were replicated and more evident in Experiment 2 (Fig. 3, bottom), which generally produced higher-quality data: participants experienced the MAE and consistently compensated for it in Adapt-Believe (Fig. 3C, bottom; $z = 4.107$, $p < 0.001$, two-sided Wilcoxon signed-rank test, 95% CI [4.107, 4.107], effect size $r = 0.619$. Furthermore, unlike in Experiment 1, there was no statistically significant difference between the noise parameters $\sigma$ from Adapt-See and Adapt-Believe in Experiment 2 ($z = -1.412$, $p = 0.158$, two-sided Wilcoxon signed-rank test, bootstrapped 95 % CI: [-2.974, 0.829], effect size $r = 0.213$). Additionally, unlike in Experiment 1, Experiment 2 participants tended to exhibit a systematic over-compensation as psychometric curves in Adapt-Believe were often

shifted rightward beyond the control condition (Fig. 3A and C, Wilcoxon signed-rank test for MAE compensation above 1 was $z = 3.945$, $p < 0.001$, two-sided Wilcoxon signed-rank test, 95% CI [2.651, 4.111], effect size $r = 0.595$ for Experiment 2, versus $z = 1.185$, $p = 0.236$, 95 % CI [−0.862, 2.746], effect size $r = 0.179$ for Experiment 1). In sum, our participants experienced illusory motion and were able to either report the distorted percept (in Adapt-See) or discount it and compensate (or overcompensate) for it (in Adapt-Believe). Critically, they responded differently in Adapt-See and Adapt-Believe despite experiencing identical (adaptor and test) sensory stimuli across these two conditions.

MAE compensation could in principle reflect a more general compensatory strategy not due to insight gained from training and applied during the task. However, we empirically confirmed via MAE strength estimation that participants knew about the illusion and expected to have MAE roughly consistent in magnitude with the observed MAE effect during Adapt-See, albeit not perfectly calibrated (Supplementary Fig. S1). Furthermore, because participants received no feedback during the task, it is unlikely that they could compensate for the MAE via trial-and-error learning.

## MAE compensation is consistent with an intermediate inferential process
We then tested whether MAE compensation was consistent with perceptual insight. Optimal perceptual insight would imply an adjustment at the intermediate stage of inference that results in the computation of the perceptual-decision variable $d$ (the log-posterior ratio) under Adapt-Believe (Fig. 1C, right). In contrast, late compensation—which may or may not denote insight and which constitutes a suboptimal strategy—would only imply a change at the late-response stage, such as an offset in $k_{\text{choice}}$ (Fig. 1C, center), similar to that induced by response priming or asymmetric reward payouts[31,32,39,40]. The observed shifts in

psychometric curves (Fig. 3) are theoretically consistent with either of these two scenarios. However, among these two, only adjustments at the intermediate stage should cause commensurate shifts in perceptual-decision uncertainty, -|d|, the negative distance of $d$ from the point of maximal perceptual uncertainty ($d = 0$), that should be reflected in confidence and RT curves shifting in tandem with psychometric curves[31,32]. Under late compensation, $d$ should instead be identical in Adapt-See and Adapt-Believe, and so should be any measures reflecting perceptual-decision uncertainty (i.e., confidence and RT); late compensation should thus lead to an isolated shift in the psychometric curves between these conditions (Fig. 1C, center)[31,32,39].

Consistent with previous work, MAE-related shifts in the psychometric curves were accompanied by commensurate changes in confidence and RT curves (Fig. 4). Furthermore, biases under Adapt-See in these three different measures correlated across individuals (all $0.43 < \rho < 0.87$, all $p < 0.047$; Supplementary Fig. S2, Left). Critically,

psychometric curves defining MAE compensation also shifted in tandem with confidence and RT curves (Fig. 4). Under Adapt-Believe, individuals with larger biases in psychometric curves also exhibited larger biases in confidence and RT curves (all $0.63 < \rho < 0.85$, all $p < 0.002$; Supplementary Fig. S2, Right). This shows that MAE compensation involves a shift in perceptual-decision uncertainty, consistent with an adjustment at the intermediate inferential stage and inconsistent with a change restricted to a late response stage.

An alternative explanation for this pattern of compensation is a change at the early stage of sensory encoding, but this seems less tenable. First, participants reported continuing to experience the MAE illusion after receiving instructions and during the Adapt-Believe condition. Second, adaptation induces changes in early sensory neurons, such as reductions in firing rates associated with hyperpolarization[41–45], that are unlikely to be easily reversed at task-relevant timescales.

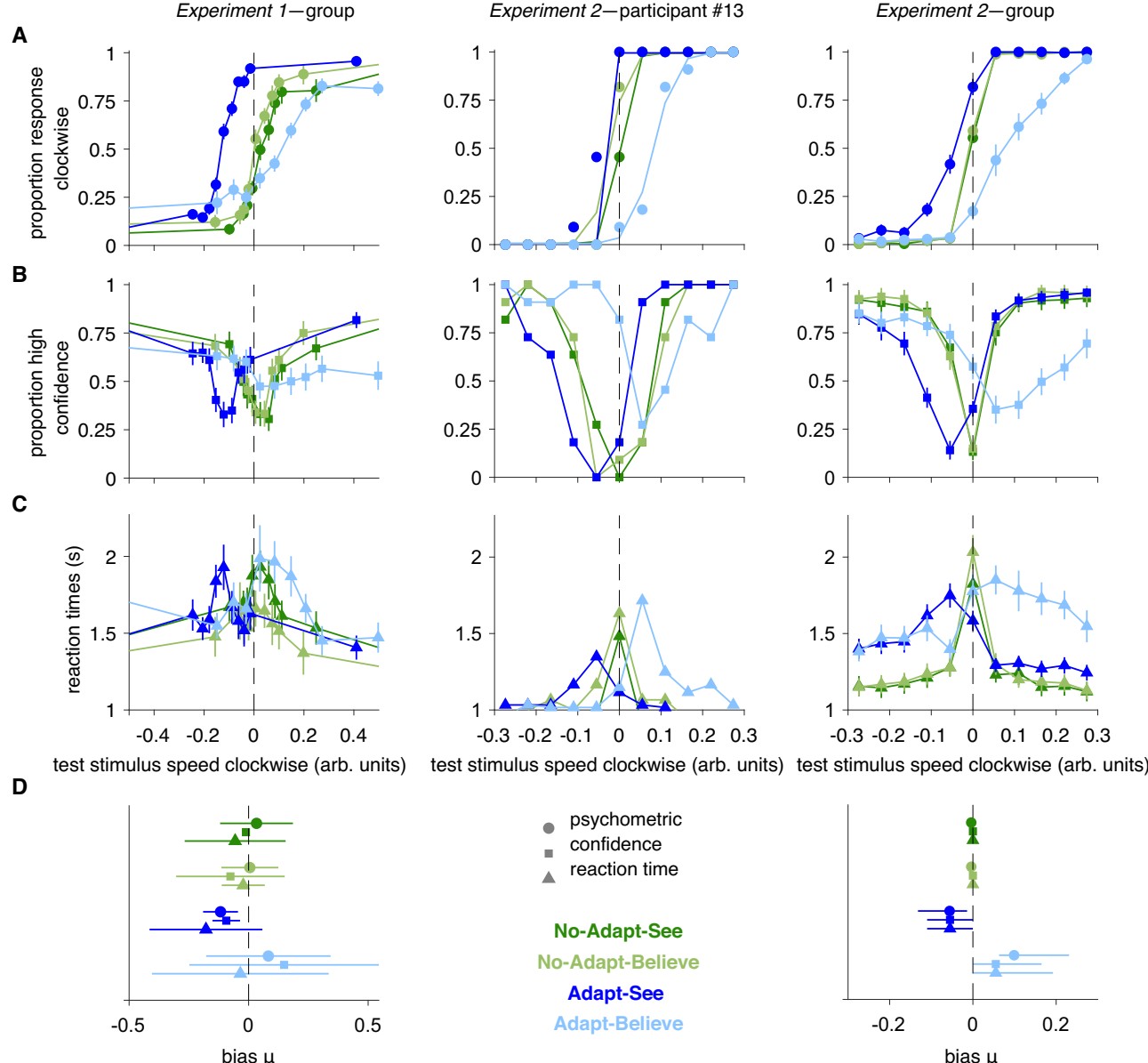

**Fig. 4 | Psychometric curves for MAE and MAE compensation shift in tandem with confidence and RT curves. A** Psychometric curves. **B** Confidence curves. **C** RT curves. Shown are group (mean ± SEM) data across the $N = 22$ participants from Experiment 1 (left), sample participant from Experiment 2 (center), and group (mean ± SEM) data across the $N = 22$ participants from Experiment 2 (right). **D** Concordance of $\mu$ bias estimates (median ± 95% CI, across the $N = 22$ participants) for psychometric, confidence, and RT curves for each condition. See Fig. S2 for interindividual correlations.

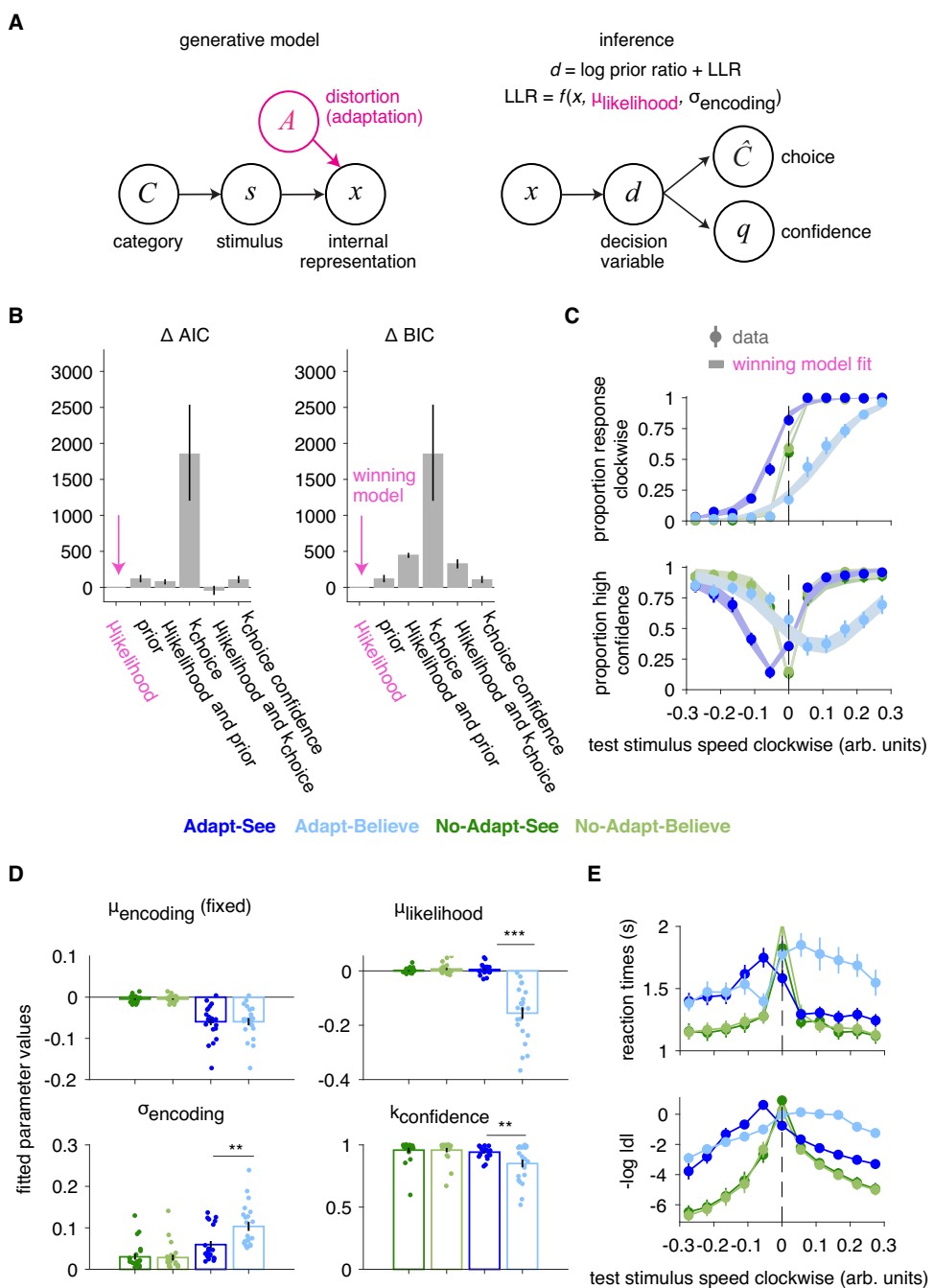

**Fig. 5 | Modeling of Experiment 2 data supports the perceptual-inference model.** **A** Simplified schematic of generative model and inference in the perceptual-insight model (recapitulating Fig. 1). Inference in the perceptual-insight model takes into account knowledge of the distortion through the term $\mu_{likelihood}$, which represents the observer's estimate of the distortion due to the factor A from the generative model. **B** Model comparison selects the perceptual-insight or $\mu_{likelihood}$ model as winning model. Bar plots represent summed ΔAIC and ΔBIC across N = 22 participants, and the error bars are 95% bootstrapped confidence intervals over 1000000 samples with replacement. **C** Fits from winning model (shaded areas) capture well choice and confidence curves (mean ± SEM across the N = 22

participants, as in Fig. 4); corresponding fitted parameter values (mean ± SEM in (**D**)). ***$p < 0.001$ and **$p < 0.01$, with results being based on planned post-hoc two-sided $t$-tests. Of interest, we find significant differences in Adapt-See vs. Adapt-Believe in $\mu_{likelihood}$ ($t(21) = 7.938, p < 0.001$, effect size Cohen's $d = 1.692$, 95% CI [1.249, 2.136]), $\sigma_{encoding}$ ($t(21) = -3.765, p = 0.001$, effect size Cohen's $d = -0.803$, 95% CI [-1.246, -0.359]), and $k_{confidence}$ ($t(21) = 3.032, p = 0.006$, effect size Cohen's $d = 0.646$, 95% CI [0.203, 1.089]). **E** RT curves mirror the perceptual-decision uncertainty, $-|d|$, from the winning model (shown in log space). Data are mean ± SEM across the N = 22 participants, for each bin.

## Bayesian modeling supports perceptual insight

To more directly test the model of perceptual insight (Fig. 1, right) versus other compensatory strategies, we fitted an adapted version of this model to data from Experiment 2 (Fig. 5). We specified the perceptual-insight and alternative models building from an extended Bayesian model of perceptual inference shown to capture variability in

confidence reports[22,23] (see Methods "Computational models", "Standard perceptual decision model"), the architecture of which (Fig. 5A) roughly maps onto the three stages of processing discussed above. An early stage in all models consisted of encoding the stimulus s into a noisy internal representation x, which was offset under the Adapt conditions. We assumed x was equally offset under Adapt-See and

Adapt-Believe reflecting an unavoidable effect of the adaptor at this early stage. At the intermediate stage, the posterior probability was computed and converted into a perceptual-decision variable $d$ consisting of the log-posterior ratio, which was then used at the late stage to produce a binary choice $\hat{C}$ by comparing $d$ to the threshold $k_{choice}$. Confidence responses $q$ reflected the posterior probability of the chosen response $p(C = \hat{C}|x)$[21,22], binarized via a comparison with the threshold $k_{confidence}$. Critically, unique to the perceptual-insight model was the incorporation of knowledge about factor $A$, here the adaptor, in its intermediate inferential stage $P(x|C, A)$ via a shift in the likelihood term $p(x|s, A)$ (see Methods "Computational models", "Perceptual-insight model"), which produced a shift in $d$ sufficient to compensate for the distortion without need for biasing its response rule ($k_{choice} = 0$); in contrast, the late-compensation model inferred $C$ without knowledge about $A$, as $P(C|x)$, requiring a biased response rule ($k_{choice} \neq 0$) to compensate for the MAE. Because the former scenario produces shifts of psychometric curves and confidence curves in tandem and the latter produces isolated shifts in psychometric curves (Fig. 1D), we deemed this basic model architecture flexible enough to capture the range of relevant behavioral patterns.

Formal comparison of models fitted jointly to choice and confidence reports (Table 1, see Methods "Computational models", "Model fitting") selected the perceptual-insight model over the late-compensation model, a hybrid model, alternative models that allowed changes in other inferential variables, including the category prior, and a late-compensation process simultaneously biasing choice and confidence reports (Fig. 5B). All models allowed for shifts in the internal representation $x$ during the Adapt conditions, relative to the control No-Adapt conditions, via two fixed parameters (one $\mu_{encoding}$ value for Adapt and one for No-Adapt), and additionally had condition-specific free parameters for sensory or encoding noise ($\sigma_{encoding}$) and confidence thresholds $k_{confidence}$ (Table 1). Similarly to the perceptual-insight ($\mu_{likelihood}$) model, the category prior model can compensate for the MAE via an intermediate inferential process manifesting as shifts in tandem, but achieves this compensation through a distinct suboptimal mechanism (using an incorrect category prior and an incorrect generative model); while the perceptual-insight model compensates by flexibly modifying the perceptual-variable $d$ in a way that scales with encoding noise (Equation (30), Supplementary Fig. S4 and Supplementary Fig. S5), the prior model simply and incorrectly assumes a change in the category prior that produces fixed shifts in $d$ regardless of encoding noise.

Importantly, the winning perceptual-insight ($\mu_{likelihood}$) model yielded meaningful parameters and satisfactorily captured the pattern of shifts in tandem associated with the MAE and MAE compensation (Fig. 5C, compare visually with the response-bias model fits in Supplementary Fig. S6), accomplishing the latter through condition-specific free $\mu_{likelihood}$ parameters that shifted the likelihood of $x$ under Adapt-Believe relative to the other conditions (Fig. 5D). Fitted $\mu_{likelihood}$ differed between Adapt-See and Adapt-Believe ($z = 4.107, p < 0.001$, 95% CI [4.107, 4.107], two-sided Wilcoxon signed-rank test, effect size $r = 0.619$), as did the fitted $\sigma_{encoding}$ ($z = -3.457, p < 0.001$, 95% CI [−4.053, −1.219], $r = -0.521$) and $k_{confidence}$ ($z = 2.613, p = 0.009$, 95% CI [0.731 3.622], $r = 0.394$). These parameter differences were meaningful as parameter recovery was successful (Supplementary Fig. S7). While our simplified insight model (see Methods "Computational models", "Perceptual-insight model") assumed no additional uncertainty related to the Adaptor ($\sigma_A \approx 0$), it captured increased encoding noise in Adapt-Believe relative to Adapt-See via condition-specific $\sigma_{encoding}$, likely reflecting a contribution of $\sigma_A$ (Supplementary Fig. S8). Critically, the change in $\mu_{likelihood}$ correlated with psychometric-curve shifts between Adapt-See and Adapt-Believe ($\rho = -0.95, p < 0.001$, 95% CI [−0.989, -0.866], Spearman correlation). Changes in the other parameters ($\sigma_{encoding}$ and $k_{confidence}$) did not show significant correlations with the psychometric-curve shifts between Adapt-See and Adapt-Believe (both

$p > 0.634$). Altogether, these results suggest that participants can compensate for an unavoidable perceptual distortion originating at an early sensory stage by incorporating knowledge about this internal distortion at an intermediate stage of perceptual inference, even if they tend to overcompensate (possibly due to incorrect knowledge of the illusion, see Discussion). This supports that the observed MAE compensation reflects genuine perceptual insight rather than alternative, or additional processes manifesting as adjustments at the late response stage.

To further confirm that our results did not reflect a generic compensation pattern, following previous work[39], we performed a pilot experiment (see Methods "Control Experiment") where we replaced the Believe conditions with 'Bias' blocks in which participants were instructed to respond left when uncertain. This response-bias manipulation induced the expected shift in the psychometric curves in the absence of corresponding shifts in the confidence and RT curves, a pattern that was explained by a change in the response-bias parameter $k_{choice}$, and not by $\mu_{likelihood}$ (Supplementary Fig. S9C). These data thus support the notion that perceptual insight can be dissociated from response bias and speak against a generic compensation pattern.

To further validate our winning model of perceptual insight, we examined its ability to capture RTs not used for model fitting (Fig. 5E). We assumed that RTs reflected decision difficulty based on the perceptual-decision uncertainty, $-|d|$, derived from the model. We thus tested whether ranked $|d|$ could predict ranked RT on a trial-by-trial basis using a generalized linear mixed-effect model (GLME) controlling for condition (see Methods "Statistical analyses"). Indeed, greater uncertainty, smaller $|d|$, correlated with longer RT ($t_{10643} = -10.023, p < 0.001$, 95% CI [-11.983, -8.063], linear mixed-effects model), providing a good explanation for the RT curves across conditions; this GLME showed no significant effects of condition on RT (all $t < 1.153, p > 0.249$; Fig. 5E). Furthermore, this GLME provided a better fit ($R^2_{adjusted} = 0.615$ and indices AIC = 191610, BIC = 191770) than one predicting ranked RT from the ranked objective stimulus strength $|s|$ ($t_{10643} = -8.927, p < 0.001$, 95 % CI [-10.887, -6.967], linear mixed-effects model, with $R^2_{adjusted} = 0.541$ and indices AIC = 193390 and BIC = 193540) and condition, which in contrast did show residual condition effects ($t_{10643} = 3.092, p = 0.002$, 95 % CI [1.294, 5.777] for Adapt-See and $t_{10643} = 5.143, p < 0.001$, 95 % CI [3.556, 7.937] for Adapt-Believe). Therefore, RTs could be parsimoniously explained by decision uncertainty reflecting uncertainty of posterior beliefs in the perceptual-insight model.

## Pupillometry further validates the perceptual-insight model

Arousal-related pupil dilation tracks with perceptual-decision uncertainty and relies on norepinephrine and brainstem circuits distinct from motor execution of button presses[33,46–54], and thus from RTs in this task. We used pupillometry as an objective physiological readout of intermediate processing stages (as opposed to late-stage processes involved in planning and executing the button-press response) to corroborate our interpretation of the data in terms of perceptual insight. We specifically hypothesized that pupil dilation in the relevant task periods (pink and yellow regions in Fig. 6A; see Methods "Experiment 2", "Eye tracking and pupillometry") would mirror RTs and more directly reflect the internal perceptual-decision uncertainty, $-|d|$, variable from the perceptual-insight model.

Indeed, during relevant decision-related periods where the sluggish pupil responses should be most apparent, the pupil dilation patterns roughly mirrored the confidence and RT curves, showing subtly yet visibly shifted pupil-dilation peaks for Adapt-See versus Adapt-Believe (Fig. 6B)–a remarkable difference considering that these conditions were matched on the adaptor and stimuli and only differed in the required response. These differences manifested as statistically significant interactions between test stimulus speed and condition

## Table 1 | Bayesian model variants and their parameters

| Stage | Model | | | | | |
|---|---|---|---|---|---|---|
| | Perceptual insight "$\mu_{likelihood}$" | "prior" | "$\mu_{likelihood}$ + prior" | Late compensation "$k_{choice}$" | "$\mu_{likelihood}$ + $k_{choice}$" | "$k_{choice\ confidence}$" |
| Early | $\mu_{encoding} \times 2$ (fixed) $\sigma_{encoding} \times 4$ | $\mu_{encoding} \times 2$ (fixed) $\sigma_{encoding} \times 4$ | $\mu_{encoding} \times 2$ (fixed) $\sigma_{encoding} \times 4$ | $\mu_{encoding} \times 2$ (fixed) $\sigma_{encoding} \times 4$ | $\mu_{encoding} \times 2$ (fixed) $\sigma_{encoding} \times 4$ | $\mu_{encoding} \times 2$ (fixed) $\sigma_{encoding} \times 4$ |
| Intermediate | $\mu_{likelihood} \times 4$ | prior $\times 4$ | $\mu_{likelihood} \times 4$ prior $\times 4$ | | $\mu_{likelihood} \times 4$ | |
| Late | $k_{confidence} \times 4$ | $k_{confidence} \times 4$ | $k_{confidence} \times 4$ | $k_{confidence} \times 4$ $k_{choice} \times 4$ | $k_{confidence} \times 4$ $k_{choice} \times 4$ | $k_{confidence} \times 4$ $k_{choice} \times 4$ |

The cells show the model names and the parameters of each of the model variants, as well as their associated stages. × 4 means one parameter for each condition: No-Adapt-See, No-Adapt-Believe, Adapt-See and Adapt-Believe. × 2 (fixed) means that the $\mu_{encoding}$ parameters are fixed to the values fitted from the psychometric curves for No-Adapt-See and Adapt-See. In the last model, $k_{choice\ confidence}$, the $k_{choice}$ factor depicted in Fig. 1 is allowed to shift simultaneously the confidence curves (as if apart from the arrow depicted from $k_{choice}$ to C there was another arrow from $k_{choice}$ to q).

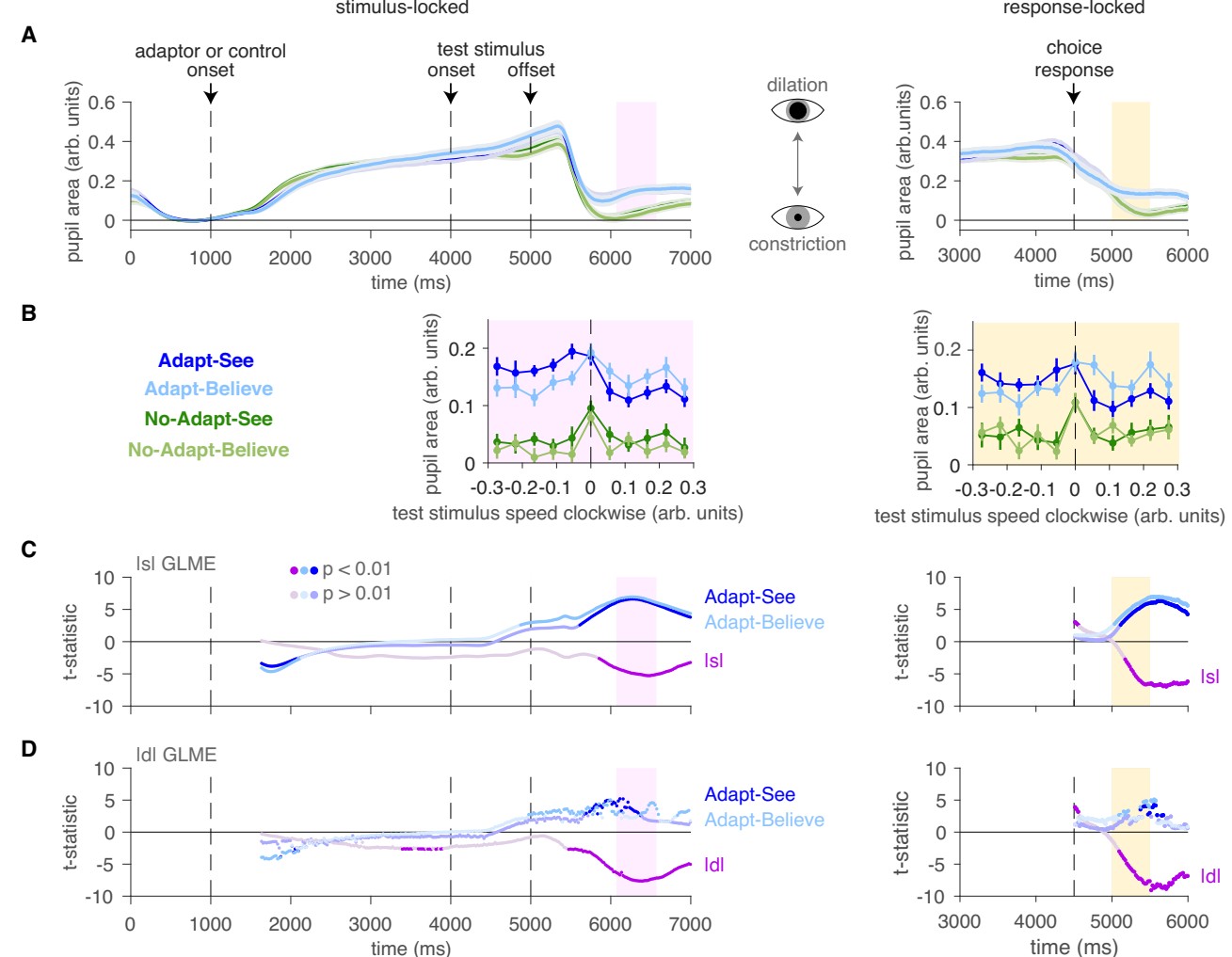

**Fig. 6 | Pupillometric signature of perceptual-decision uncertainty supports the perceptual-insight model. A** Time series of normalized pupil area (mean ± SEM) across trials ($N_{trials} = 121$) and participants ($N = 22$) by task condition. Left: stimulus-locked data (left) showing a decision-related window (pink). Right: response-locked data showing a decision-related window (yellow). **B** Pupil dilation peaks (mean ± SEM) as a function of stimulus strength show subtle yet visible shifts between Adapt-See and Adapt-Believe (mirroring confidence and RT data in Fig. 4). Here, SEM was calculated as in ref. 147 to account for within-participant effects.

**C, D** Moving-window GLMEs showing the fixed-effect t-statistics time series (see Methods Statistical analyses). Brighter colors indicate statistical significance at $p < 0.01$. **C** The effect of absolute objective stimulus strength $|s|$ is apparent around the decision-related periods. **D** The effect of model-derived perceptual-decision uncertainty $|d|$ (from the winning model) is apparent around the decision-related periods, and is stronger than that for $|s|$ in (**C**). Effects for Adapt-See and Adapt-Believe (relative to No-Adapt-See) are weaker in (**D**) than in (**C**).

(Adapt-See, Adapt-Believe) in two-way repeated-measures ANOVAs in both a stimulus-locked decision-related period (2000 – 2500 ms after stimulus onset, $F_{(10, 210)} = 4.105$, $p < 0.001$, $\eta^2 = 0.016$) and in a response-locked decision-related period (500 – 1000 ms post

response, $F_{(10, 210)} = 3.073$, $p = 0.001$, $\eta^2 = 0.016$). Thus, pupil dilation patterns during relevant decision-related periods differed between Adapt-See and Adapt-Believe, possibly consistent with shifts in perceptual-decision uncertainty.

To test this further, in parallel to our RT analyses above, we ran a GLME predicting ranked pupil area on a trial-by-trial basis as a function of ranked $|d|$ and condition during the stimulus-locked decision-related period. As expected, smaller $|d|$ indicating greater uncertainty predicted greater pupil area ($t_{10643} = -7.631$, $p < 0.001$, 95% CI [-9.592, -5.671], $R^2_{adjusted} = 0.376$). To control for response contamination, we next performed analyses of response-locked changes in pupil area during the decision-related period, which confirmed the effect of $|d|$ ($t_{10643} = -5.389$, $p < 0.001$, 95% CI [-7.348, -3.428], $R^2_{adjusted} = 0.346$), even when controlling for RT ($t_{10642} = -2.693$, $p = 0.007$, 95% CI [-4.653, -0.732], in a model which showed a better fit $R^2_{adjusted} = 0.386$). Generally, $|d|$ was more closely related to pupil area than the objective stimulus strength $|s|$ ($t_{10643} = -3.835$, $p < 0.001$, 95% CI [-5.795, -1.875], $R^2_{adjusted} = 0.319$), with GLMEs featuring $|s|$ showing poorer fits and larger condition effects than the corresponding GLME above featuring $|d|$. Finally, these results were confirmed in moving-window GLMEs (see Methods Statistical analyses) showing strong effects of $|d|$ at the relevant time periods. Response-locked analyses confined to the matched Adapt conditions (Adapt-See and Adapt-Believe) showed consistent results for $|d|$ ($t_{5321} = -6.073$, $p < 0.001$, 95% CI [-8.033, -4.112], $R^2_{adjusted} = 0.297$) that held even when controlling for RT ($t_{5320} = -2.883$, $p = 0.004$, 95% CI [-4.843, -0.922], $R^2_{adjusted} = 0.381$), demonstrating key model-predicted changes in the pupillometric signature of $d$ that were distinct from RT changes. Leveraging objective pupil dilation data to index the subjective perceptual-decision variable[33,51], we thus confirmed the predicted changes in the pupillometric signature of this internal variable between Adapt-See and Adapt-Believe, further supporting the validity of the perceptual-insight model.

## Drift-diffusion modeling supports an intermediate inference-level explanation of perceptual insight

We observed an MAE compensation that manifested as shifts in tandem in psychometric and confidence curves (Fig. 4). This pattern was well captured via a Bayesian process model of perceptual insight that was jointly fitted to choice and confidence responses and explained insight via shifts in a perceptual-decision variable at an intermediate inferential stage. However, this model assumed shared computations across See and Believe confidence reports and one of several possible architectures (see Methods "Computational models",[55–58]).

To assess the robustness of our conclusions, we used an alternative framework based on drift-diffusion models (DDMs). To avoid strong assumptions about confidence generation (as in some DDM formulations[59]), we opted for standard DDMs that jointly model choice and RTs, and which have proved useful to parse biases in decision-making[60,61]. In its classic form, the DDM assumes that evidence supporting one of two decisions is initially unbiased or biased (with a starting-point bias of 0, or different than 0, respectively), and accumulates over time with speed determined by the *mean drift rate* and tracked via a noisy decision variable at an intermediate stage. When this decision variable reaches one of two *decision bounds* at a later stage, after some motor preparation and production time (part of *nondecision time*), the observer reports their decision.

Previous DDM work has suggested that decision-making biases can result either from changes in the starting-point bias of the evidence-accumulation process or biases in the drift rate of accumulation towards one decision. While a non-zero starting-point bias would favor a specific choice by starting closer to one decision bound, a drift-rate bias would increase the rate of evidence accumulation towards one decision. These two scenarios have dissociable patterns on the conditional response function (Fig. 7A) of the choice bias as a function of RT quantiles: under a starting-point bias, the choice bias manifests at short RTs and disappears quickly as RTs increase; under a drift-rate bias, the choice bias decays slowly with increasing RTs. Potentially consistent with a starting-point bias, we saw that biases in

the conditional response function plots in Adapt-See (MAE) and Adapt-Believe (MAE compensation) were most apparent at shortest RTs and disappeared relatively quickly with increasing RTs (Fig. 7A).

To quantitatively parse the source of the choice bias, we fit DDM variants using a previously validated method[62]. We considered a "base" variant including 3 free parameters (per condition) for mean drift rate, nondecision time and decision bound and extended variants including a free parameter (per condition) for starting-point bias, one for drift-rate bias, or both (Fig. 7B). The model with starting-point bias as an additional parameter per condition fit the data best across all conditions according to BIC (Fig. 7B); the winning model captured the choice and RT data reasonably well (Fig. 7C) for the limited number of trials for DDM analysis[63]. The parameters of the winning model were informative (Fig. 7D). First, Adapt-See differed from No-Adapt-See in the mean drift rate and starting-point bias, partially consistent with previous work[63] and with sensory-level explanations for MAE[41]. Critically, Adapt-See and Adapt-Believe differed in the mean drift rate and in the starting-point bias. Differences between Adapt-Believe and Adapt-See in the mean drift rate correlated with corresponding differences in $\sigma_{encoding}$, also indexing sensory noise in the perceptual-insight model (Fig. 5C, $\rho = -0.578$, $p = 0.006$, 95% CI [-0.814, -0.122]). Most critically, differences in the starting point correlated strongly with the degree of MAE compensation ($\rho = 0.851$, $p < 0.001$, 95% CI [0.669, 0.946]) and changes in the perceptual-insight $\mu_{likelihood}$ parameter ($\rho = -0.885$, $p < 0.001$, 95% CI [-0.961, -0.724]); differences in the starting point did not significantly correlate with changes in other parameters from the perceptual-insight model (both $p > 0.864$). A bias in the starting point of perceptual evidence accumulation (obtained via joint choice-RT fitting) thus corresponded with the shift in the perceptual-decision variable in the perceptual-insight model (obtained via joint choice-confidence fitting) that explained MAE compensation. Importantly, we found no statistically significant differences in decision bounds between conditions (finding such differences may have supported a late-stage response process). Overall, our DDM results support an implementation of perceptual insight at an intermediate stage of perceptual decision-making, and argue against a mechanism confined to a late-response stage.

## Discussion

We have presented a formal modeling framework that portrays perceptual insight as the ability to acknowledge distortions in one's internal representations, allowing online adjustments in support of adaptive decision-making. We further designed a controlled "cognitive psychophysics" paradigm[38] to capture this form of in-the-moment reality testing while minimizing memory demands. We first demonstrated that people can compensate—and often over-compensate—for distorted percepts associated with a complex MAE illusion, showing that they can adjust their decisions to counteract the MAE illusion. We then leveraged confidence reports and RTs[31,32] to show that MAE compensation involved shifts in decision uncertainty more consistent with adjustments at an intermediate inferential stage. Model comparison lent further support for a model of perceptual insight involving adjustments at this inferential level. The shifting perceptual-decision variable in our model provided a parsimonious explanation for RT and pupil-dilation patterns, providing further support for our model of perceptual insight and suggesting humans' ability to deploy such insightful strategies to circumvent internal biases. Further support for an interpretation of perceptual insight in terms of the hypothesized shifts in the perceptual-decision variable came from DDM analyses (jointly fitting choice and RTs but not confidence reports), suggesting robustness of our conclusions to assumptions about confidence generation.

As the adage "seeing is believing" implies, conscious perceptual experiences generally dictate one's beliefs. In contrast, our results exemplify a rare case of a compensatory strategy that systematically

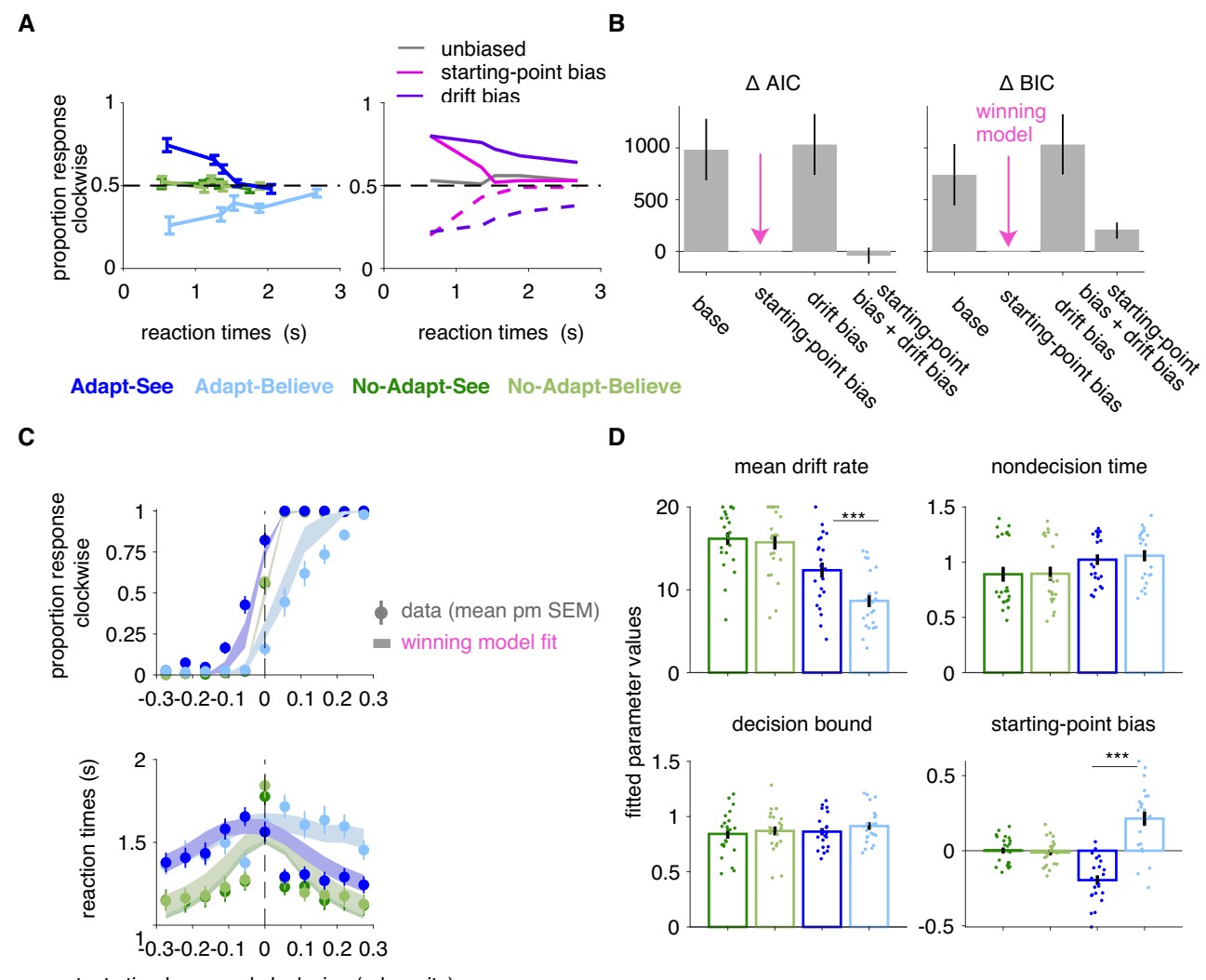

**Fig. 7 | Drift-diffusion modeling. A** Conditional response function[60, 61] for our data (left, mean ± SEM across the *N* = 22 participants for each bin per condition) and schematic (right, adapted from[60, 61]) illustrating differential effects of starting-point biases versus drift-rate biases. RTs are binned into 5 quantiles. **B** Using the PyDDM package[62], we implemented a base DDM (with parameters mean drift rate, decision bound and nondecision time) as well as variants with additional parameters (starting-point bias, drift-rate bias or both for each condition). Model comparison selects the starting-point bias DDM variant as the winning model. Bar plots represent summed ΔAIC and ΔBIC across participants, and the error bars are 95% bootstrapped confidence intervals over 1000000 samples with replacement. **C** Fits of the winning DDM with starting-point bias capture choice and RT data satisfactorily. Dots end error bars depict mean data ± SEM across the *N* = 22 participants for each bin and shaded areas depict model predictions, mean ± SEM. **D** Fitted parameter values from the winning DDM model above (mean ± SEM across the *N* = 22 participants from Experiment 2). *** indicates *p* < 0.001, with results being based on planned post-hoc two-sided *t*-tests.

prevents conscious perceptual experience—of the MAE—from driving beliefs about the true state of the world[64–66]. Put simply, they show that seeing need not imply believing. Under our operationalization, perceptual insight is at the center of this distinction: perceptual insight allows for a flexible mapping between seeing and believing because it incorporates introspective knowledge that internal experiences depend upon factors other than external stimuli and therefore do not always represent the external world faithfully. As such, perceptual insight may enable reality testing. The systematic dissociation between perceptual experience and belief we achieved empirically, and our multi-stage perceptual-insight framework, conceptually relates to previous hierarchical views of conscious perception and metacognition; these views have generally proposed that beliefs about reality arise from higher-level processes (possibly supported by higher-order prefrontal regions such as anteromedial prefrontal or paracingulate cortex involved in source monitoring and metacognition[8,11,67]) distinct from lower-level

processes supported by earlier sensory regions. Our observed dissociation between perceptual experience and beliefs seems consistent with this notion, and could perhaps suggest the involvement of higher-order regions in monitoring and *correcting* biases in lower-order sensory representations. Note however that our algorithmic model does not speak directly to neural implementation and that the 'intermediate-level processes' we refer to may encompass different neural hierarchical levels—e.g., from posterior parietal regions routinely involved in perceptual inference[68], to above-mentioned prefrontal regions which could be more selectively involved in insightful inference requiring top-down signals for sensory bias correction. Our computational account of reality testing is also conceptually related to a previous proposal based on generative adversarial algorithms[69,70]. Future work is warranted to evaluate and directly compare our framework to these theoretical and neuroanatomical accounts, as well as to other experimental paradigms of reality testing[8,71–73].

The assumptions of our perceptual-insight model are broadly consistent with previous theoretical and empirical neuroscience work. As in previous work, the assumption that the MAE predominantly affects the early stage of sensory encoding is based on the observed decreases in neuronal firing rates[44,74] and fMRI signals in motion-selective sensory regions under MAE[75]. Further, microstimulation of these sensory regions produces shifts of psychometric, confidence, and RT curves in tandem[76,77] similar to those we and others[31,32] observed during the MAE. We also assumed that the offset in the sensory measurement under MAE could not be (fully) corrected, since adaptation produces changes in the membrane potential of neurons[41–45] unlikely to be readily malleable. Moreover, the MAE illusion was experienced regardless of explicit knowledge about its illusory nature and of the required (See or Believe) responses. Given this, and consistent with recent work[78], we interpreted the observed shifts in tandem associated with MAE compensation as changes at an intermediate perceptual-decision stage ('intermediate' here broadly referring to a process preceding response implementation). Consistent with this interpretation, microstimulation of decision regions such as the lateral intraparietal area (LIP) also produces shifts in tandem[79]. An explanation in terms of late compensation seemed less tenable given that changes in the response rule at late stages—likely implemented in distinct downstream regions[80,81]—can manifest as isolated shifts in psychometric curves[39], (as reproduced in our response-bias pilot experiment; see Supplement S1.4, Fig. S9), in contrast to our main results. DDM analyses of choice and RT data—free of assumptions on confidence construal or generative processes—similarly favored a perceptual-decision bias during MAE compensation and weighed against a late-response process. Beyond pointing to an intermediate-stage process, our results supported a model whereby perceptual insight reflects the incorporation of knowledge that factors other than external stimuli (here, the adaptor) influence internal representations. Modeling suggested this knowledge was incorporated at the level of the likelihood term in the inferential process, or equivalently at the level of decoding of the distorted internal representation. Although speculative, this process could be neurally implemented in decision regions (e.g., LIP) akin to optimal-decoding solutions based on synaptic reweighting of sensory representations[82–87]. Alternatively or additionally, insightful inference may require higher-order prefrontal regions involved in metacognitive processes[8,67,88].

Potential limitations include our reliance on the dissociation between psychometric-curve shifts in tandem with confidence and RT curves versus isolated shifts in psychometric curves to arbitrate between intermediate inferential biases and late-response biases, respectively. While previous findings (and our pilot results; Fig. S9) are consistent with such dissociation[31,32,39,89], other results are less clear[40]. Nonetheless, our interpretation of MAE compensation in terms of an inferential process (incorporating introspective knowledge about the internal distortion) is further supported by evidence (1) that our participants had detailed knowledge about the MAE illusion (its direction and approximate strength; Fig. S1); (2) from our model comparison favoring the perceptual-insight model even over other inferential models (e.g., category prior model; Fig. 5B, Fig. S5); (3) from the pupillometric data that more likely reflect internal inferential processes than response-related processes, and which held after controlling for RTs (more likely reflecting the latter); and (4) from DDM results showing that MAE compensation related strongly to biases in the starting point for the perceptual-decision variable but not to changes in decision bounds associated with response termination, consistent with shifts in inferential processes and not with changes limited to response implementation (for a similar interpretation, see ref. 61). In Experiment 2, but not statistically in Experiment 1, participants overcompensated for the MAE, a result we did not predict. This, together with a lack of calibration between expected illusion strength and MAE compensation (Fig. S1), could suggest a suboptimal compensation strategy. However, model comparison favored the optimal perceptual-insight model over other potential alternatives, including suboptimal compensation strategies at intermediate (e.g., category prior model) or late (e.g., late compensation) stages which could still reflect some form of insight—understood broadly in the sense of having knowledge of the illusion and attempting to use that knowledge to adjust decisions. But how could participants over-compensate if they were presumably using an optimal insightful strategy consistent with the perceptual-insight model? One potential explanation is that they employed the correct strategy and generative model but nonetheless used incorrect expectations about MAE illusion strength, which could have been plausibly induced by our experimental design (due to our pre-task instruction using longer-lasting adaptors to illustrate the MAE, which was particularly emphasized in experiment 2; see Methods "Experiment 1" and "Experiment 2"). Future work systematically manipulating expectations about MAE strength is needed to test this directly. Furthermore, the interindividual variability in the MAE compensation hints at a variety of factors—some explicitly identified in our perceptual-insight model, including misestimation of the distortion strength or uncertainty—potentially driving overuse or underuse of introspective knowledge in health and illness. We discussed that the category prior model differs from the perceptual insight model in that the former incorrectly assumes a change in the category prior that produces fixed shifts in $d$ regardless of encoding noise (Supplementary Fig. S5), and this may be tested in future experiments through manipulations of encoding noise (for instance via changes in stimulus contrast). Finally, future work should improve upon the current task design to further minimize potential order effects (although see Methods "Experiment 1" and "Experiment 2", "Experiment structure and block order") and to allow comparisons between pre- and post-acquisition of knowledge about the MAE.

A key future goal of this research is to advance our understanding of elusive insight impairments that are central to psychosis[1,2,4]. In this context, our model may suggest that impaired insight in psychotic individuals stems from a failure to recognize that, in contrast with lifelong experience, their internal representations do not match the external world. This failure could thus relate to the sort of inflexible belief-updating proposed to underlie other aspects of psychosis[90–92]. Overall, this work will hopefully provide a foundation for the quantitative study of insight impairments and contribute to the development of objective markers, e.g., based on pupillometry[93,94].

In conclusion, we have developed a modeling framework and a controlled psychophysics paradigm that captures perceptual insight as a quantifiable compensation for distorted perception. Our results collectively suggest that perceptual insight can be used to counteract upstream distortions in sensory measurements via downstream adjustments at an inferential readout stage. By decoupling percepts and beliefs, perceptual insight reveals a key interface that may prove helpful in advancing our understanding of introspection and conscious perception.

## Methods
### Experimental approach
Experiments 1 and 2 used variants of our perceptual-insight task to quantify compensation and test our candidate mechanism of perceptual insight. We will first present the core components of our perceptual-insight task based on the motion after-effect (MAE) and discuss small differences across experiments, first in brief and then in more detail.

**Perceptual-insight task.** The task, trial structure and overall experiment structure were mostly shared across the two experiments. The exact task design of Experiment 2 is in Fig. 2. The task requires the participants to perform left/right motion discrimination: is the test spiral rotating counterclockwise (left response) or clockwise (right

response)? In order to measure implicitly the strength of the motion after-effect, we used the nulling method[36,95–97], which entails the presentation of two spirals, the first one being the adaptor, and the second one being the test stimulus. Observers are then required to give their responses about the second test spiral. Stimuli are full screen spiral textures, adapted from ref. 98, with a small circle marker in the center to indicate fixation. As depicted in Fig. 2, on every trial, participants first saw a fixation screen for 1000 ms, followed by a screen of 3000 ms with the first adaptor spiral (moving with a fixed high speed on Adapt blocks or static in No-Adapt blocks), and then a screen for a maximum of 1000 ms with the second test spiral (rotating at variable speed). Afterwards, a first change in the color of the central marker prompted the left/right choice about motion direction, and then another change in its color prompted a confidence report about their choice (very sure [high confidence] or not very sure [low confidence]) (Fig. 2). Overall, the experimental task had a 2-by-2 design -- with two types of conditions No-Adapt (static first spiral) and Adapt (rotating first spiral)—and 2 types of responses required -- which direction participants See the second spiral moving or which direction they Believe the second spiral is actually moving. The behavioral measures of interest shared across both experiments were: choice responses (left/right), confidence responses (high/low) and reaction times.

**Overview of Experiments 1 and 2**. Experiments 1 and 2 were in-person and both measured behavior; Experiment 2 additionally employed eye-tracking. Experiment 2 had three main developments relative to Experiment 1. First, it made use of enhanced instructions which contained more informative feedback. Second, it presented the participants with a fixed set of stimuli drawn from a uniform distribution (with bounds informed by the data from Experiment 1), thus making the data amenable to fitting the proposed Bayesian model of insight. Third, by using eye-tracking, we ensured that participants maintained fixation during the trial and also measured their eye positions and pupil sizes continuously throughout the experiment. Specifically, during the fixation, adaptor and test stimulus periods (a total of ≈ 5 s), if the participants either blinked or moved their eyes relative to the screen away from a circle centered at fixation with radius 3 degrees of visual angle (dva), the trial stopped and the same trial was again presented immediately after until the participant completed it without blinking or breaking fixation. During a 3 s post-stimulus period, if the participants either blinked or moved their eyes away from a circle centered at fixation with radius 24 dva, the trial was also stopped and the same trial was presented again. This approach served to control for the potential influence of eye movements and blinks[99] on the motion after-effect and to rule out certain strategies that could be used to minimize it.

Other minor differences across Experiments 1 and 2 were: Experiment 1 had a slightly shorter fixation window (890 ms vs. 1000 ms) and different colors of fixation circles and choice prompts. Additionally, the adaptor before the blocks was presented for 15 s in Experiment 1 and 30 s in Experiment 2. 5 out of 22 participants from Experiment 1 were presented with the adaptor spiral rotating clockwise; their data was combined with data from the other participants by flipping directions accordingly (such that the reference direction of the adaptor was counterclockwise). Note that all the reported effects were present even if we exclude these 5 participants from Experiment 1.

## Experiment 1
**Participants**. 25 participants were recruited. 3 participants were excluded after the instructions because they did not experience the illusion as their responses in the second part of the training were not consistent with the illusion on a sufficient number of trials (less than 8 out of 10). The remaining 22 participants that completed the task and were used for analyses comprised 11 males, and 11 females, and had a

median age of 26 years old (range 21 to 50). No statistical method was used to predetermine sample size. All participants provided informed consent. The study conformed to the Declaration of Helsinki and was approved by the Institutional Review Board of New York State Psychiatric Institute.

**Apparatus**. The stimuli were displayed on a 13-inch MacBook Pro laptop (2017 model) in a dark psychophysics room. The width of the viewable portion of the screen was 11.97 inches (30.4 cm) and the screen resolution was 2560 × 1600 pixels and 60 Hz refresh rate (1 frame lasting ~16.67 ms). The MacBook Pro laptop had installed Matlab 9.6 (2019a, MathWorks, Massachusetts, USA) with the Psychtoolbox extension, version 3.0.15[100–102]. Matlab and Psychtoolbox controlled the presentation of the stimuli and recorded the participants' responses. Participants were seated at a distance from the laptop of ~20 cm.

**Instructions, demonstration, training**. This section contained instructions and training trials and was divided into 8 parts with a few additional quiz questions. Collectively, these elements were used to gradually build and ensure an adequate construal of the task. In the first part, participants were asked to perform 10 trials of left/right motion discrimination of a single rotating spiral and received feedback after each trial. Each participant was asked to perform this part until performance was higher or equal than 70% correct, for a maximum of 3 times. The second part entailed 10 trials showing two sequential spirals, the first one rotating for 3 s and the second one static for 1 s, and participants were asked to report whether they saw the second spiral moving left or right. Here they received feedback about whether the responses were consistent with the motion-after effect illusion. The participants were asked to perform this part until performance was consistent with the illusion in 80% of the trials, for a maximum of 3 repetitions. The third part repeated the structure of the second part, but importantly entailed the presentation of a sound to mark the onset of the second spiral. Participants were encouraged to convince themselves that the second spiral was indeed static and that looking at the first moving spiral caused them to perceive the second static spiral as moving in the opposite direction. The fourth part built on the third part by introducing motion in the second spiral and asking the participants to report across 10 trials whether they saw the second spiral as moving left/right. Participants were told not to be concerned with the actual motion in the second spiral, foreshadowing that this would be relevant to the 'Believe' condition introduced next. The fifth part introduced the Believe condition. Across 10 trials, participants reported which direction they believed the second spiral was actually moving. To ensure they had explicit knowledge about the illusion, participants were then asked to respond to the following questions: "Imagine that the first spiral was moving [clockwise/counter-clockwise], and then you saw the second spiral as still. Which way do you believe the second spiral was actually moving?". The sixth part introduced the confidence responses ("very sure"/"not very sure") to augment the left/right responses when discerning the direction of motion of a spiral across 10 trials. Participants were told on how many trials they responded "not very sure" so they could reflect on their confidence responses and attempt to balance them, vs. disproportionately responding with just one button (i.e., 9 vs. 1). The seventh part again presented two spirals and asked participants across 5 trials to report the direction they saw the second spiral moving and then their confidence about this response. The eighth part finally asked participants across 5 trials to report the direction they believed the second spiral was moving and their confidence about this response.

**Payment**. In Experiment 1 participants received $30 for the first hour and $20 for each additional hour. They were instructed to try their best in all conditions and incentivized with a bonus of $10 determined

based on adequate compliance with task instructions and the experimenter's examination of the data, including consistency in behavior.

**Stimuli.** Adaptors of complex motion (e.g., rotational) have been shown to induce stronger adaptation effects relative to ones of simple motion (e.g., translational)[34,35]. We aimed to choose spiral stimuli to achieve the strongest motion after-effect (or spiral after-effect) illusion with the shortest presentation times of adaptor and test spirals. Within spiral stimuli, we considered their type (Archimedean[103] vs logarithmic[104,105]), spatial frequency and temporal frequency[106,107]. Based on the literature and pilot data, we decided to use Archimedean spirals, with spatial frequency parameters $w_1 = 30$ and $w_2 = 3$. A $w_1$ value of 30 corresponds to 30 pixels being traversed during a cycle. Based on $w_1$, screen width, the distance from the screen and the screen resolution, we calculated the spatial frequency to be 1.16 cycles/°.

The Archimedean spiral equation is:

$$\text{spiral} = \frac{\text{white}\left(1 - \cos\left(\frac{r}{w_1} + \theta w_2\right)\right)}{2} \quad (1)$$

A third parameter, which captured the temporal frequency of the motion of the spiral, was set to $w_3 = 9$ for the adaptor and varied for test stimuli (Experiment 1: [-2,2] and Experiment 2: [-0.3,0.3]), thus determining the velocity or speed of spirals, as the spatial frequency was always the same across trials (see Equation (2)). To implement rotational motion, the spiral was offset relative to the previous screen by a particular angle value, determined by $w_3$. Given the refresh rate of 60 Hz, the temporal frequency was 1.5 cycles/s. Velocity (here used interchangeably with speed) of the adaptor spiral, given by the ratio of the temporal frequency and the spatial frequency, was 1.293°/s.

$$\text{velocity}(°/s) = \frac{\text{temporal frequency (cycles / s)}}{\text{spatial frequency(cycles /°)}} \quad (2)$$

We implemented the adaptor and test spirals based on ref. 98. In contrast to some other studies (i.e., refs. 108–110), we decided to keep the contrast of the adaptor and the test spiral both equal to the maximum (1), to avoid introducing interindividual differences due to contrast sensitivity and luminosity.

**Test stimulus generation.** The speed of the test stimulus was based on the participant's previous responses according to an adaptive procedure, a type of Bayesian staircase, applied separately for each of the 4 conditions. Each one of the 4 conditions contained 120 trials, divided into 2 blocks of 60 trials each. We used the Matlab implementation[111], based on[112] with extensions to include the lapse rate[113,114] (this exact staircase was used in ref. 115). This procedure maintains a posterior distribution over the parameters and updates it after each trial. The next stimulus value is chosen to minimize the entropy of the updated posterior given the stimulus, averaged over the participant's possible responses weighted by their respective probabilities[112]. We defined the space of parameters for the computation of the posterior distribution in[111] as follows: for $\mu$, we used a linear grid of 51 points from -0.5 to 0.5, for $\sigma$ a logarithmic grid of 25 points from 0.001 to 0.5, and for $\lambda$ a linear grid of 25 points from 0 to 0.3.

The test stimuli were substantially slower than the adaptor. Each one of the 4 staircases generated on every trial a value $w_3$ within the range $[-2, 2]$ that dictated the temporal frequency and the velocity (speed) of the test stimulus. The test stimulus values thus corresponded to velocities in the range [-0.285°/s, 0.285°/s]. The actual values selected through the staircase and used in the experiment had a narrower range (typically within [−0.5, 0.5] in arb. units; see Fig. 3A, top). These values amount to an adaptor/ test velocity ratio of 4.5 or higher, consistent to some extent with values in the range [2, 5] which

were shown to achieve the maximum motion after-effect in previous work[83].

**Trial structure.** Trials had the basic structure described in Fig. 2. Participants first saw a fixation screen of mid-level gray (RGB: [128 128 128]) for 890 ms with a central small white circle. This was followed by a screen of 3000 ms with the first adaptor or control spiral (moving with a fixed high speed or static), which was followed by a screen lasting a maximum of 1000 ms (or until the participant responded if the reaction time was under 1000 ms) with the second test spiral (moving with variable speed as determined by the adaptive staircase procedure). The transition from the first (adaptor or control) spiral to the second spiral (test stimulus) was marked by the change in color of the small fixation circle from white to either yellow, for the See conditions, or blue, for the Believe conditions. See and Believe trials were blocked (see below), but this color coding provided an additional reminder of the required responses. If the participant submitted a left/right choice response during the 1000 ms when the test stimulus was on the screen, the next screen was immediately presented featuring a pink-colored fixation circle prompting for a confidence response, specifically 'how sure are you of your response?" (requiring an up/down key press). If the participant did not respond within 1000 ms, the test stimulus screen was replaced with a gray screen with no changes in the fixation-dot color (yellow or blue depending on the trial type) which was displayed until response. There was no deadline, making the task self-paced. After response, the screen with the pink-colored fixation circle prompted for the confidence response until this was submitted. In Experiment 1, if the total post-stimulus time (including the left/right choice and up/down confidence response times) was shorter than 2000 ms, the fixation screen was presented to ensure a minimum post-stimulus total time of 2000 ms.

**Experiment structure and block order.** Before the experimental task trials, participants performed a short illusion-reproduction task. They saw a 15 s rotating spiral followed by a second static spiral and were asked to remember the strength of the illusory motion on the static spiral. Afterwards, they were asked to reproduce the strength of the illusion in 10 consecutive trials using the computer trackpad. During the reproduction, they saw the static spiral with a green fixation circle and controlled its speed using the trackpad (scrolling away from the center to control direction and speed) until the speed of the static spiral matched that of the illusory motion they had just experienced. These illusion-reproduction estimates were collected once again after the Adapt blocks, and together they served as confirmation that participants had explicit knowledge about the motion-after effect illusion, including its direction.

The main task had a fixed order and was structured as follows: Adapt blocks were presented first, including two Adapt-See blocks followed by two Adapt-Believe blocks, each starting with an extended 15 s presentation of the adaptor spiral (as in refs. 37,63) before the presentation of the experimental trials. These 4 Adapt blocks were followed by 10 more trials of illusion reproduction. Afterwards, participants took a break of at least 10 minutes to minimize any possible lasting effects associated with adaptation. After the break, participants performed the No-Adapt blocks: two No-Adapt-See blocks and then two No-Adapt-Believe blocks, each starting with an extended 15 s presentation of a control static spiral. Adapt blocks were presented before the break to reduce fatigue for the active conditions (although 3 participants performed No-Adapt blocks before the break instead and we did not observe systematic differences between these and the rest other than general speeding of responses as the experiment progressed). The fixed order of blocks also served to remind participants of the block-relevant instructions before starting the experimental trials and required responses and to minimize switching costs.

## Experiment 2

**Participants.** 26 participants were recruited. One participant was excluded due to medication, 2 because they did not experience the illusion, and one dropped out without completing all procedures. The 22 completers comprised 6 males and 16 females, and had a median age of 24.5 years (ranging from 18 to 34 years old). No statistical method was used to predetermine sample size. All participants provided informed consent. The study conformed to the Declaration of Helsinki and was approved by the Institutional Review Board of New York State Psychiatric Institute.

**Apparatus.** The experiment took place in a dark psychophysics room. The computer used was a Mac mini (Late 2014 model) with a 3 GHz Intel Core i7 processor, 16 GB 1600 MHz DDR3 memory and Intel Iris 1536 graphics. The width of the viewable portion of the screen was 59 cm and the screen resolution was 1920 × 1080 pixels and 60 Hz refresh rate (1 frame lasting 16.67 ms). The Mac mini had installed Matlab 8.3 (2014a, MathWorks, Massachusetts, USA) with the Psychtoolbox 3.0.13 extension[100–102].

In addition to other procedures similar to Experiment 1, in Experiment 2 we also monitored participants' fixation and recorded their eye movements and pupil sizes. We used a remote infrared video-oculographic system (EyeLink 1000 Plus version 5.15; SR Research, Ltd, Mississauga, Ontario, Canada) with a 1 kHz sampling rate. Participants were seated such that the screen distance (eye to center of the screen) was ~66 cm. For the majority of the participants (17/22), we set the heuristic filtering option 'ON'. For the 5 participants for which we did employ the Eyelink's online heuristic filter, we implemented a post-hoc Kalman smoother on the x and y eye position time series as in ref. 116. The eye tracker was calibrated using the 9-point calibration routine before every block.

**Instructions, demonstration, training.** This section contained the same 8 parts as in Experiment 1 in addition to a new part and other enhancements. The instructions were enhanced (based on our experience and feedback from participants in Experiment 1) and provided more detailed and informative feedback so that participants could have more precise knowledge about the illusion. After the first three parts transpired as in Experiment 1, participants first experienced how they can control the speed and direction of a static spiral by moving the mouse farther away to the right or to the left. In the fourth part, participants practiced 10 trials of illusion-reproduction. This was not included in the training from Experiment 1, just in the Experiment 1 itself. As in Experiment 1, participants saw the first rotating spiral for 30 s, then a static spiral and were told to remember the strength of their illusion. Note that this was done to clearly illustrate the MAE illusion, although it may have induced expectations of stronger MAE than that experienced during the task following shorter adaptors. Then they saw 10 static spirals each with a green fixation circle and each time controlled its speed using the mouse until the speed of the static spiral matched that of their illusory motion. This served as training for the illusion-reproduction trials from the experiment itself. In the fifth part, participants saw an adaptor spiral followed by the test spiral and performed motion discrimination, as in Experiment 1. Here however they also received feedback with the actual velocity of the test spiral, presented as the percent of the speed of the adaptor spiral, and the direction of motion (same or opposite) relative to the adaptor spiral. In the sixth part introducing the Adapt-Believe condition (corresponding to the fifth part from Experiment 1), participants saw an adaptor spiral followed by a second test spiral with different motion speeds and reported their beliefs about its true motion. Here, participants received feedback on their accuracy based on the true motion of the test spiral and again received information about the actual velocity of the test spiral (as noted in the discussion, this may have helped consolidate a stronger expectation of MAE strength).

Participants next completed the same quiz as in Experiment 1. The last three parts introducing confidence responses were as in Experiment 1, but had more trials (10 vs. 5) and additional feedback of the actual speed of the test spiral relative to the adaptor spiral. The ninth part consisted of 10 Adapt-Believe practice trials. If response accuracy on these trials was below 70%, participants were asked to repeat this set of 10 trials until their accuracy was above 70%, for a maximum of 3 attempts. Overall, the instructions and training for Experiment 2 were similar to those in Experiment 1 and mainly differed in that they contained more detailed feedback and information including the actual relative speed of the second test spiral.

**Payment.** Participants were compensated with $30 for the first hour, and $10 for each additional hour. They were instructed to try to be as consistent as possible in all conditions and as accurate as possible in the Believe conditions. They were incentivized with a bonus of $30. As in Experiment 1, this bonus was determined based on adequate compliance with task instructions and the experimenter's examination of the data, including consistency in behavior.

**Stimuli.** The stimuli were also Archimedean spirals with the same parameters as in Experiment 1: $w_1 = 30$ and $w_2 = 3$. The temporal frequency was the same for the adaptor spiral ($w_3 = 9$) but differed for the test spirals in that was drawn from a uniform distribution between $[−0.3, 0.3]$. The screen width, distance from the screen and the screen resolution were different from those in Experiment 1. Given this, the spatial frequency in Experiment 2 was 1.14 cycles/° and the velocity of the adaptor was 1.5/1.14 = 1.31°/s .

**Test stimulus generation.** In contrast to the adaptive staircase in Experiment 1, Experiment 2 used test stimuli drawn from a uniform distribution (in a pseudorandom order fixed across participants). The bounds of this uniform distribution were informed by the empirical distributions of the test stimuli from Experiment 1. The values were between $[−0.3, 0.3]$, corresponding to test stimulus velocities between $[−0.044°/s, 0.044°/s]$. This range was maintained across conditions to facilitate direct comparison of responses and provide a parametric distribution satisfying assumptions of, and allowing for model fitting based on, Bayesian models. Each of the 4 conditions contained 121 trials, divided into 2 blocks of 60 and respectively 61 trials. We chose 121 trials (vs. 120 in Experiment 1) to include the velocity value of 0, corresponding to a static test spiral.

**Trial structure with gaze fixation control.** As in Experiment 1, trials had the basic structure described in Fig. 2. A major difference was that fixation was enforced for a time window encompassing the fixation, adaptor, and test stimulus periods (5 s total) and blinking was also enforced for this time window plus an additional post-stimulus interval of 3 s, for a total of about 8 s. If participants blinked or deviated their eyes outside a circle of 3 dva centered at the fixation central dot during this period (fixation, adaptor or test stimulus) or blinked or deviated their eyes outside a circle of 24 dva during the post-stimulus period, the trial was immediately interrupted and the same trial was repeated until responses were provided without blinks or breaking fixation. Participants were encouraged to keep their eyes within the fixation area throughout the trial and were encouraged to blink at the end of their trial and briefly rest their eyes as needed. The moment they decided to resume fixation dictated the initiation of the next trial.

Participants first saw a fixation screen of dark gray (RGB: [85 85 85] vs. [128 128 128] in Experiment 1) for 1000 ms (vs. 890 ms in Experiment 1) with a central gray dot of RGB [28.7 28.7 28.7], equivalent to [10 0 0] in CIE-L*ab color space, with $L = 10$. The darker color of the fixation screen (RGB [85 85 85] equivalent to [36 0 0] in CIE-L*ab, with $L = 36$) was close in overall luminance to the spiral texture, which minimized spurious changes in pupil size due to luminance and minimized eye-

muscle strain under the more demanding fixation-enforcement procedure. Next, a screen of 3000 ms with the first adaptor or control spiral was presented. This was followed by a screen lasting a maximum of 1000 ms with the second test spiral (moving with variable speed). The transition from the first spiral (adaptor or control) to the second spiral (test stimulus) was marked by a change in color of the small fixation circle, from gray to a yellow of RGB [81 75 0] and respectively [31 -6 40] in CIE-L*ab, with $L = 31$, close to the luminance of the gray of the screen ($L = 36$). This yellow dot signaled the onset of the test spiral in both See and Believe conditions (unlike in Experiment 1, where different colored dots were used for See vs. Believe) to keep stimulation identical between these conditions. In Experiment 2, even if the participant made a left/right response about the test stimulus faster than 1000 ms, the test stimulus was always kept on screen for the entire 1000 ms (precisely 1033 ms due to the refresh rate). If the participant did not respond within this window, the test stimulus screen was replaced with the same gray screen maintaining the yellow fixation circle until response. After the left/right response was recorded, a new screen with a blue fixation circle (the shade of blue, RGB [0 85 140] and respectively CIE-L*ab [34 0 -36], with $L = 34$, being approximately isoluminant with the yellow in the previous fixation circle) was presented prompting for a confidence response (requiring an up/down key press). If the total post-stimulus time was shorter than 3000 ms, the fixation screen was presented to ensure a minimum post-stimulus total time of 3000 ms (vs. 2000 ms in Experiment 1). These differences between the experiments ensured that stimulation across conditions was identical between Adapt-See and Adapt-Believe, which allowed direct comparisons of pupil responses (see below).

**Experiment structure and block order.** The experiment structure and block order were comparable to Experiment 1. Adapt blocks started with a 30 s presentation of the adaptor spiral and No-Adapt blocks with a 30 s presentation of the control spiral (vs. 15 s for each in Experiment 1). Adapt blocks were completed before No-Adapt blocks. The only difference from Experiment 1 was that the No-Adapt blocks (No-Adapt-See vs. No-Adapt-Believe blocks first or second) were randomized across participants to minimize fatigue, rushing, and task familiarity effects in the control conditions. Adapt-See blocks were followed by Adapt-Believe blocks in a fixed (non-randomized) order because the Adapt-Believe condition involved all elements in Adapt-See plus maintaining and using information about the illusion, and thus benefited from additional exposure to the basic task setup in Adapt-See.

The fixed order of some of our conditions could in principle have led to differential fatigue or practice effects, and systematic differences in RTs, across conditions, but the results of RT analyses speak against this possibility. First, RT did not differ significantly between conditions in the GLME featuring $|d|$ (all $p > 0.25$). Additionally, in the context of the DDM, fatigue has been argued to produce general slowing of stimulus encoding and response production times, which should be mainly reflected in the nondecision time parameter[117]. Nondecision times from the winning DDM variant showed no differences between Adapt-See and Adapt-Believe (two-sided $t$-test, $t(21) = -0.93$, $p = 0.36$), further arguing against meaningful differences in fatigue between these two critical conditions.

**Task and illusion comprehension checks.** Several checks were used to ensure that participants understood the task and had explicit knowledge about the illusion. First, participants' verbal expression of their understanding of the task and illusion in their own words was deemed to be acceptable by the experimenters. Second, their responses on the second part of the training had to be consistent with the illusion in at least 80% of the trials. If this did not happen within 3 repetitions, participants were paid for their time so far and excluded. Third, responses on key questions of the quiz during training were consistent with their proper understanding of the illusion or prompted

clarification until understanding was established by the experimenter. Fourth, the training from Experiment 2 included a performance criterion of 70% accuracy during Adapt-Believe practice trials that demonstrated directional knowledge of the illusion. Fifth and most definitively, in both experiments participants demonstrated explicit directional knowledge of the illusion in illusion-reproduction trials: the estimated strength of their motion after-effect illusion showed that they expected to experience illusory motion opposite to the direction of the adaptor, reasonable in magnitude, and consistent within participant (Fig. S1).

**Control Experiment**
To test whether observers could potentially employ a compensation strategy distinguishable from the one prescribed by the perceptual-insight model, we performed a pilot control experiment. We specifically used a response-bias manipulation combined with MAE to test whether we could observe a compensation-like pattern in the form of isolated shifts in the psychometric function in line with the late-compensation "$k_{choice}$" model (Fig. 1, Fig. 5B, Table 1) or instead whether the response-bias manipulation would result in similar behavioral patterns (shifts in tandem) to those in Experiment 2. To this end, we used a response-bias manipulation based on previous work (Experiment 1 in ref. 39) in addition to our adaptation manipulation. Specifically, observers performed the No-Adapt-See and Adapt-See conditions as in Experiment 2 and additionally performed No-Adapt-Bias and Adapt-Bias conditions (instead of the Believe conditions) were they were instructed to choose a left counterclockwise response (in the direction that would compensate for the clockwise MAE) when they were uncertain about the direction of the test stimulus, via the instruction: "If unsure, press Left". Instead, during the See conditions in the control experiment (and in all conditions in Experiment 2) the instruction was: "If unsure, guess" (aiming to balance guesses across left and right responses).

For this experiment, 10 participants were recruited. 3 participants were excluded due to low quality of their data, specifically their responses not being captured well by psychometric curves. All participants provided informed consent. The study conformed was approved by the Institutional Review Board of New York State Psychiatric Institute. The payment structure was as in Experiment 2: $30 for the first hour and $10 for each additional hour, plus a $30 completion bonus. The 7 completers comprised 4 males and 3 females, and had a median age of 25 years (ranging from 21 to 47 years old). Participants were seated at a screen distance of 57 cm (versus in Experiment 2, 66 cm).

**Statistical analyses**
All statistical analyses were performed in Matlab R2019a. Assumptions of parametric tests were violated for most of the main variables, so our main analyses consisted of Wilcoxon signed-rank tests (Matlab's `ranksum` and `signrank`) or Spearman correlations. To compute the 95% confidence intervals of statistics—specifically signed-rank tests and Spearman correlations—we used a bias-corrected and accelerated bootstrapping procedure[118–120], in which we set the significance level $\alpha = 0.05$ and number of bootstraps to 100000. Following[121], we computed the effect sizes for the Wilcoxon signed-rank tests as $\frac{Z}{\sqrt{2 \cdot N}}$, where $N$ is the number of participants. To compute the effect sizes for the two-way repeated-measures ANOVA analyses, we used the formula $\eta^2 = \frac{SS_{effect}}{SS_{total}}$, where SS is the sum of squares.

To visualize the psychometric, confidence, reaction-time and pupil curves as a function of test stimulus speed, we divided the values into 11 bins (with 11 data points each). Data was visualized as mean and standard error of the means (usually across participants) or as median and 95% bootstrapped confidence intervals where non-parametric tests were used. To calculate the 95% bootstrapped confidence

intervals for parameter values, we took samples of the same size as the data with replacement across 5000 iterations using Matlab's `randsample` and calculated the sample median for each iteration. The 95% confidence intervals were based on the 2.5th and 97.5th percentiles of the distribution of medians across iterations.

**Generalized linear mixed-effect models (GLMEs).** We used GLMEs to quantify the relationships between reaction times and either absolute stimulus strength $|s|$ or absolute perceptual-decision certainty $|d|$ (Fig. 5D), as well as the relationship between pupil area and these independent variables (Fig. 6C, D). The main independent variables ($|s|$ or $|d|$ depending on the GLME) were ranked. All GLMEs (implemented using Matlab's `fitlme`) included fixed effects for the intercept, for the independent variables of interest ($|s|$ or $|d|$ depending on the GLME), and for condition (coded as 3 dummy variables for each condition with No-Adapt-See as the reference condition), as well as random intercepts and slopes for all variables. The following GLMEs were used (formulas in Wilkinson notation):

$$\text{Reaction time} \propto 1 + |s| + \text{Condition} + (1 + |s| + \text{Condition}|\text{Participant}) \quad (3)$$

$$\text{Reaction time} \propto 1 + |d| + \text{Condition} + (1 + |d| + \text{Condition}|\text{Participant}) \quad (4)$$

$$\text{Pupil area} \propto 1 + |s| + \text{Condition} + (1 + |s| + \text{Condition}|\text{Participant}) \quad (5)$$

$$\text{Pupil area} \propto 1 + |d| + \text{Condition} + (1 + |d| + \text{Condition}|\text{Participant}) \quad (6)$$

$$\text{Pupil area} \propto 1 + \text{RT} + |d| + \text{Condition} + (1 + \text{RT} + |d| + \text{Condition}|\text{Participant}) \quad (7)$$

For the GLMEs with pupil area as the dependent variable, we computed the average of the normalized pupil area within a 500 ms time window of interest (respectively pink and yellow regions in Fig. 6A). To check that these results were robust to the choice of window and for intepretability, similar to previous work[122], we used a sliding window of 200 ms which we applied in steps of 20 ms across the entire pupil area time series (Fig. 6C, D).

## Eye tracking and pupillometry

In Experiment 2, we monitored and recorded eye position and pupil size using an infrared video-oculographic system (Eyelink 1000 Plus; SR Research, Ltd, Mississauga, Ontario, Canada) with 1000 Hz sampling rate. We recorded monocular data. Stimulus presentation and response collection were controlled by a Mac computer running Matlab 7.1 (MathWorks, Massachusetts, USA) with Psychtoolbox 3[100–102] and EyeLink software[123]. Participants used a chin rest to stabilize their head (Fig. 2B). The chin rest which was located at ~55 cm from the top knob of the desktop mount camera and the distance from the eye to the center of the screen was 66 cm. The eye tracker was calibrated using the 9-point calibration routine before every block[124–126].

**Fixation monitoring.** Fixation was strictly imposed within each trial during a period encompassing the fixation, adaptor and stimulus periods. If the participant's gaze deviated outside of a circle of radius 3 degree of visual angle (dva) centered at fixation, or if they blinked, the trial with the exact same stimulus value was restarted (see Trial Structure above for exact details). This approach ensured that participants fixated during the adaptor and stimulus presentation and thus maximized adaptation effects. Additionally, if participants responded earlier than 3000 ms after stimulus onset, blinks post response still lead to a restart of the trial. Thus, participants avoided blinking for about 8000 ms in each trial. We instructed participants to blink as much as needed and rest their eyes sufficiently after each trial, and that resuming fixation afterwards would initiate the next trial.

**Pupillometry data analysis.** For each participant and block, we defined the relevant task periods within a trial based on the Eyelink timestamps and concatenated the pupil area time series across trials. Mean time-series per condition are shown in Fig. 6A, left. Artifact removal was conducted based on[124]. Segments of the time series within a trial where pupil dilation speeds exceeded a threshold were removed. This threshold was determined based on threshold = median(dilation speeds) + n · MAD(dilation speeds), where MAD is the median absolute deviation and n is a multiplicative factor which we set to 16. In keeping with recommendations[124], we chose this value since our empirical checks showed that it was sufficient to eliminate visible outliers in our data (note that our strict control for blinks yielded relatively clean time series). Each removed segment exceeding this threshold was replaced with interpolated values based on 20 ms before and 20 ms after the segment. Next, the pupil area timeseries were filtered on each trial using a Butterworth filter of order 2 and bandpassed between 0.01 and 10 Hz, implemented in two steps as in ref. 33 with Matlab's functions `butter` and then `filtfilt`. For normalization, based on previous work[122,125,126], we first subtracted the average baseline period—specifically the mean pupil dilation in the last 400 ms of the fixation period—from the pupil area time series for each trial. We concatenated all baseline-subtracted time-series across trials and blocks corresponding to each condition. For each participant, we then calculated the maximum value across all conditions and divided the time series for each condition by that value.

Our main time windows of interest lasted 500 ms (consistent with previous work[48,122]) and started either at 2000 ms post-stimulus or at 500 ms post-response to capture the delayed peak in pupil responses (~1000 ms) while ensuring pupil stabilization following luminance transitions (e.g., between the brighter stimulus screen and the darker gray response screen).

For the moving window analyses in Fig. 6C and D, we applied the GLMEs across sliding windows of 200 ms each, shifted in steps of 20 ms. Note that in the response-locked data from Fig. 6C and D, we do not show pre-response t-statistics as they are not interpretable. This is because differences in reaction times create artifactual differences in pupil dilation due to the latency of pupil stabilization following test stimulus offset (Fig. 6A).

## Psychometric curves

**Psychometric curves and parameters.** We fit psychometric curves based on Gaussian cumulative density functions (cdfs) to the observers' left/right responses as a function of the test stimulus speed (Fig. 3A). $s$ denotes the test stimulus value on a given trial (in arbitrary units of speed, with negative sign indicating counterclockwise motion and positive sign indicating clockwise motion; conversions to standard units of velocity are provided above). Test stimuli ranged between [-2, 2] in Experiment 1 and [-0.3, 0.3] in Experiment 2. We assumed the Gaussian cdf psychometric curve formula[127], in which a response $r$ depends on a stimulus $s$ and the parameters—$\mu, \sigma, \lambda$—as follows:

$$p(r=1|s;\mu,\sigma,\lambda) = \frac{1}{2} \cdot \lambda + (1-\lambda) \cdot \Phi(s;\mu,\sigma), \quad (8)$$

where $r = 1$ stands for a "clockwise" response. The parameters are the bias or point of subjective equality (PSE) denoted with $\mu$, the noise or inverse slope parameter denoted $\sigma$—both of which are inputs to the Gaussian cumulative density function ($\Phi$)—and the lapse rate, $\lambda$.

We first fitted a psychometric curve with 3 parameters for each one of the 4 conditions: No-Adapt-See, No-Adapt-Believe, Adapt-See and Adapt-Believe. However, model comparison selected a model with a shared lapse $\lambda$ across the 4 conditions based on AIC and BIC, so our main psychometric-curve cdf model had free parameters $\mu$ and $\sigma$ for each of the 4 conditions, and a single free parameter $\lambda$ shared across conditions (9 free parameters total; Fig. 3B).

**Psychometric curve parameter estimation.** We performed maximum-likelihood estimation of the psychometric curve free parameters (4 $\mu$, 4 $\sigma$, and 1 shared $\lambda$). The likelihood of a parameter combination is the probability of the data given that parameter combination, with the log likelihood denoted as LL. We assumed that trials were independent and summed the log likelihoods across all trials. The LL for all the trials across all conditions is:

$$\text{LL(parameters)} = \log p(\text{data}|\text{parameters})$$

$$= \sum_{\text{No−Adapt−See trials } j} \log p(r_j|s_j; \mu_{\text{No−Adapt−See}}, \sigma_{\text{No−Adapt−See}}, \lambda)$$

$$+ \sum_{\text{No−Adapt−Believe trials } j} \log p(r_j|s_j; \mu_{\text{No−Adapt−Believe}}, \sigma_{\text{No−Adapt−Believe}}, \lambda)$$

$$+ \sum_{\text{Adapt−See trials } j} \log p(r_j|s_j; \mu_{\text{Adapt−See}}, \sigma_{\text{Adapt−See}}, \lambda)$$

$$+ \sum_{\text{Adapt−Believe trials } j} \log p(r_j|s_j; \mu_{\text{Adapt−Believe}}, \sigma_{\text{Adapt−Believe}}, \lambda)$$

where $s_j$ and $r_j$ are the stimulus and the participant's response on the $j$th trial, respectively.

Parameter estimation was performed via grid searches. For Experiment 1, we searched on a multidimensional grid with 101 values as follows: for each value of $\lambda$ linearly spaced from 0 to 0.3, we performed 4 2-dimensional searches to find the corresponding $\mu$ and $\sigma$ that maximized the LL per condition, with $\mu$ linearly spaced from -0.3 to 0.5 and $\sigma$ logarithmically spaced from 0.001 and 0.5. We then found the global maximum LL across all the possible $\lambda$ values and picked the corresponding set of 4 separate values of $\mu$ and $\sigma$. We used the same strategy for Experiment 2, but each parameter grid had 201 values instead. $\mu$ was linearly spaced from -0.2 to 0.2, $\sigma$ was logarithmically spaced from 0.0001 and 0.2, and $\lambda$ was linearly spaced from 0 to 0.3.

**Estimation of bias $\mu$ from confidence curves and reaction-times curves.** We also estimated the bias $\mu$ from confidence curves and reaction-times curves for each individual (Fig. 4). As these curves were not well captured by Gaussians in our data, we estimated the bias $\mu$ as the mean value of the bin (bin center) with the minimum confidence or maximum RT for each individual. If several bins tied on the minimum or maximum values, the bias $\mu$ was calculated as the mean of these bin centers.

**Drift-diffusion modeling**
We implemented the drift diffusion models with the PyDDM package from[62]. Briefly, the differential equation for the decision variable $v$ across time $t$ is:

$$d\nu = \mu(\nu, t, \ldots)dt + \sigma(\nu, t, \ldots)dW \tag{9}$$

Here, $\mu(v, t, \ldots)$ is the instantaneous drift rate, which depends on the decision variable and time (Drift class in PyDDM), $\sigma(v, t, \ldots)$ is the instantaneous noise (Noise class in PyDDM), and $dW$ is the standard Wiener process. Initial conditions (IC class) are drawn from the distribution $v \sim v_0(\ldots)$, with a default Kronecker Delta function at $v = 0$. The diffusion process terminates when the decision variable $v$ exceeds a bound $v(t, \ldots) >= B(t, \ldots)$, with the exact form dependent on the Bound class chosen in PyDDM. As it is common for DDMs, PyDDM uses

the Fokker-Planck partial differential equation to represent the probability density of the decision variable $v$ with respect to $v$ and time $t$[62].

Practically, before fitting the DDM, we removed trials with reaction times larger than 3 s (there were no trials shorter than 100 ms), as recommended to improve fitting. For each condition, the free parameters of our base model were mean drift rate, decision bound ("BoundConstant") and nondecision time ("OverlayNonDecision"). The noise parameter which captures the standard deviation of the drift process was assumed constant ("NoiseConstant") and fixed to 1, as in previous work[62]. Across all DDM variants, we assumed that drift varies linearly with the absolute value of the test stimulus speed (Drift class set to "DriftCoherence" in PyDDM). The DDM variants we used allowed for either starting-point biases (Biased initial conditions, implemented in PyDDM with "ICPointSideBiasRatio") or drift-rate bias ("DriftCoherenceRewBias") or both, separately for each condition. We used the following parameter ranges: mean drift rate: [0, 20]; decision bound [1, 3]; nondecision time [0, 2]; starting-poing bias [-1, 1]; and drift-rate bias [-1, 1]. The fitting method to optimize the parameters was "differential evolution" on a discretized grid with $\Delta v = 0.01$, $\Delta t = 0.0005$ and the loss function minimized by this method was robust negative log-likelihood ("LossRobustLikelihood")[62].

## Computational models
We now specify the standard perceptual decision-making model and use it to build the perceptual-insight model. As in refs. 128,129 and elsewhere, we first specify the generative or encoding models assumptions, then the decision rules, and then the generation of model predictions.

### Standard perceptual decision model
**Generative model.** We present the generative model of the standard perceptual decision model in Fig. 5A, left. We denote by $C$ the category of the spiral motion, which can take the values of $C = 1$ for clockwise or $C = -1$ for counterclockwise. The two categories occur about half the time each such that the correct prior would be: $p(C = 1) = p(C = -1) = \frac{1}{2}$. The stimulus $s$ is the speed of motion of the test stimulus and takes a value in the interval [-0.3, 0.3], uniformly distributed: $s \sim \mathcal{U}(s; -0.3, 0.3)$ (similar to a case in ref. 130). The generative model further assumes that the observer's brain encodes a noisy measurement $x$ of the stimulus $s$, which here we assume to be corrupted by Gaussian noise with standard deviation $\sigma$.

$$p(x|s) = \mathcal{N}(x; s, \sigma) \tag{10}$$

**Generative model with distortion.** In the generative model with distortion (Fig. 5A, center), we assume the internal representation $x$ is also offset by the factor A, with some value which we also call $A$. This value would plausibly be very close to 0 in the No-Adapt conditions. In the Adapt conditions, we assume A is drawn from $N(A; \mu_A, \sigma_A)$. Thus, $x$ depends on $s$, $A$ and $\sigma$ as follows:

$$p(x|s, A) = \mathcal{N}(x; s - A, \sigma) \tag{11}$$

Note that previous models of adaptation described two main features: the repulsion bias and increased sensitivity around the adaptor[84]. Here, we focused on the repulsion bias, or the offset in $x$ induced by $A$. Increased sensitivity around the adaptor was not relevant to our data as the adaptor was much faster than the test stimuli and thus we could not measure the sensitivity around the adaptor.

**Decision rule.** We assume that the observers compute the optimal decision rule to eventually generate their response for category $\hat{C}$ and confidence $q$ (Fig. 1C). The inference process assumes that the Bayesian observer inverts the generative model to eventually produce their

responses; it is depicted in Fig. 5A, right. As mentioned in the introduction, before producing their responses, we assume that the observer computes the posterior $p(C|x)$:

Using the decision variable, we can write the posterior:

$$p(C=1|x) = \frac{p(C=1|x)}{p(C=1|x)+p(C=-1|x)} = \frac{1}{1+\left(\frac{p(C=-1|x)}{p(C=1|x)}\right)} = \frac{1}{1+e^{-d}} \quad (12)$$

Correspondingly, $p(C=-1|x) = \frac{1}{1+e^d}$.

We assume the observer picks the option $\hat{C}$ with the highest posterior (MAP):

$$\hat{C} = \underset{C}{\mathrm{argmax}}\, p(C|x) \quad (13)$$

This MAP decision rule is equivalent to reporting:

$$\hat{C} = \begin{cases} 1, & \text{if } d > 0 \\ -1, & \text{otherwise} \end{cases} \quad (14)$$

$$\hat{C} = \begin{cases} 1, & \text{if } d > k_{\text{choice}} \\ -1, & \text{otherwise} \end{cases} \quad (15)$$

Instead of comparing $d$ to 0, the observer may use a different criterion or threshold $k_{\text{choice}}$ (Fig. 1C, center). While we do not delve into this here, the observer might perform the computation of the decision variable $d$ in a way affected by decision noise, as assumed by ref. 23 and others (originally[131]).

The observer also reports the confidence associated with their choice. We assume that the observer computes and reports confidence based on the Bayesian confidence hypothesis[22,23], where confidence $q$ depends on the perceptual-decision variable and represents the posterior probability of the observer's choice, an assumption in line with substantial empirical evidence and theoretical work[21,132–134]. As the reported confidence $q$ was binary (low or high, coded respectively as 0 or 1), we assume that observers's confidence response are determined via a confidence threshold $k_{\text{confidence}}$, as:

$$\text{confidence} = p(C=\hat{C}|x) = \frac{1}{1+e^{|d|}}$$
$$q = \begin{cases} 1, & \text{if confidence} > k_{\text{confidence}} \\ 0, & \text{otherwise} \end{cases} \quad (16)$$

Alternative models of confidence have been proposed in the literature[55–58]. These include models with similar architecture to our basic model but which account for additional sources of noise and uncertainty (e.g., variability in the confidence criterion $k_{\text{confidence}}$[135–138] or uncertainty about the confidence variable[139]). We opted for the basic, more established, architecture described above because it was more parsimonious yet appropriate for our design and since it provided a good description of our data. That said, we also considered a model where a $k_{\text{choice}}$ bias could simultaneously shift choice and confidence curves (see $k_{\text{choice and confidence}}$ variant below) to account for alternative model architectures for confidence generation.

**Model predictions.** While the decision rule prescribes how to jointly compute observer's responses $\hat{C}$ and subsequently confidence responses $q$ when their internal measurement $x$ is known, we emphasize that on every trial, the observers' internal measurement $x$ is not known, but the test stimulus $s$ is known. Thus, to find the predicted responses we need to compute the integral $p(\hat{C},q|s)$. To do this, we have to marginalize over all the possible measurements $x$:

$$p(\hat{C},q|s) = \int p(\hat{C},q|x) p(x|s) dx \quad (17)$$

We computed the probability of responses $p(\hat{C},q|s)$ by estimating the above integral by sampling: we simulated several samples (here $N_{\text{samples}} = 500$) of $x$ from $s$ and averaged the predictions over the corresponding outcomes $p(\hat{C},q|x)$.

**Perceptual-insight model**
**Generative model with distortion.** The generative model is the exact same model as above in "Generative model with distortion".

**Decision rule.** In the perceptual-insight model, the observer takes into account the correct generative model with the distortion factor A and then the computation of their posterior $p(C|x)$ is as follows:

$$p(C|x) \propto p(x|C)\, p(C) = p(C) \int p(x|C,A)\, p(A)\, dA$$
$$= p(C) \iint p(x|s,A)\, p(s|C)\, p(A)\, ds\, dA \quad (18)$$

Below, we unpack the decision variable (which consists of the log-posterior ratio as above) using the distributional assumptions spelled out in the generative model with distortion:

$$
\begin{aligned}
d &= \log \frac{p(C=1|x)}{p(C=-1|x)} \\
&= \log \frac{p(C=1) \iint p(x|s,A)p(A)p(s|C=1)\,ds\,dA}{p(C=-1) \iint p(x|s,A)p(A)p(s|C=-1)\,ds\,dA} \\
&= \log \frac{p(C=1)}{p(C=-1)} \frac{\iint \mathcal{N}(x;s-A,\sigma)\mathcal{N}(A;\mu_A,\sigma_A)\mathcal{U}(s;s_{11},s_{12})\,ds\,dA}{\iint \mathcal{N}(x;s-A,\sigma)\mathcal{N}(A;\mu_A,\sigma_A)\mathcal{U}(s;s_{21},s_{22})\,ds\,dA} \\
&= \log \frac{p(C=1)}{p(C=-1)} \frac{\frac{1}{s_{12}-s_{11}}\iint \mathcal{N}(x;s-A,\sigma)\mathcal{N}(A;\mu_A,\sigma_A)\,ds\,dA}{\frac{1}{s_{22}-s_{21}}\iint \mathcal{N}(x;s-A,\sigma)\mathcal{N}(A;\mu_A,\sigma_A)\,ds\,dA}
\end{aligned} \quad (19)
$$

$$
\begin{aligned}
&= \log \frac{p(C=1)}{p(C=-1)} \frac{\frac{1}{s_{12}-s_{11}}\int_{-\infty}^{\infty}\int_{s=s_{11}}^{s=s_{12}} \frac{1}{\sigma\sqrt{2\pi}}e^{-\frac{(x-(s-A))^2}{2\sigma^2}}\frac{1}{\sigma_A\sqrt{2\pi}}e^{-\frac{(A-\mu_A)^2}{2\sigma_A^2}}\,ds\,dA}{\frac{1}{s_{22}-s_{21}}\int_{-\infty}^{\infty}\int_{s=s_{21}}^{s=s_{22}} \frac{1}{\sigma\sqrt{2\pi}}e^{-\frac{(x-(s-A))^2}{2\sigma^2}}\frac{1}{\sigma_A\sqrt{2\pi}}e^{-\frac{(A-\mu_A)^2}{2\sigma_A^2}}\,ds\,dA} \\
&= \log \frac{p(C=1)}{p(C=-1)} \frac{\frac{1}{s_{12}-s_{11}}\frac{1}{\sigma\sqrt{2\pi}}\frac{1}{\sigma_A\sqrt{2\pi}}\int_{-\infty}^{\infty}\int_{s=s_{11}}^{s=s_{12}} e^{-\frac{(x-(s-A))^2}{2\sigma^2}}e^{-\frac{(A-\mu_A)^2}{2\sigma_A^2}}\,ds\,dA}{\frac{1}{s_{22}-s_{21}}\frac{1}{\sigma\sqrt{2\pi}}\frac{1}{\sigma_A\sqrt{2\pi}}\int_{-\infty}^{\infty}\int_{s=s_{21}}^{s=s_{22}} e^{-\frac{(x-(s-A))^2}{2\sigma^2}}e^{-\frac{(A-\mu_A)^2}{2\sigma_A^2}}\,ds\,dA} \\
&= \log \frac{p(C=1)}{p(C=-1)} \frac{\frac{1}{s_{12}-s_{11}}\int_{-\infty}^{\infty}\int_{s=s_{11}}^{s=s_{12}} e^{-\frac{(s-(A+x))^2}{2\sigma^2}}e^{-\frac{(A-\mu_A)^2}{2\sigma_A^2}}\,ds\,dA}{\frac{1}{s_{22}-s_{21}}\int_{-\infty}^{\infty}\int_{s=s_{21}}^{s=s_{22}} e^{-\frac{(s-(A+x))^2}{2\sigma^2}}e^{-\frac{(A-\mu_A)^2}{2\sigma_A^2}}\,ds\,dA}
\end{aligned} \quad (20)
$$

This integral can be solved numerically. However, for the models we test (Table 1), to make the calculations more tractable, we make the assumption that the observer has negligible uncertainty about their offset A, such that they have essentially no uncertainty over it ($\sigma_A$ asymptotically close to 0). Nonetheless, the model had flexibility in this respect as it allowed $\sigma_{\text{encoding}}$ parameters to vary in each condition, which could—and seemingly did—absorb additional uncertainty likely associated with A in the Adapt-Believe condition. Simulations of the full model allowing a $\sigma_A$ different from zero showed negligible effects on measured values of the $\mu_{\text{likelihood}}$ and $\sigma_{\text{encoding}}$ (with the latter absorbing non-zero $\sigma_A$ values at low levels of encoding noise), supporting the validity of our simplified model (see Fig. S8).

Normalized Gaussian functions with the width tending to 0, $\lim_{w\to 0}$, will yield Dirac delta functions $\delta$:

$$\delta_w(x) = \lim_{w\to 0}\frac{1}{|w|\sqrt{\pi}}e^{-\left(\frac{x}{w}\right)^2} \quad (21)$$

$$\delta(x) = \begin{cases} \infty, & \text{if } x = 0 \\ 0, & \text{otherwise} \end{cases} \quad (22)$$

The Delta function has the following property for any continuous function $f(x)$:

$$\int_{-\infty}^{\infty} f(x)\delta(x-x_0)dx = f(x_0) \tag{23}$$

Assuming that in $p(A)\sigma_A$ is asymptotically close to 0, we will get:

$$N(A;\mu_A,\sigma_A) \propto \frac{1}{|\sigma_A|\sqrt{\pi}} e^{-\left(\frac{A-\mu_A}{\sigma_A^2}\right)} = \delta_{\sigma_A}(A-\mu_A) \tag{24}$$

Substituting the relevant terms into Eq. (19) and using the property of the $\delta$ function we get:

$$\iint \mathcal{N}(x;s-A,\sigma)\mathcal{N}(A;\mu_A,\sigma_A)ds\,dA = \iint \mathcal{N}(x;s-A,\sigma)\delta_{\sigma_A}(A-\mu_A)ds\,dA$$
$$= \int \mathcal{N}(x;s-\mu_A,\sigma)ds \tag{25}$$

Therefore:

$$d = \log\frac{p(C=1)}{p(C=-1)}\frac{\int_{s_{11}}^{s_{12}} \mathcal{N}(x;s-\mu_A,\sigma)\mathcal{U}(s;s_{11},s_{12})ds}{\int_{s_{11}}^{s_{12}} \mathcal{N}(x;s-\mu_A,\sigma)\mathcal{U}(s;s_{21},s_{22})ds}$$
$$= \log\frac{p(C=1)}{p(C=-1)}\frac{\frac{1}{s_{12}-s_{11}}\int_{s_{11}}^{s_{12}} \mathcal{N}(x;s-\mu_A,\sigma)ds}{\frac{1}{s_{22}-s_{21}}\int_{s_{21}}^{s_{22}} \mathcal{N}(x;s-\mu_A,\sigma)ds} \tag{26}$$

For the set of test stimuli in Experiment 2, we have: $s_{11}=0$, $s_{12}=0.3$ and $s_{21}=-0.3$, $s_{22}=0$. We make the change of variable $t=s-\mu_A$, with $dt=ds$, we substitute the limits of integration accordingly and write:

$$d = \log\frac{p(C=1)}{p(C=-1)}\frac{\int_{s_{11}-\mu_A}^{s_{12}-\mu_A} \mathcal{N}(x;t,\sigma)dt}{\int_{s_{21}-\mu_A}^{s_{22}-\mu_A} \mathcal{N}(x;t,\sigma)dt}$$
$$= \log\frac{p(C=1)}{p(C=-1)}\frac{\int_{s_{11}-\mu_A}^{s_{12}-\mu_A} \frac{1}{\sigma\sqrt{2\pi}}e^{-\frac{(x-t)^2}{2\sigma^2}}dt}{\int_{s_{21}-\mu_A}^{s_{22}-\mu_A} \frac{1}{\sigma\sqrt{2\pi}}e^{-\frac{(x-t)^2}{2\sigma^2}}dt}$$
$$= \log\frac{p(C=1)}{p(C=-1)}\frac{\int_{s_{11}-\mu_A}^{s_{12}-\mu_A} \frac{1}{\sigma\sqrt{2\pi}}e^{-\frac{(t-x)^2}{2\sigma^2}}dt}{\int_{s_{21}-\mu_A}^{s_{22}-\mu_A} \frac{1}{\sigma\sqrt{2\pi}}e^{-\frac{(t-x)^2}{2\sigma^2}}dt}$$
$$= \log\frac{p(C=1)}{p(C=-1)}\frac{\int_{s_{11}-\mu_A}^{s_{12}-\mu_A} \mathcal{N}(t;x,\sigma)dt}{\int_{s_{21}-\mu_A}^{s_{22}-\mu_A} \mathcal{N}(t;x,\sigma)dt} \tag{27}$$

We use the property that : $Pr[La \leq x \leq Lb] = \int_{La}^{Lb} N(x;\mu,\sigma)dx = \frac{1}{2}\cdot\left[\operatorname{erf}\left(\frac{Lb-\mu}{\sqrt{2}\sigma}\right) - \operatorname{erf}\left(\frac{La-\mu}{\sqrt{2}\sigma}\right)\right]$

$$d = \log\frac{p(C=1)}{p(C=-1)}\frac{\frac{1}{2}\left[\operatorname{erf}\left(\frac{s_{12}-\mu_A-x}{\sqrt{2}\sigma}\right) - \operatorname{erf}\left(\frac{s_{11}-\mu_A-x}{\sqrt{2}\sigma}\right)\right]}{\frac{1}{2}\left[\operatorname{erf}\left(\frac{s_{22}-\mu_A-x}{\sqrt{2}\sigma}\right) - \operatorname{erf}\left(\frac{s_{21}-\mu_A-x}{\sqrt{2}\sigma}\right)\right]} \tag{28}$$

Because $\Phi(x;s,\sigma) = \frac{1}{2}\left[1+\operatorname{erf}\left(\frac{x-s}{\sqrt{2}\sigma}\right)\right]$ and thus $\operatorname{erf}\left(\frac{x-s}{\sqrt{2}\sigma}\right) = 2\Phi(x;s,\sigma)-1$ we write:

$$d = \log\frac{p(C=1)}{p(C=-1)}\frac{[2*\Phi(s_{12}-\mu_A-x;0,\sigma)-1-2*\Phi(s_{11}-\mu_A-x;0,\sigma)+1]}{[2*\Phi(s_{22}-\mu_A-x;0,\sigma)-1-2*\Phi(s_{21}-\mu_A-x;0,\sigma)+1]}$$
$$= \log\frac{p(C=1)}{p(C=-1)}\frac{[\Phi(s_{12}-\mu_A-x;0,\sigma)-\Phi(s_{11}-\mu_A-x;0,\sigma)]}{[\Phi(s_{22}-\mu_A-x;0,\sigma)-\Phi(s_{21}-\mu_A-x;0,\sigma)]} \tag{29}$$

We plug in the values $s_{11}=0$, $s_{12}=0.3$ and $s_{21}=-0.3$, $s_{22}=0$, take the log and write out the decision variable:

$$d = \log\frac{p(C=1)}{1-p(C=1)} + \log\left(\frac{\Phi(0.3-\mu_A-x;0,\sigma)-\Phi(-\mu_A-x;0,\sigma)}{\Phi(-\mu_A-x;0,\sigma)-\Phi(-0.3-\mu_A-x;0,\sigma)}\right) \tag{30}$$

An insightful observer will know that they have an offset $\mu_A \neq 0$ and incorporate this into their decision variable. To distinguish between $\mu$ from the generative model and this value used in inference, we refer to them respectively as $\mu_{\text{encoding}}$ and $\mu_{\text{likelihood}}$. In the case of the Adapt conditions, $\mu_{\text{encoding}}$ is offset. In Adapt-See, observers may ignore this offset and thus use $\mu_{\text{likelihood}} \approx 0$. In Adapt-Believe, instead, observers are asked to incorporate their knowledge of the offset and will thus try to use a $\mu_{\text{likelihood}}$ approximating their estimate of $\mu_{\text{encoding}}$. Inference can be insightful even if $\mu_{\text{likelihood}} \neq \mu_{\text{encoding}}$, but the closer these values are, the more accurate the compensation will be for perceptual distortions.

Once the decision variable is computed as shown here, its use in the decision rule to yield choice $\hat{C}$ and confidence $q$ is assumed to be as described above by the MAP decision rule in Equation (15).

**Model predictions.** "Model predictions" are again as in the previous section.

## Bayesian model variants
We tested the model variants presented in Table 1. All models had 6 parameters for the early sensory-encoding stage (2 fixed for $\mu_{\text{encoding}}$, 4 free $\sigma$ parameters) and to allow for MAE compensation either had free parameters for intermediate-stage processes ($\mu_{\text{likelihood}}$ and/or category prior), or late-stage processes ($k_{\text{choice}}$), or both. All models also had free parameters for the confidence threshold $k_{\text{confidence}}$. The perceptual-insight model used for fitting only had free parameters for $\mu_{\text{likelihood}}$ at the intermediate inference stage. The critical competing model was able to compensate for the illusion based solely on changes in $k_{\text{choice}}$ at a late choice stage. Other hybrid models were evaluated for completeness. Model comparison based on goodness of fit is presented in Fig. 5B.

## Model fitting
We fitted the Bayesian model variants jointly to the choice and confidence data separately for each condition. Across conditions, we used the model fitting strategy described below. We fixed the two $\mu_{\text{encoding}}$ parameters to the $\mu$ values fitted from the psychometric curves for the No-Adapt-See and Adapt-See conditions. We performed maximum-likelihood estimation (MLE) of the remaining parameters in the Bayesian models (with parameters as in Table 1). For a particular model, the likelihood of a set of parameters $\theta$ is the probability of the data given those parameters, $p(\text{data}|\theta)$. We denote the log likelihood with LL. We assumed that trials are independent of each other and thus we could sum the log likelihoods across all trials:

$$LL(\theta) = \log p(\text{data}|\theta) = \log\left(\prod_{j=1}^{N_{\text{trials}}} p(\text{responses}_j|s_j,\theta)\right)$$
$$= \sum_{j=1}^{N_{\text{trials}}} \log p(\text{responses}_j|s_j,\theta)$$

We denote the subject's responses on the $j^{\text{th}}$ trial with responses$_j$ above, which could be $\hat{C}_j=1$ for clockwise or respectively $\hat{C}_j=0$ for counterclockwise, and $q=1$ for high confidence or $q=0$ for low confidence.

We replaced extreme values (0 or 1) of $p(\hat{C},q|s)$ with $\frac{1}{N_{\text{samples}}}$ for 0 and with $1-\frac{1}{N_{\text{samples}}}$ for 1.

As in the "Model predictions" section, we approximated the $p(\text{responses}_j|s_j, \theta)$ through sampling. Even with 500 samples, the log likelihood can be considered noisy. To find the parameters $\theta$ that maximize $LL(\theta)$ we used an optimization method called Bayesian adaptive direct search (BADS)[140] that is especially suited for noisy functions. For each dataset and model, we ran BADS with 14 starting points and chose the best fitting parameters among those. Within BADS, we set the estimated noise size to 1 (`options.NoiseSize=1`).

The parameter ranges were $[\log(0.001), \log(0.15)]$ for $\sigma$, $[0.0001, 0.0099]$ for the category prior right, $[0.50000, 0.99999]$ for $k_{\text{confidence}}$, $[-10, 10]$ for $k_{\text{choice}}$ and $[-3, 6.5]$ for $\mu_{\text{likelihood}}$ For these parameters, we set the hard bounds equal to these plausible bounds, except: $[0.4, 0.6]$ for category prior right and $[-5, 5]$ for $k_{\text{choice}}$. None of the participants had the parameters fitted to their upper or lower bounds, suggesting that these ranges were meaningful.

**Model comparison.** We performed model comparison based on the Akaike Information Criterion (AIC)[141] and the Bayesian Information criterion (BIC)[142]. These metrics are defined as $\text{AIC} = -2LL^* + 2n_{\text{parameters}}$ and $\text{BIC} = -2LL^* + n_{\text{parameters}} \log n_{\text{trials}}$, respectively, where $LL^*$ is the maximum log likelihood and $n_{\text{parameters}}$ and $n_{\text{trials}}$ the number of free parameters and the number of trials, respectively.

To statistically determine the winning model that best captured the data, we substracted the AIC and BIC values of the reference model from the AIC and BIC values of every other model and summed these differences across participants as in ref. 143; to get 95% bootstrapped confidence intervals for these sums, we took 1000000 samples of 22 participants with replacement.

### Reporting summary
Further information on research design is available in the Nature Portfolio Reporting Summary linked to this article.

## Data availability
The processed behavioral data generated in this study has been deposited in a public github repository https://github.com/lianaan/Insight, with the following corresponding Zenodo link https://zenodo.org/record/8411332[144]. The processed eye tracking data is available at the links accessible from the github repository. The raw behavioral and eye tracking data are available on OSF at https://doi.org/10.17605/OSF.IO/DSYHC.

## Code availability
Analysis and modeling code have been deposited on GitHub at https://github.com/lianaan/Insight, under the following Zenodo https://zenodo.org/record/8411332[144]. Experimental code is available upon request to the corresponding authors.

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

## Acknowledgements

This work was supported by the National Institute of Mental Health, specifically awards R01MH117323 (G.H.) and R01MH114965 (G.H.). We thank Kyo Iigaya, Mike Shadlen, Hakwan Lau, Daniel Wolpert, Amy Rapp, Ramon Nogueira and John Morrison for useful discussions and feedback on this project. We thank Ken Wengler, Brandon Ashinoff, Seth Baker, Lisa Clark and Isabella Rosario for discussions and comments on this manuscript. We thank Alissa Fogelson, Anastasia Velikovskaya, Garrett Salzman and Jocelyn Kim for assistance with data collection. We thank Carmen Mendes de Leon for help with control analyses and Prady Sepulveda and Ahad Butt for their help with Python and discussions on the DDM. We also thank the participants for their time and work.

## Author contributions

G.H. and A.M. conceived the original ideas and designed the task. A.M., M.B., F.D.M.R. performed the experimental studies and analyzed data and contributed to computational modeling and interpretation. A.M. carried out the final data analyses, implemented computational modeling, and wrote the first manuscript draft. A.M. and G.H. discussed and interpreted the results and wrote the final version of the manuscript. G.H. acquired funding.

## Competing interests

The authors declare no competing interests.
