## [Peer Review File · Nature Communications]

Editorial Note: This manuscript has been previously reviewed at another journal outside of the Nature Portfolio. This document only contains reviewer comments and rebuttal letters for versions considered at *Nature Communications*. Mentions of the other journal have been redacted.

Reviewers' Comments:

Reviewer #1:

Remarks to the Author:

This study investigates how people can use knowledge about how their experience can deviate from reality to determine what is real – akin to how knowledge about their disease might help people with psychosis better navigate their altered reality. Using a novel psychophysics experiment building on the motion after effect (MAE) in combination with computational Bayesian modeling, the results show that perceptual insight building is achieved through incorporating knowledge about the MAE at an intermediate stage of the perceptual-decision making process. These findings substantially increase our knowledge about the cognitive mechanisms supporting perceptual insight and have important implications for our understanding of reality testing disorders such as psychosis. [Redacted]

Several methodological details that were previously missing have now been added. The conceptual interpretation of the prior model and how this differs from the likelihood model was made clearer but the cognitive interpretations of these results are still not entirely clear to me. I.e. can we draw any interesting conclusions from the fact the likelihood model is a better fit than the prior model? What does it mean psychologically that a model in which the compensatory shifts in d scale with encoding noise is the better explanation for the data?

However, unfortunately, the majority of my previous comments were not addressed, listed below:

Major comments

- I find the conclusion that participants are able to 'compensate' for the illusion confusing because they clearly over-compensate, meaning that the resulting response is still not veridical, and therefore not adaptive. It would be more accurate to say that the knowledge about illusion 'can be taken into account to change the decision' or 'influences the decision'.
- Relatedly, because there is no baseline condition in which no training about the illusion was given to participants, the authors cannot be certain that the adapt-believe condition reflects insight gained by training and not just some other general compensatory strategy. I do not expect the authors to run additional experiments to investigate this, but it would be good to discuss this potential limitation in the paper.
- Furthermore, there needs to be some more discussion/explanation on potential order-effects. For example, Adapt-Believe blocks were always done after Adapt-See blocks, which means that slower RTs for Believe conditions might just be due to fatigue. It is argued that Adapt-Believe has to be second because it's harder and thus benefits from the additional task exposure, but you can similarly argue that for this reason, it should be presented earlier so that it is less influenced by fatigue.
- Was illusion reproduction (i.e. illusions strength) correlated with the compensatory bias so that participants who experienced a stronger illusion also compensated for it more?
- Considering that the perceptual-insight model in which the compensation happened at an intermediate, perceptual inference level was the only model that predicted an in-tandem shift in the confidence curves and that such a shift was indeed observed, what exactly do the model comparison and the pupillometry add to this (besides the control that participants are really looking at the adaptor)? As the manuscript is written now, it seems like both analyses are merely another way of showing the same thing. Can the pupil data and the model fits discount any potential alternative explanations that cannot be ruled out based on the confidence data alone?
- Fig. 5A in the text seems to refer to the graphical models but Fig. 5A only presents the model comparison results, not the model architectures. This makes it a bit difficult to follow this section.
- Furthermore, after rereading, I now find the discussion section a bit unclear. Specifically, this section starts by stating that compensation of illusionary perception happens at the 'inferential' level, but for someone who has not read the entire paper, this is quite vague. Please reiterate in a sentence or two what that means and why it is interesting. Finally, it is stated that the current study provides evidence that beliefs about reality arise from higher-level processes, but how? Don't the results show that these are implemented at a perceptual-decision level, so at an intermediate stage? More explanation about this is necessary as well as more discussion about how these results fit with other theories of reality testing.

Minor comments:

- In experiment 2, the overcompensation in the Adapt-Believe condition seems even stronger than the illusion effect in the Adapt-See condition compared to control conditions – is this statistically significant and do the authors have an explanation for this?
- “Despite the broad societal implications of reality testing, the cognitive mechanisms underlying this ability are almost completely unknown” – I find this a very strong statement, especially since there is quite some recent work by e.g. Phil Corlett into the cognitive mechanisms underlying insight. Please discuss how the current study relates to that existing work.
- Please mention briefly in the text that the colours of the fixation crosses were changed in experiment 2 to keep external input the same between conditions for eye-tracking.
- Please add some more details on why the assumption that the offset A has negligible uncertainty in the perceptual insight model is an appropriate assumption to make and how it might influence results.
- The second sentence of the discussion is grammatically incorrect.
- The discussion reads ‘model comparison and ‘validation’ – what is meant by ‘validation’ in this context? Relatedly, what does ‘supporting our conceptualization of perceptual insight’ mean?

Reviewer #2:

Remarks to the Author:

Overall impression: This is an interesting paper with careful measurement and modeling on a difficult topic. Some of the framing could be clearer and there might be alternative explanations for the observed phenomena.

Summary: This paper asks whether and how people can discount their sensory responses to an illusion when they know they are experiencing an illusion. Specifically, subjects are given some brief instructions and demonstrations of how motion aftereffects work, and then they make judgments about the direction of motion of test stimuli while experiencing motion aftereffects. They also make confidence judgments in each trial. In one condition, similar to the usual adaptation experiments, subjects are asked to report what they see. This condition produces the expected features of an adaptation experiment: adaptation to a counterclockwise rotating spiral causes a shift in the psychometric function toward the adaptor, and reaction time is slowest (and confidence lowest) at the point of subjective equality rather than at the physically neutral point. In contrast, when subjects are instead asked to report what they *believe* the stimulus is doing (rather than what it looks like), they undo the shift in the psychometric function (and even overcompensate a bit). The most important result for the authors is that in this “believe” condition, uncertainty, as reflected in slow response times and low confidence, shifts with the choice (i.e., subjects are slowest and least confident near where the choice curves are at 50/50). The interpretation, based both on the qualitative pattern of results and on formal model fitting and model comparison, is that subjects incorporate their knowledge of the distortion at an intermediate stage of processing, that is later than sensory encoding but earlier than a response bias. The authors describe the effect as perceptual insight that shifts the perceptual decision variable.

Issues / questions

1. To be frank, I could not make much sense of the exposition about “perceptual insight”. I do not know what the authors mean by this term, perhaps because I am not an expert in the metacognition literature (though I am familiar with sensory adaptation). I could not really figure it out from the manuscript, and I would recommend making the meaning of this clearer, since it is the central point of the paper. That said, the models are concrete and specific, and the differences in predictions between a late effect (shift in choice curve but not in confidence/RT curves) vs an intermediate effect (shift in both) are clear enough. The data are also reasonably clear (all functions shift, not just choice). Hence the difficulty in understanding the terminology is only a modest concern.

2. Is it possible that when the task changes (report what you believe rather than what you see), subjects do not use metacognition (i.e., do not use Bayesian inference that takes into account knowledge of the distortion), but instead simply use different sensory information to inform their

decision? For example, suppose that in the see condition, subjects make judgments based on local changes in position (velocity), which is strongly affected by adaptation, whereas in the believe condition they make judgments based on the change in position relative to a reference, such as the square stimulus aperture or the screen? Put another way, there may be sufficient information in the stimuli to make reasonably accurate judgments even when experiencing a strong aftereffect. If the answer to my question is yes, would this change the interpretation?

3. Because there is only one adapt condition (one direction and one strength), it seems not possible to distinguish whether participants have a quantitative internal model of the distortion vs just knowing that there is a distortion (with a point estimate of its effect). Does this matter to the interpretation? This differs from some other literatures, such as reinforcement learning literature, in which experimentalists test whether participants have an internal model of a task or process or setting.

4. [Minor] In the introduction, a perceptual distortion is described as "a discrepancy between an objective stimulus and its corresponding percept". I believe this description does not make sense, since a stimulus and a percept can never be the same (the stimulus is a physical object, the percept is not). Perhaps it would be better to say, "a discrepancy between a property of a stimulus and an estimate of that property"?

5. [Minor] Figure 3A, top: it looks like the curve in the adapt-believe condition is centered slightly to the right of 0, which should produce a bias greater than 0, yet the bias is a bit less than 0 in the bar plot in panel B. Am I misreading the graph or misunderstanding the model parameters?

6. Last: I don't know what the journal's policy is or the author's intentions are with respect to data sharing, but it might be valuable for the community if the data were made public.

Reviewer #3:

Remarks to the Author:

Mihali et al investigated the capacity for human subjects to adjust their responses to illusory visual stimuli based on their knowledge of the illusory source of the appearance of the stimuli. Using a novel task design that asked observers what they believed the stimulus was in addition to what they perceived the stimulus to be, the authors found that observers (over)compensated their 'belief' responses to adapt to the fact that they had knowledge of the illusory nature of their perceptions. Modeling of choice and confidence data suggest that the compensatory effect took place at the intermediate level of the decision variable representation, rather than at a sensory stage or at a late (post-decision-variable formation) stage. The introduction of a novel task to measure a clinically relevant feature of cognition and behavior (reality monitoring and insight) along with computational modeling of behavioral data and validation with pupillometry make this a methodologically strong paper with conclusions likely interesting to a broad audience. I have a few concerns that, if addressed, could hopefully the conclusions of the paper a bit clearer.

My primary concern is about the use of confidence and RT data to make strong inferences about the effect of insight on decision-variable representations as opposed to on response-level criteria. The authors state that a tandem shift in confidence and choice psychometric functions indicate an intermediate decision-level effect, whereas an isolated shift in choice but not confidence curves would indicate a late-stage effect on choice criterion. However, this predicted result seems to be an artificial byproduct of the fact that the model used does not allow confidence criteria to be updated with knowledge in the same way that choice is allowed to vary (i.e., the influence of k -choice on c -prime in the model). It seems like a reasonable hypothesis that prior knowledge could influence confidence criteria in addition to choice criterion and that this could also lead to tandem shifts in the confidence psychometric functions as a result of a late-stage effect. From an evidence accumulation perspective, where choice, confidence, and RT have been argued to reflect the amount of accumulated evidence at the time a decision is reached, changes to criteria or changes to the decision variable could produce effects on choice, RT, and confidence. Thus, the use of confidence and RT data here do not seem to uniquely point to an intermediate-level effect. Ideally, some more model comparisons and implementation of a drift-diffusion framework to

capture RT effects could be used to bolster the main claim about a decision-level effect.

Related to the point above I don't quite see the theoretical argument for why a late-stage effect would not amount to true insight. If prior knowledge has the effect of changing response criteria such that my behavior is adapted in light of my prior beliefs, why wouldn't this count as a form of insight? Some more discussion of these different mechanisms (e.g., decision versus response) and their implications would be insightful.

Lastly, I think more could be said in the primary manuscript about the nature of the confidence report. Typically, confidence is construed as the probability that a decision was correct and subjects are instructed to rate higher confidence when they felt their choice was more likely to be correct. However, this only applies when the first-order choice is about a stimulus property (as in the 'believe' condition here). In the 'see' condition, participants know their choices are often incorrect but are responded to provide a confidence judgment anyway. My first comment is just for clarification – how were confidence reports construed to the subjects in both of the tasks? My second comment is whether this inherent difference in what confidence is referring to in the two tasks could partly drive differences in confidence between the see and believe conditions.

Nature communications

Dear Reviewers,

With this letter, we are submitting a revised version of our paper (NCOMMS-22-39720A) incorporating several changes aimed at addressing comments from reviewers. We appreciate the constructive feedback. Below we provide point-by-point answers to this feedback. We also enclose the revised version of the paper (with changes in colored text), which we hope is now acceptable for publication. Thank you again for your time and consideration and do not hesitate to ask for any additional information.

Sincerely,
Andra Mihali, PhD, and Guillermo Horga, MD, PhD

REVIEWER COMMENTS

Dec 14

Reviewer #1 (Remarks to the Author):

This study investigates how people can use knowledge about how their experience can deviate from reality to determine what is real – akin to how knowledge about their disease might help people with psychosis better navigate their altered reality. Using a novel psychophysics experiment building on the motion after effect (MAE) in combination with computational Bayesian modeling, the results show that perceptual insight building is achieved through incorporating knowledge about the MAE at an intermediate stage of the perceptual-decision making process. These findings substantially increase our knowledge about the cognitive mechanisms supporting perceptual insight and have important implications for our understanding of reality testing disorders such as psychosis. *[Redacted]*

R1.P1. Several methodological details that were previously missing have now been added. The conceptual interpretation of the prior model and how this differs from the likelihood model was made clearer but the cognitive interpretations of these results are still not entirely clear to me. I.e. can we draw any interesting conclusions from the fact the likelihood model is a better fit than the prior model? What does it mean psychologically that a model in which the compensatory shifts in d scale with encoding noise is the better explanation for the data?

This is a good question and we appreciate the opportunity to elaborate on this point. While both the likelihood (perceptual-insight) model and the category-prior model can compensate for the MAE illusion, they differ in that only the likelihood model does so by incorporating an accurate generative model and prior. The likelihood model is optimal in that it can have accurate knowledge about the internal distortion affecting its internal sensory representation and use the correct 0.5 category prior; in contrast, the category-prior model is never optimal in the sense that – *by design* – it uses an incorrect category prior (e.g., the prior probability of a clockwise rotating spiral the model uses may be 0.15 when the actual probability is 0.5) to compensate for a distortion it has no explicit knowledge of. In other words, the category-prior model can “hack” this problem and compensate for MAE because it has sufficient flexibility, but it does so

suboptimally – that is, with an incorrect prior and an incorrect generative model that ignores the internal distortion and its effect on the sensory measurement. So what does this mean psychologically? Our result that the perceptual-insight model outperforms the category-prior model may, in principle, suggest that people are able to use an optimal strategy that incorporates knowledge about internal distortions, even if the estimation about the level of internal distortion is not fully accurate (which could account for overcompensation, as mentioned below). That is, participants can use a correct generative model, or at least approximate it, even if their parameters are off (versus a completely incorrect generative model as in the category-prior model).

To illustrate this, we have generated simulations comparing the μ likelihood model and the category-prior model under the Adapt conditions, with different levels of compensation (from no compensation, as in Adapt-See, to [over]compensation in Adapt-Believe). Most importantly, we introduce increasing levels of sensory noise (up to high levels of σ encoding) to capture a scenario where the test stimulus is similar or substantially fainter than in the actual task. Intuitively, ideal observers should be able to compensate for the distortion such that they correct for the inferred bias in a noise-robust manner: under a given level of expected distortion, on average they should shift their psychometric and confidence curves to the same extent regardless of the level of sensory noise associated with the test stimulus, i.e., their curves should become shallower but the bias should remain the same. In new simulations (Figure S5), we show that this is the case for the μ likelihood (perceptual-insight) model, but not for the category-prior model. For a given change in the category prior, the category-prior model exhibits shifts in responses that are exacerbated with increasing levels of noise. This means that under a situation like a low-contrast test stimulus, the category-prior model does a poor job of compensating, illustrating its shortcomings as a compensation strategy.

We have now included additional information on this point in the Results (page 12) and a slightly modified version of the paragraph above in Section S3.5 (page 44) and display Figure S5 (page 45) and its caption below.

In the Results (page 12): “Similarly to the perceptual-insight (μ likelihood) model, the category prior model can compensate for the MAE via an intermediate inferential process manifesting as shifts in tandem, but achieves this compensation through a distinct **suboptimal mechanism (using an incorrect category prior and an incorrect generative model)**; while the perceptual-insight model compensates by flexibly modifying the perceptual-variable d in a way that scales with encoding noise (Equation 16 and Supplementary Fig. S4 and **Supplementary Fig. S5**), the prior model simply and **incorrectly** assumes a change in the category prior that produces fixed shifts in d regardless of encoding noise. ”

Figure S5: Simulations of the A) $\mu_{\text{likelihood}}$ model and B) the category-prior model show different scaling with the noise σ_{encoding} . Each line resulted from the choice and confidence responses on 50000 simulated trials generated with the listed parameters, and responses averaged into 11 bins defined based on stimulus strength (just as in the real data). The model-specific parameters used to generate these simulations are listed on top of the subplots and the σ_{encoding} values are color-coded and displayed in the legend; the other parameters are informed by median values in data in the Adapt-Believe condition ($\mu_{\text{encoding}} = -0.055$, $k_{\text{confidence}} = 0.8$).

However, unfortunately, the majority of my previous comments were not addressed, listed below:

We apologize for this. We believe that we may have at least partially addressed some of these points in the last version, although we did not have the opportunity to provide more context and point the reviewer to the changes we had made. That said, we have now made substantial additions attempting to fully address the remaining concerns from the reviewer.

Major comments

R1.P2.- I find the conclusion that participants are able to ‘compensate’ for the illusion confusing because they clearly over-compensate, meaning that the resulting response is still not veridical, and therefore not adaptive. It would be more accurate to say that the knowledge about illusion ‘can be taken into account to change the decision’ or ‘influences the decision’.

We take the reviewer's fair point that we do not have evidence that people can compensate perfectly. We have now toned down our language implying the compensation is well calibrated or optimal, and have included more context on point.

Indeed, people overcompensate in Experiment 2. However, in Experiment 1, appropriate nonparametric statistical tests fail to show any evidence of overcompensation ($z = 1.19$, $p = 0.24$; Fig 3C). While we do not have a definitive explanation for these differences, our take is that overcompensation happens but is not universal (this is also backed by other additional data we have been collecting but is not shown here). One potential explanation relates to the differential training and experimental procedures between Experiments 1 and 2. Experiment 2 had enhanced training with more emphasis and information on MAE strength and longer adaptor presentation at the beginning of each block. In particular, during some parts of the training in Experiment 2 participants received as feedback the value of the test speed spiral (in percentages relative to the maximum shown as tests, with the adaptor being much faster at 3000%). This numerical feedback may have influenced the participants' impressions of illusion strength. In Experiment 2, participants also practiced reports for MAE strength estimates (unlike in Experiment 1), so they were exposed more times to the longer MAE illusion, which could have reinforced their expectation for a stronger MAE. Thus, one possibility is that participants in Experiment 2 expected a stronger MAE and overcompensated as a result.

We also note that, based on choice evidence for MAE compensation or overcompensation alone, we cannot be sure that participants incorporate knowledge about the illusion to influence their decisions (in line with the proposed language by the reviewer). That is, shifts in choice alone need not reflect knowledge of the illusion, which was precisely a motivating reason for our additional analyses of confidence, RT, and pupillometry, as well as for computational model comparison (see below).

That said, in response to this comment, we have made it clearer that participants tended to overcompensate. We have toned down the language and used terminology such as that proposed by the reviewer in several places throughout the paper, including in a Discussion paragraph detailing limitations (pages 20-21):

“In Experiment 2, but not statistically in Experiment 1, participants overcompensated for the MAE, a result we did not predict. This, together with a lack of calibration between expected illusion strength and MAE compensation (Fig. S1), could suggest a suboptimal compensation strategy. However, model comparison favored the optimal perceptual-insight model over other potential alternatives, including suboptimal compensation strategies at intermediate (e.g., category-prior model) or late (e.g., late compensation) stages which could still reflect some form of insight – understood broadly in the sense of having knowledge of the illusion and attempting to use that knowledge to adjust decisions. But how could participants overcompensate if they were presumably using an optimal insightful strategy consistent with the perceptual-insight model? One potential explanation is that they employed the correct strategy and generative model but nonetheless used incorrect expectations about MAE illusion strength, which could have been plausibly induced by our experimental design (due to our pre-task instruction using longer-lasting adaptors to illustrate the MAE, which was particularly emphasized in experiment 2; see Methods S1.2, S1.3). Future work systematically manipulating expectations about MAE strength is needed to test this directly. “

R1.P3- Relatedly, because there is no baseline condition in which no training about the illusion was given to participants, the authors cannot be certain that the adapt-believe condition reflects insight gained by training and not just some other general compensatory strategy. I do not

expect the authors to run additional experiments to investigate this, but it would be good to discuss this potential limitation in the paper.

This is a good point and something that indeed we have considered. There is indeed no baseline condition in which no training about the motion-after effect (MAE) illusion was given to the participants. The reason for this is that we assumed that at least some of our participants would be familiar with the MAE illusion or related illusions, which our experience has confirmed. Because some participants come to the experiment knowing about the illusion and some others do not, this difference in a priori knowledge could introduce heterogeneity and could interfere with our results and interpretation. Most importantly, this precludes a homogeneous baseline for a potential pre-training condition. We therefore opted to train everyone to reach a common level of knowledge about the MAE illusion and high familiarity with it.

Note also that, as discussed above, we provide evidence for perceptual-insight as a plausible compensatory strategy and against alternative strategies (e.g., category prior shift), including strategies that may be more expected as general compensatory strategies (e.g., a response bias with isolated psychometric shifts such as that induced via response cueing or learned via asymmetric reward feedback in previous work (Gallagher et al., *Scientific Reports*, 2019; Morgan et al., *Attention, Perception and Psychophysics*, 2011; Locke et al., *Attention, Perception and Psychophysics*, 2020), and that observed in our pilot control experiment with response cueing, Supplementary Section 3.9, Fig. S9, pages 49-50). The lack of feedback during the task also reduces the likelihood of this type of generic compensatory strategies.

Critically, our main question was not whether the insight was gained through training alone, but whether, at the end of training, everyone had knowledge about the illusion they could use during the task.

Per the reviewer's suggestion, we now discuss this point as a potential limitation, both in the results section "Participants compensate for distorted perception" (page 8) and in the discussion, at the end of a paragraph about limitations (page 21).

On page 8: "MAE compensation could in principle reflect a more general compensatory strategy not due to insight gained from training and applied during the task. However, we empirically confirmed via MAE strength estimation that participants knew about the illusion and expected to have MAE roughly consistent in magnitude with the observed MAE effect during Adapt-See, albeit not perfectly calibrated (Supplementary Fig. S1). Furthermore, because participants received no feedback during the task, it is unlikely that they could compensate for the MAE via trial-and-error learning. "

On page 21: "Finally, future work should improve upon the current task design to further minimize potential order effects (although see Methods S1.3) and to allow comparisons between pre- and post-acquisition of knowledge about the MAE (Methods S1.3)."

R1.P4- Furthermore, there needs to be some more discussion/explanation on potential order-effects. For example, Adapt-Believe blocks were always done after Adapt-See blocks, which means that slower RTs for Believe conditions might just be due to fatigue. It is argued that Adapt-Believe has to be second because it's harder and thus benefits from the additional task exposure, but you can similarly argue that for this reason, it should be presented earlier so that it is less influenced by fatigue.

While we had added some new analyses on this point in the last version (which were probably difficult to find), we now incorporate more discussion on potential order effects. We also add new analyses that rule out possible explanations of our results in terms of order effects.

If the differences in RTs between the critical conditions – Adapt-See and Adapt-Believe – was due to order effects, we would expect that such differences would not be fully explained by the perceptual decision-uncertainty in the insight model. This is because the insight model does not account for order effects and only considers changes in the decision variable for each stimulus strength and condition. Inconsistent with an explanation in terms of order effects, our GLME analyses of the perceptual decision-uncertainty $-|d|$ showed no residual effects of condition (e.g., Adapt-See and Adapt-Believe) on RTs after accounting for the changes in $-|d|$ by condition.

In new analyses (see point 1 from reviewer 3) we have used a drift-diffusion model (DDM) to provide further support for our interpretations of insight in terms of shifts in the decision variable. In the context of the DDM, fatigue has been argued to produce general slowing of stimulus encoding and response production times, which should be mainly reflected in the non-decision time parameter (Ulrichsen et al., *European Journal of Neuroscience* 2020). We analyzed non-decision times from the winning DDM variant and found no differences between Adapt-See and Adapt-Believe ($t(21)=-0.93$, $p=0.36$), arguing against meaningful differences in fatigue between the two critical conditions.

These results suggest that the main differences in RTs between Adapt-See and Adapt-Believe relate to changes in the decision process rather than more general differences explained by fatigue or order effects. We have incorporated these points in the Methods Section S1.3 (page 28):

“The fixed order of some of our conditions could in principle have led to differential fatigue or practice effects, and systematic differences in RTs, across conditions, but the results of RT analyses speak against this possibility. First, RT did not differ significantly between conditions in the GLME featuring $|d|$. Additionally, in the context of the DDM, fatigue has been argued to produce general slowing of stimulus encoding and response production times, which should be mainly reflected in the non-decision time parameter [119]. Non-decision times from the winning DDM variant showed no differences between Adapt-See and Adapt-Believe ($t(21)=-0.93$, $p=0.36$), further arguing against meaningful differences in fatigue between these two critical conditions. “

R1.P5- Was illusion reproduction (i.e. illusions strength) correlated with the compensatory bias so that participants who experienced a stronger illusion also compensated for it more?

The illusion reproduction did not significantly correlate with the compensatory bias ($\rho = -0.27$, $p = 0.22$). This is not entirely surprising given that the first illusion strength reports were elicited under stronger adaptation (6 s during the training vs 3 s during the task) for the purpose of training and so participants may have plausibly expected for the illusion experienced during the task to be stronger than it actually was. As we explained above, this may have contributed to overcompensation and to the non-significant correlation. Note that our intention in recording illusion strength estimates was not primarily to test for calibration, but to gather independent evidence that participants had knowledge about the direction and approximate strength of the illusion. Also, note that the illusion strength estimates were not expected to be fully precise because they are given while the participant may have experienced some residual MAE. That said, one possible interpretation is that participants do not calibrate perfectly, suggesting that incorrect estimates of MAE distortion (μA) could represent a potential source of

suboptimality for the insight computation. Indeed, we would speculate that insight impairments could arise through a similar process (e.g., where μA is zero where it should be non-zero).

We now add this explicitly in the discussion (pages 20-21, same quote as for R1.P2) and in the legend of Figure S1 (page 40).

In the legend of Figure S1 (page 40): “However, the explicit estimates of MAE strength and the fitted absolute μ likelihood values were not significantly correlated with each other across participants ($\rho = -0.27$, $p = 0.22$). Note that these illusion-reproduction estimates were based on longer adaptor exposures than the 3-s adaptor stimuli participants saw on each trial of the main experimental task, which precludes a direct comparison.”

R1.P6.- Considering that the perceptual-insight model in which the compensation happened at an intermediate, perceptual inference level was the only model that predicted an in-tandem shift in the confidence curves and that such a shift was indeed observed, what exactly do the model comparison and the pupillometry add to this (besides the control that participants are really looking at the adaptor)? As the manuscript is written now, it seems like both analyses are merely another way of showing the same thing. Can the pupil data and the model fits discount any potential alternative explanations that cannot be ruled out based on the confidence data alone?

The reviewer brings up an important set of points that we now address in more detail. We apologize for the lack of clarity in this regard, but in contrast with the reviewer’s suggestion, the perceptual-inference model is not the only model that can explain an in-tandem shift in choice and confidence curves. Indeed, the category-prior model predicts a qualitatively similar shift in tandem (Figure S4 and the newly added Figure S5 described for R1.P1). Therefore, the formal model comparison allows for a more meaningful arbitration between different computational/cognitive strategies, providing more meaningful support for the perceptual-insight model by showing this is the best-fitting and therefore the most plausible explanation.

With regard to the pupillometry data, we believe that it provides independent and converging support for our interpretation. Unlike the subjective reports of confidence, the pupillometry data represents a more objective readout of the internal decision process (see Joshi and Gold, 2020). Pupil dilation is also less likely to be contaminated by late-stage response biases related to motor responses (i.e., button press) and contains temporal information that allows us to isolate the relevant time period during which intermediate processing should occur. Considering these qualitative differences, the pupil data provides unique non-redundant information. And we now added independent analyses that provide convergent support. In particular, we performed a model-agnostic repeated-measures ANOVA that shows differences in pupil dilation for different stimulus strengths between Adapt-See and Adapt-Believe (interaction: $p=0.013$). Critically, the observed patterns of pupil dilation (Fig. 6B) driving this interaction were consistent with the changes in perceptual decision uncertainty in the insight model, as tested more formally using model-based analyses in Fig. 6C and D. We have added more information on this point in the Results section “Pupillometry further validates the perceptual-insight model” (page 15):

“These differences manifested as statistically significant interactions between test stimulus speed and condition (Adapt-See, Adapt-Believe) in two-way repeated-measures ANOVAs in both a stimulus-locked decision-related period (2000-2500 ms after stimulus onset, $F(10,210) = 5.7$, Greenhouse-Geiser corrected $p = 7.7 \cdot 10^{-5}$) and in a response-locked decision-related period (500 - 1000 ms post response, $F(10,210) = 3.1$, Greenhouse-Geiser corrected $p =$

0.013). Thus, pupil dilation patterns during relevant decision-related periods differed between Adapt-See and Adapt-Believe, possibly consistent with shifts in perceptual-decision uncertainty.
“

R1.P7- Fig. 5A in the text seems to refer to the graphical models but Fig. 5A only presents the model comparison results, not the model architectures. This makes it a bit difficult to follow this section.

We agree with the reviewer and now modified figure 5 (page 14) such that fig. 5A recapitulates the graphical model from Figure 1 and indicates the decision rule for our model with perceptual insight.

A**B****C**
Adapt-See Adapt-Believe No-Adapt-See No-Adapt-Believe

D**E**
R1.P8- Furthermore, after rereading, I now find the discussion section a bit unclear. Specifically, this section starts by stating that compensation of illusory perception happens at the 'inferential' level, but for someone who has not read the entire paper, this is quite vague. Please reiterate in a sentence or two what that means and why it is interesting. Finally, it is stated that the current study provides evidence that beliefs about reality arise from higher-level processes, but how? Don't the results show that these are implemented at a perceptual-decision level, so at an intermediate stage? More explanation about this is necessary as well as more discussion about how these results fit with other theories of reality testing.

We appreciate the opportunity to clarify this point. When we claim that MAE compensation is likely to occur at the intermediate inferential level, we are referring to a higher-order decision process such as that implemented in LIP and in some prefrontal regions in the context of perceptual decision-making. We consider this to be higher level relative to lower-level sensory areas representing momentary sensory evidence, such as MT in the relevant case of motion perception. This is similar to previous perceptual decision-making models of motion discrimination, where lower-level sensory regions like MT encode motion-related sensory evidence and higher-level "decision or inference" regions like LIP (and likely a broader parieto-fronto-striatal network), which receive direct projections from MT, accumulate this evidence to facilitate a decision (Beck et al., Neuron 2008). Evidence accumulation like that supported by these higher-order regions is mathematically equivalent to Bayesian inference (Gold and Shadlen, Annual Review of Neuroscience 2007; Liu et al, Neural Computation, 2009; Bitzer et al, Frontiers in Human Neuroscience 2014), so we speculate that inferential operations in the context of our motion discrimination paradigm may involve some of the same regions. Note that we consider LIP to be part of this higher-order network, in line with substantial modeling and monkey physiology work (Beck et al., Neuron 2008). This is because, similar to prototypical higher-order areas (e.g., DLPFC, and despite some differences), LIP exhibits classic higher-level signatures such as persistent neuronal firing during delay periods and relatively long neural timescales, which are thought to enable higher-order computations such as online maintenance of information and sensory integration/accumulation (this is precisely why LIP has been so heavily studied in monkey electrophysiology work).

Because these higher-order regions (including LIP and other higher-order frontal regions) are thought to be involved in inferential processes, and because our converging evidence suggests that MAE compensation involves an inferential mechanism, we suggested that perceptual insight likely involves higher-order brain regions. Note critically that although we call it an intermediate level, we are considering the perceptual inference level a relatively higher-level process, and we use the term intermediate in the temporal sense (rather than an anatomical hierarchy) to contrast it with a late process related to response implementation. Also, our algorithmic model does not speak to specific neural implementation, and it is entirely possible that multiple and even higher hierarchical levels are involved in the insight-related operation. For instance, it is possible that unlike standard inference (which could be based on LIP or more likely a network including fronto-parietal and striatal regions), insightful inference requires additional input from frontal regions such as AMPFC, paracingulate cortex, and others involved in source monitoring and metacognition (reviewed in Dijkstra et al., Neuroscience and Biobehavioral Reviews 2022). We now have added more discussion on this point.

We have also attempted to link our findings to various theories and notions reviewed in the above paper. In general, we believe that our results are consistent with hierarchical views of reality testing and metacognition (e.g., Gershman, Frontiers in Artificial Intelligence 2019; Lau,

PsyArXiv 2019), with a likely contribution of higher-order regions in perceptual insight that is separate from lower-level sensory regions. We provide ample discussion in the paper for why an early sensory process is unlikely to drive MAE compensation (in particular because the illusory percept is still experienced during compensation). These hierarchical models have focused on the evaluation by higher-order regions of the reliability of lower-level sensory representations to infer whether the latter reflect the external world rather than internal signals (Dijkstra et al., 2022). In the context of our task, we hypothesize that higher-order regions may similarly be involved in inferring the external state of the world through identifying and correcting systematic biases in lower-level representations, where this type of bias correction is explicitly built into the insightful inferential process (through the change in the likelihood term incorporating knowledge about the internal distortion A). While more meaningful connections to other theories and literature, including the line of work on imagery-related reality testing, will require additional work, we have attempted to integrate more of these concepts in the discussion, to at least connect our work to these other lines of work more directly.

We find it particularly interesting to contrast our approach to imagery-related paradigms (work by Dijkstra et al.) in that the MAE percept, unlike imagery, is not self-generated and is unlikely to depend meaningfully on top-down mechanisms involving higher-order regions (i.e., evidence suggests that MAE mostly involves a low-level form of saturation or stimulus-specific adaptation in sensory neurons). The fact that we observe a dissociation between conscious experience and explicit beliefs is particularly interesting in this case, where the potentially confused signal (MAE percept) is internally driven, which could suggest that our paradigm taps into higher-level processes selectively at the level of evaluation or bias correction. Because of this, our results dissociating experience and beliefs may be strongly suggestive of the presence of a higher-order process selectively involved in inferences about reality, a process which is separable from conscious perceptual experiences (even if these are not confined to lower sensory levels and also involve distinct higher-level circuits, in line with some work, (e.g. Van Vugt et al., Science 2018). We discuss that contrasting our paradigm to imagery-based paradigms may be a particularly fruitful avenue in dissecting different hierarchical processes involved in reality testing.

In the Discussion (pages 19-20), we now have the following parts:

“The novel systematic dissociation between perceptual experience and belief we achieved empirically, and our multi-stage perceptual-insight framework, are conceptually related to previous hierarchical views of conscious perception and metacognition; these views have generally proposed that beliefs about reality arise from higher-level processes (possibly supported by higher-order prefrontal regions such as anteromedial prefrontal or paracingulate cortex involved in source monitoring and metacognition [8, 11, 90]) distinct from lower-level processes supported by earlier sensory regions. Our observed dissociation between perceptual experience and beliefs seems consistent with this notion, and could perhaps suggest the involvement of higher-order regions in monitoring and *correcting* biases in lower-order sensory representations. Note however that our algorithmic model does not speak directly to neural implementation and that the ‘intermediate-level processes’ we refer to may encompass different neural hierarchical levels – e.g., from posterior parietal regions routinely involved in perceptual inference [70], to abovementioned prefrontal regions which could be more selectively involved in insightful inference requiring top-down signals for sensory bias correction. Our computational account of reality testing is also conceptually related to a previous proposal based on generative adversarial algorithms [71, 72]. Future work is warranted to evaluate and directly compare our framework to these theoretical and neuroanatomical accounts, as well as to other experimental

paradigms of reality testing [8, 73–75] (particularly imagery-based paradigms that may require higher-order processes for imagery generation and reality monitoring [11]). “

“Given this, and consistent with recent work [80], we interpreted the observed shifts in tandem associated with MAE compensation as changes at an intermediate perceptual-decision stage (‘intermediate’ here broadly referring to a process preceding response implementation). Consistent with this interpretation, microstimulation of decision regions such as the lateral intraparietal area (LIP) also produces shifts in tandem [81]. An explanation in terms of late compensation seemed less tenable given that changes in the response rule at late stages - likely implemented in distinct downstream regions [82, 83] - can manifest as isolated shifts in psychometric curves [41], (as reproduced in our response-bias pilot experiment; Supplement S3.9), in contrast to our main results.”

Minor comments:

R1.P9- In experiment 2, the overcompensation in the Adapt-Believe condition seems even stronger than the illusion effect in the Adapt-See condition compared to control conditions – is this statistically significant and do the authors have an explanation for this?

Yes, it is (page 8); indeed, the MAE compensation depicted in Fig. 3C, bottom, is significantly higher than MAE (i.e., higher than 1, $z = 3.94$, $p = 7.99 \cdot 10^{-5}$), which indicates overcompensation. We incorporated a possible explanation for this in our response to the first major point (please also see the Discussion text added in response to reviewer’s point 1, R1.P1). Briefly, we believe that due to observers seeing a longer adaptor for illustrative purposes during some parts of training and the estimation task, they may have believed that the illusion is stronger and thus overcompensated. We do not believe overcompensation is a general feature, as we do not have statistical evidence for it in Experiment 1 (or in additional ongoing experiments not shown here), but rather a potential result of our experimental choices to facilitate training (e.g., exposure in training to stronger MAE than that in the task).

R1.P10- “Despite the broad societal implications of reality testing, the cognitive mechanisms underlying this ability are almost completely unknown” – I find this a very strong statement, especially since there is quite some recent work by e.g. Phil Corlett into the cognitive mechanisms underlying insight. Please discuss how the current study relates to that existing work.

We appreciate this feedback. We agree the wording was too strong and we have toned it down (to “insufficiently understood”). While we are very familiar with Phil Corlett’s work, and we conducted an in-depth search for any work on insight from his group, we were unable to find papers in this area. We emailed him and shared our preprint, and he said he had not done work directly related to insight that would be relevant to our paper. He however mentioned his group’s paper on mediated learning in rats (Fleming et al., *Psychopharmacology* 2022) as his closest work that might be of relevance. Perhaps the reviewer was referring to this paper? In this work, Fleming et al. state that mediated learning “probes the extent to which animals can distinguish between real percepts and internally retrieved representations of cues”. However, this is about the only mention in the paper that may directly touch on reality testing or related constructs. Indeed, the discussion does not seem to emphasize this angle, only suggesting potential links with strong prior models of hallucinations. Furthermore, our reading of the literature (including using parallel tasks in humans, such as Wimmer and Shohamy, *Science* 2012) is that standard

models and interpretations of mediated-learning effects are at the level of associative learning mechanisms, and do seem to invoke any concepts along the lines of reality testing or monitoring (i.e., humans seem to be able to engage in mediated learning without any form of source confusion that we are aware of, at least in the standard paradigms). Given this, we mentioned in the discussion (page 19, see below) the need to understand the links between our paradigm and other existing paradigms that may tap into related processes, citing this and other relevant work in general terms. We would happily include more specific information if the reviewer considers it pertinent and can clarify further. Here, we provide the text added to the Discussion section, in page 19:

“Future work is warranted to evaluate and directly compare our framework to these theoretical and neuroanatomical accounts, as well as to other experimental paradigms of reality testing [8, 73–75] (particularly imagery-based paradigms that may require higher-order processes for imagery generation and reality monitoring [11]). “

R1.P11- Please mention briefly in the text that the colours of the fixation crosses were changed in experiment 2 to keep external input the same between conditions for eye-tracking.

We have incorporated this point in the Methods description of Experiment 2, under “Trial structure with gaze fixation control” (page 27).

“This yellow dot signaled the onset of the test spiral in both See and Believe conditions (unlike in Experiment 1, where different colored dots were used for See vs. Believe) to keep stimulation identical between these conditions.”

R1.P12- Please add some more details on why the assumption that the offset A has negligible uncertainty in the perceptual insight model is an appropriate assumption to make and how it might influence results.

As we now mention in the Methods section and address in the Supplement, we performed simulations and parameter recovery to provide evidence that the simplified perceptual-insight model captures the relevant features from the full perceptual-inference model, confirming a negligible effect of the uncertainty around the distortion factor A (and flexibility to capture small effects within the simplified model architecture). In the Discussion (page 12), in the Methods (page 35) and in more detail in the Supplementary section S3.8 “Simulation of full perceptual-insight model” (page 48), we now address the reviewer’s point with the following text and figure:

In the Discussion (page 12):” While our simplified insight model (Section S2.2) assumed no additional uncertainty related to the Adaptor ($\sigma_A \approx 0$), it captured increased encoding noise in Adapt-Believe relative to Adapt-See via condition-specific σ_{encoding} , likely reflecting a contribution of σ_A (Supplementary Fig. S8). “

In the Methods (page 35): “Nonetheless, the model had flexibility in this respect as it allowed σ_{encoding} parameters to vary in each condition, which could – and seemingly did – absorb additional uncertainty likely associated with A in the Adapt- Believe condition. Simulations of the full model allowing a σ_A different from zero showed negligible effects on measured values of the μ likelihood and σ_{encoding} (with the latter absorbing non-zero σ_A values at low levels of encoding noise), supporting the validity of our simplified model (see Fig. S8). “

In Supplementary, S3.8 (page 48): “ Additionally, to account for the possibility of the full generative model including uncertainty over the magnitude of the distortion due to factor A - uncertainty σ_A around the expected magnitude μ_A - we generated simulated data according to this full generative model and fitted the simplified perceptual- insight model we use as our main model in the paper. The $\mu_{\text{likelihood}}$, the main parameter of interest, is mostly robust to σ_A and can be recovered well (Fig. S8A, Spearman correlation $\rho = 0.64$, $p = 8.0 \times 10^{-23}$). σ_A can have a small effect on σ_{encoding} , although this is limited to low levels of σ_{encoding} . In Fig. S8B, we see that fairly low levels of σ_{encoding} possibly comparable to those in the baseline conditions could be moderately overestimated in the additional presence of high σ_A . However, for higher levels of σ_{encoding} similar to the values from Adapt-See and Adapt-Believe, we see that σ_A does not additionally influence fitted σ_{encoding} values (Fig. S8B). An increase in σ_A does not lead to an increase in the fitted σ_{encoding} (Fig. S8B), but the fitted σ_{encoding} does increase when the underlying value of σ_{encoding} does (Fig. S8C). In sum, while the simplified perceptual-insight model ignores σ_A , we showed that this should not affect our main measure of $\mu_{\text{likelihood}}$; underlying σ_A has a non-additive effect on the fitted σ_{encoding} , but this effect should have a negligible for higher levels of σ_{encoding} or else be absorbed by the fitted σ_{encoding} parameter at lower underlying levels of σ_{encoding} . Given this, it is plausible that the increased σ_{encoding} values we observed under the Adapt-Believe condition may at least partly represent uncertainty around the expected distortion A . “

Figure S8. Simulations showing limited effect of σ_A on model fitting via the main (simplified) perceptual-insight ($\mu_{\text{likelihood}}$) model. Simulations show fitted parameters using the $\mu_{\text{likelihood}}$ model to simulated data based on the full generative model of perceptual insight (including uncertainty σ_A around the expected magnitude μ_A of the distortion factor A) We fixed

$k_{\text{confidence}}$ to 0.9 and simulated 5 sets of 121 trials for every combination of parameters. We show the fitted parameters as a function of the parameters used to simulate the data: A) $\mu_{\text{likelihood}}$ (Spearman $\rho = 0.64$, $p = 8 \cdot 10^{-23}$). B) σ_{encoding} C) σ_A .

R1.P13- The second sentence of the discussion is grammatically incorrect.

Thank you for catching this. We fixed it in the revised version, which reads:

“We further designed a controlled “cognitive psychophysics” paradigm [40] to capture this form of in-the-moment reality testing while minimizing memory demands.”

R1.P14- The discussion reads ‘model comparison and ‘validation’ – what is meant by ‘validation’ in this context? Relatedly, what does ‘supporting our conceptualization of perceptual insight’ mean?

For clarity, we now simplified to “model comparison” (page 19):

“Model comparison lent further support for a model of perceptual insight involving adjustments at this inferential level. The shifting perceptual-variable in our model provided a parsimonious explanation for RT and pupil-dilation patterns, providing further support for our model of perceptual insight and suggesting humans’ ability to deploy such insightful strategies to circumvent internal biases. Further support for an interpretation of perceptual insight in terms of the hypothesized shifts in an inferential perceptual- decision variable came from DDM analyses (jointly fitting choice and RTs but not confidence reports), suggesting robustness of our conclusions to assumptions about confidence generation. “

Reviewer #2 (Remarks to the Author):

Overall impression: This is an interesting paper with careful measurement and modeling on a difficult topic. Some of the framing could be clearer and there might be alternative explanations for the observed phenomena.

Summary: This paper asks whether and how people can discount their sensory responses to an illusion when they know they are experiencing an illusion. Specifically, subjects are given some brief instructions and demonstrations of how motion aftereffects work, and then they make judgments about the direction of motion of test stimuli while experiencing motion aftereffects. They also make confidence judgments in each trial. In one condition, similar to the usual adaptation experiments, subjects are asked to report what they see. This condition produces the expected features of an adaptation experiment: adaptation to a counterclockwise rotating spiral causes a shift in the psychometric function toward the adaptor, and reaction time is slowest (and confidence lowest) at the point of subjective equality rather than at the physically neutral point. In contrast, when subjects are instead asked to report what they *believe* the stimulus is doing

(rather than what it looks like), they undo the shift in the psychometric function (and even overcompensate a bit). The most important result for the authors is that in this "believe" condition, uncertainty, as reflected in slow response times and low confidence, shifts with the choice (i.e., subjects are slowest and least confident near where the choice curves are at 50/50). The interpretation, based both on the qualitative pattern of results and on formal model fitting and model comparison, is that subjects incorporate their knowledge of the distortion at an intermediate stage of processing, that is later than sensory encoding but earlier than a response bias. The authors describe the effect as perceptual insight that shifts the perceptual decision variable.

Issues / questions

R2.P1. To be frank, I could not make much sense of the exposition about "perceptual insight". I do not know what the authors mean by this term, perhaps because I am not an expert in the metacognition literature (though I am familiar with sensory adaptation). I could not really figure it out from the manuscript, and I would recommend making the meaning of this clearer, since it is the central point of the paper. That said, the models are concrete and specific, and the differences in predictions between a late effect (shift in choice curve but not in confidence/RT curves) vs an intermediate effect (shift in both) are clear enough. The data are also reasonably clear (all functions shift, not just choice). Hence the difficulty in understanding the terminology is only a modest concern.

We appreciate the opportunity to clarify this point. We have attempted to improve our exposition of "perceptual insight" throughout by adding more intuition in the introduction and the model description section. In simpler terms, our goal is to understand how people can have perceptual distortions or hallucinations and simultaneously know that such perceptual experiences constitute distortions or hallucinations (that is, are not based in reality). Some people (e.g., patients with neurodegenerative disease or strokes, or people under the effects of psychostimulant drugs) can sometimes have florid hallucinations like seeing a snake in front of them that they confidently know is not real – that is, they can have hallucinations with intact reality testing (e.g., Naasan et al, Brain 2021). In contrast, others (e.g., patients with schizophrenia) have hallucinations or distortions that they attribute to the external world with full conviction despite others assuring them they are not real – that is, typical hallucinations with impaired reality testing. We use the term "perceptual insight" to refer to this type of reality testing, which is clinically simply referred to as "insight". To study this phenomenon, we induce a distortion (MAE) and test whether people can reality test by judging the actual motion direction of a spiral stimulus while they perceive illusory motion. Our motivating question was whether people could have the MAE and know that what they perceive is not real? And could they use that knowledge to judge the actual motion of the spiral stimulus while experiencing the illusion? Our data shows that the answer to both questions is yes.

We have made several changes throughout the Introduction, as well as in the Results section "Formal model of perceptual insight" (page 3), highlighted in color.

R2.P2. Is it possible that when the task changes (report what you believe rather than what you see), subjects do not use metacognition (i.e., do not use Bayesian inference that takes into account knowledge of the distortion), but instead simply use different sensory information to inform their decision?

For example, suppose that in the see condition, subjects make judgments based on local changes in position (velocity), which is strongly affected by adaptation, whereas in the believe condition they make judgments based on the change in position relative to a reference, such as the square stimulus aperture or the screen? Put another way, there may be sufficient

information in the stimuli to make reasonably accurate judgments even when experiencing a strong aftereffect. If the answer to my question is yes, would this change the interpretation?

The reviewer brings up a good set of points. While we cannot fully rule out some alternative explanations, we now show evidence that suggests that observers use similar sensory information across Adapt-See and Adapt-Believe to inform their decisions and indeed perform Bayesian inference in both conditions.

First, we want to clarify the setup and show a picture of it. We realize that the way we originally depicted Figure 2 does not clearly convey how large the stimulus is on the screen and we hope that the figure now accomplishes this. Since the spiral filled almost the entire screen (viewable portion of the screen of 59 cm) and their fixation was imposed to a circle of 3 degrees of visual angle around fixation, it is not feasible for participants to use the square stimulus aperture or the screen as reference for their responses.

Second, if as the reviewer suggests observers used different sensory information in Adapt-See and Adapt-Believe, it is possible that they may sample the sensory information differently or covertly allocate their attention differently. To test for this, we first looked at the eye fixation positions. We did not observe any systematic differences in the fixation positions between Adapt-See and Adapt-Believe (mean fixation x-position: $t(21)=1.3$, $p=0.21$; mean fixation y-position: $t(21)=1.1$, $p=0.29$). This was also true when looking across different epochs (fixation, adaptor, test stimulus, and post-stimulus; all $p>0.20$). Next, since microsaccade rates could reflect differences in fixation stability or affect MAE strength (Murakami et al, 2006, Otero-Millan et al, 2012), we quantified the microsaccade rate per condition (obtaining values of $\sim 1/s$ while fixating similar to those in the literature, Cui et al, 2009). We found no significant differences between the microsaccade rate in Adapt-See and Adapt-Believe ($t(21) = -1.1$, $p = 0.28$). This was also true when splitting by epoch (fixation, adaptor, test stimulus; all $p>0.26$). Based on these results, we do not have evidence to believe that observers sample the visual information differently to warrant differences in adaptation strengths across Adapt-See and Adapt-Believe. We condensed these points in the caption of Figure 2 (page 7):

“B) Visualization of the experimental setup and gaze fixation control (see Methods S1.3 “Trial structure with gaze fixation control”). C) The average fixation positions did not differ substantially across the four task conditions (all $p > 0.20$). Microsaccade rates could reflect differences in fixation stability or affect MAE strength [38]. There were no significant differences in the microsaccade rates, (measured following [39] and using parameters minimum velocity threshold multiplier $\lambda = 6$, minimum amplitude threshold 1 dva and minimum duration 6 ms), suggesting that fixation was successfully enforced and that fixation stability was comparable across all conditions (all $p > 0.26$). “

Based on the results above, we do not have evidence to suggest that participants use different sensory information between the two critical conditions, Adapt-See and Adapt-Believe, to inform their decisions. Based on our experience performing the task and the constraints imposed by fixation, we cannot envision a strategy to use different sensory features to respond differently across these two conditions. That said, Bayesian inference can be framed from the perspective of readout of sensory information (e.g., Jazayeri and Movshon, *Nature Neuroscience* 2006). Mathematically, inferential transformations such as those we take to occur under perceptual insight are equivalent to differentially reading out sensory information: even if input from lower-level sensory regions remains the same (e.g., under Adapt-See and Adapt-Believe), higher-level decision regions can read out this input differently, and this change in the

readout can capture the transformations in the Bayesian model that explain changes in decisions. We have now integrated this point in the Discussion (page 20):

“Although speculative, this process could be neurally implemented in decision regions (e.g., LIP) akin to optimal-decoding solutions based on synaptic reweighting of sensory representations [84–89].”

R2.P3. Because there is only one adapt condition (one direction and one strength), it seems not possible to distinguish whether participants have a quantitative internal model of the distortion vs just knowing that there is a distortion (with a point estimate of its effect). Does this matter to the interpretation? This differs from some other literatures, such as reinforcement learning literature, in which experimentalists test whether participants have an internal model of a task or process or setting.

The reviewer is correct that with our data we cannot distinguish whether the observers have a full quantitative internal model with the full distribution over the possible adaptation strengths A versus that they have a point estimate of the expected distortion. While we spell out a full model with a distribution over A , the (simplified) perceptual insight model we fit to our data and which is selected as the winning model only entails a point estimate; this point estimate reflects the mean of the distribution of adaptation strengths, which is our parameter μ likelihood. This is because we made the simplifying assumption that the information about the distortion can be boiled down to the mean summary statistic. While we did not fit the full model (to avoid higher complexity and potential issues of robustness apart from a substantially more computationally intensive fitting), fitting the simplified Bayesian model yielded very good fits. The simplification that the mean statistic captures is a good approximation to the full distribution is indeed consistent with most of reinforcement learning (RL) work, since most RL models (i.e. Q-learning, temporal-difference learning) are simplified Bayesian models that track means and do not update full distributions – i.e., much like our simplified perceptual-insight model, most RL models track (learn) mean expected values.

While with our current data we cannot distinguish between a model with a full distribution over A versus a point estimate of the mean of A , we also provide more information and simulations of the full model (Section S3.8, Figure S8, page 48). As also described for reviewer 1 (R1.P12), we see that fitting the perceptual insight model to simulated data based on the full model yields good recovery of the relevant parameters. The uncertainty of A shows negligible effects and the most critical parameter, the μ likelihood parameter representing perceptual insight, can be measured with the simplified perceptual-insight model we use as our main model. Indeed, in the main text (page 14), we show that the fitted parameters for σ encoding values are higher in Adapt-Believe compared to Adapt-See. Our simulations (Figure S8) suggest that this additional noise during Adapt-Believe could plausibly represent some level of uncertainty associated with the expected adaptation strength. Thus, although we do not have direct evidence and additional experiments are needed to confirm this, our data is at least consistent with the notion that participants may indeed use a full model that represents the distribution of the distortion A .

Finally, other data in the paper suggest that participants use a quantitative model of the distortion. First, we show using MAE strength reproduction that participants know the direction and approximate strength of the illusion (Figure S1, page 40). Also, our model comparison includes models with inferential transformations that do not require a quantitatively appropriate generative model of the distortion (e.g., the category-prior model), and the perceptual-insight model was clearly selected over these alternative models (Table S1, page 38). Therefore, while

definitive evidence is needed, several results are consistent with our interpretation that participants use a quantitative generative model of the expected distortion associated with the MAE.

R2.P4. [Minor] In the introduction, a perceptual distortion is described as "a discrepancy between an objective stimulus and its corresponding percept". I believe this description does not make sense, since a stimulus and a percept can never be the same (the stimulus is a physical object, the percept is not). Perhaps it would be better to say, "a discrepancy between a property of a stimulus and an estimate of that property"?

We thank the reviewer for this comment. We agree with the reviewer's point but believe the definition must include a contrast between an objective and *perceived* stimulus feature, and explicitly link the estimate to the perceptual experience (since an estimate of a stimulus feature could plausibly be implicit and in principle need not be consciously experienced perceptually). While attempting to address this comment, we have changed the definition of perceptual distortion to "a discrepancy between an objective stimulus feature and its subjective perceptual experience or estimate" (page 2).

R2.P5. [Minor] Figure 3A, top: it looks like the curve in the adapt-believe condition is centered slightly to the right of 0, which should produce a bias greater than 0, yet the bias is a bit less than 0 in the bar plot in panel B. Am I misreading the graph or misunderstanding the model parameters?

Great observation! We apologize for this discrepancy, which is due to the fact that the psychometric curves in A depict means across the subjects, while the parameter estimates in B depict medians. While the median takes into account the negative points, the mean is more swayed by the few extreme positive points. Given this, in the case of Experiment 1 a visualization of the psychometric curves with median and bootstrapped 95% CI could be a better representation. However, we decided to keep it as standard mean \pm SEM for consistency with the literature and Experiment 2, in which we do not have such outliers and the mean \pm SEM psychometric curve is meaningful. Note however that presentation of individual data points allows the reader to get a sense of the distribution, and that our statistics use appropriate non-parametric tests. We now add the following line in the caption of figure 3 (page 9):

"Note that psychometric curves are means \pm SEM and that to account for outliers the fitted parameter plots use medians (and 95% CI) more consistent with the corresponding non-parametric tests."

R2.P6. Last: I don't know what the journal's policy is or the author's intentions are with respect to data sharing, but it might be valuable for the community if the data were made public.

We have made the data available on github. Additionally, updated scripts are available at: <https://github.com/lianaan/Insight>

Reviewer #3 (Remarks to the Author):

Mihali et al investigated the capacity for human subjects to adjust their responses to illusory visual stimuli based on their knowledge of the illusory source of the appearance of the stimuli. Using a novel task design that asked observers what they believed the stimulus was in addition to what they perceived the stimulus to be, the authors found that observers (over)compensated their 'belief' responses to adapt to the fact that they had knowledge of the illusory nature of their

perceptions. Modeling of choice and confidence data suggest that the compensatory effect took place at the intermediate level of the decision variable representation, rather than at a sensory stage or at a late (post-decision-variable formation) stage. The introduction of a novel task to measure a clinically relevant feature of cognition and behavior (reality monitoring and insight) along with computational modeling of behavioral data and validation with pupillometry make this a methodologically strong paper with conclusions likely interesting to a broad audience. I have a few concerns that, if addressed, could hopefully make the conclusions of the paper a bit clearer.

R3.P1. My primary concern is about the use of confidence and RT data to make strong inferences about the effect of insight on decision-variable representations as opposed to on response-level criteria. The authors state that a tandem shift in confidence and choice psychometric functions indicate an intermediate decision-level effect, whereas an isolated shift in choice but not confidence curves would indicate a late-stage effect on choice criterion. However, this predicted result seems to be an artificial byproduct of the fact that the model used does not allow confidence criteria to be updated with knowledge in the same way that choice is allowed to vary (i.e., the influence of k_{choice} on c_{prime} in the model). It seems like a reasonable hypothesis that prior knowledge could influence confidence criteria in addition to choice criterion and that this could also lead to tandem shifts in the confidence psychometric functions as a result of a late-stage effect. From an evidence accumulation perspective, where choice, confidence, and RT have been argued to reflect the amount of accumulated evidence at the time a decision is reached, changes to the criteria or changes to the decision variable could produce effects on choice, RT, and confidence. Thus, the use of confidence and RT data here do not seem to uniquely point to an intermediate-level effect. Ideally, some more model comparisons and implementation of a drift-diffusion framework to capture RT effects could be used to bolster the main claim about a decision-level effect.

The reviewer is right that the confidence and RT data do not uniquely point to intermediate-level inference and that the architecture of our main generative model of confidence is not the only one possible. We indeed mention in the methods section that alternative models of confidence have been proposed in the literature. As the reviewer writes, indeed allowing for k_{choice} to also affect confidence in the model's architecture would lead to an offset decision variable " $d - k_{\text{choice}}$ " and shifts in tandem of choice and confidence. We have now included this alternative model – " $k_{\text{choice_confidence}}$ " – as part of our model comparison (Fig 5B, page 14). Importantly, this model underperforms compared to the winning perceptual-insight model. Note that this result is not surprising because the $k_{\text{choice_confidence}}$ still represents a form of late-response bias that does not shift the perceptual decision variable and does not take into account the level of σ_{encoding} (please see reply to point 1 by reviewer 1, which discusses relevant comparisons between the perceptual-insight likelihood model and the category-prior model). Below are the updated Fig. 5 and its caption incorporating the new model proposed by the reviewer.

A**B****C**
Adapt-See Adapt-Believe No-Adapt-See No-Adapt-Believe

D**E**
Figure 5: Modeling of Experiment 2 data supports the perceptual-inference model. A) Simplified schematic of generative model and inference in the perceptual-insight model (recapitulating Fig. 1). Inference in the perceptual-insight model takes into account knowledge of the distortion through the term μ likelihood, which represents the observer's estimate of the distortion due to the factor A from the generative model. B) Model comparison selects the perceptual-insight " μ likelihood" model as winning model. C) Fits from winning model (shaded areas) capture well choice and confidence curves (mean \pm SEM, as in Fig. 4); corresponding fitted parameter values (mean \pm SEM in D). *** indicates $p < 0.001$ and ** indicates $p < 0.01$, with results being based on paired t-tests. E) RT curves mirror the perceptual-decision uncertainty $-|d|$ from the winning model (shown in log space). Data are mean \pm SEM.

In response to the reviewer's comment, we also reanalyzed the data using DDM model variants. We based our approach on previous DDM work on sequential choice-history biases that aimed at disentangling whether such biases in perceptual decision tasks occurred through changes in the decision variable, and at which level (Urai et al., eLife 2019). This previous work mostly focused on changes in the starting point of evidence accumulation, which should induce an early-peaking effect on conditional response function, and separated those from drift rate biases, which should induce a more protracted effect on conditional response function. In the model-agnostic conditional response function (White and Poldrack, JEP:LMC 2014) applied to our data, we see an early-peaking effect whereby biases for both MAE and MAE compensation are apparent for short RTs and disappear almost completely for longer RTs. This is more consistent with a decision-level effect that acts early on the evidence-accumulation process, such a bias in the starting point. Model fitting of DDM variants and model comparison provided further support for this conclusion, suggesting that models with different starting points accounted best for the data. Furthermore, changes in decision bounds did not meaningfully improve model fits, inconsistent with a late-process mainly involving response biases. Altogether, we believe the DDM analyses support our initial interpretation that perceptual insight involves changes at an intermediate decision process, and appreciate this suggestion. Critically, we believe that these new analyses also help address point 3 from the reviewer (below), since DDM fitting was based on RT and choice data, and did not use confidence reports. Therefore, complementary modeling approaches with joint fitting of choice and confidence data (in our main analyses) or joint fitting of choice and RT (in the DDM analyses) converge within a cohesive explanation which does not rely on interpretations about or the model architecture for confidence. The new DDM modeling results are in a new subsection in the revised paper (Results, pages 17-18) and depicted in Figure 7. The new section reads as follows:

"Drift-diffusion modeling supports an intermediate inference-level explanation of perceptual insight

We observed an MAE compensation that manifested as shifts in tandem in psychometric and confidence curves (Fig. 4). This pattern was well captured via a Bayesian process model of perceptual insight that was jointly fitted to choice and confidence responses and explained insight via shifts in a perceptual- decision variable at an intermediate inferential stage. However, this model assumed shared computations across See and Believe confidence reports and one of several possible architectures (Methods S2.1, [58– 61]).

To assess the robustness of our conclusions, we used an alternative framework based on drift-diffusion models (DDMs). To avoid strong assumptions about confidence generation (as in

some DDM formulations [62]), we opted for standard DDMs that jointly model choice and RTs, and which have proved useful to parse biases in decision-making [63, 64]. In its classic form, the DDM assumes that evidence supporting one of two decisions is initially unbiased or biased (with a starting-point bias of 0, or different than 0, respectively), and accumulates over time with speed determined by the *mean drift rate* and tracked via a noisy decision variable at an intermediate stage. When this decision variable reaches one of two *decision bounds* at a later stage, after some motor preparation and production time (part of *nondecision time*), the observer reports their decision.

Previous DDM work has suggested that decision-making biases can result either from changes in the starting-point bias of the evidence-accumulation process or biases in the drift rate of accumulation towards one decision. While a non-zero starting-point bias would favor a specific choice by starting closer to one decision bound, a drift rate bias would increase the rate of evidence accumulation towards one decision. These two scenarios have dissociable patterns on the conditional response function (Fig 7A) of the choice bias as a function of RT quantiles: under a starting-point bias, the choice bias manifests at short RTs and disappears quickly as RTs increase; under a drift rate bias, the choice bias decays slowly with increasing RTs. Potentially consistent with a starting-point bias, we saw that biases in the conditional response function plots in Adapt-See (MAE) and Adapt-Believe (MAE compensation) were most apparent at shortest RTs and disappeared relatively quickly with increasing RTs (Fig 7A).

To quantitatively parse the source of the choice bias, we fit DDM variants using a previously validated method [65]. We considered a "base" variant including 3 free parameters (per condition) for mean drift rate, nondecision time and decision bound and extended variants including a free parameter (per condition) for starting-point bias, one for drift rate bias, or both (Fig 7B). The model with starting-point bias as an additional parameter per condition fit the data best across all conditions according to BIC (Fig 7B); the winning model captured the choice and RT data reasonably well (Fig 7C) for the limited number of trials for DDM analysis [66]. The parameters of the winning model were informative (Fig 7D). First, Adapt- See differed from No-Adapt-See in the mean drift rate and starting-point bias, partially consistent with previous work [66] and with sensory-level explanations for MAE [43]. Critically, Adapt-See and Adapt- Believe differed in the mean drift rate and in the starting-point bias. Differences between Adapt-Believe and Adapt-See in the mean drift rate correlated with corresponding differences in σ_{encoding} also indexing sensory noise in the perceptual-insight model (Fig. 5C, $\rho = -0.58$, $p = 0.0057$). Most critically, differences in the starting point, but not in other parameters (all $p > 0.34$), correlated strongly with the degree of MAE compensation ($\rho = 0.85$, $p = 5.13 \cdot 10^{-7}$) and changes in the perceptual-insight $\mu_{\text{likelihood}}$ parameter ($\rho = -0.88$, $p = 2.79 \cdot 10^{-6}$). A bias in the starting point of perceptual evidence accumulation (obtained via joint choice-RT fitting) thus corresponded with the shift in the perceptual-decision variable in the perceptual-insight model (obtained via joint choice-confidence fitting) that explained MAE compensation. Importantly, we found no significant differences in decision bounds between conditions (which may have supported a late-stage response process). Overall, our DDM results support an implementation of perceptual insight at an intermediate stage of perceptual decision-making, and argue against a mechanism confined to a late-response stage. "

Figure 7: Drift-diffusion modeling. A) Conditional response function [63,64] for our data (left, mean \pm SEM) and schematic (right, adapted from [63, 64]) illustrating differential effects of starting-point biases versus drift-rate biases. RTs are binned into 5 quantiles. B) Using the PyDDM package [65], we implemented a base DDM (with parameters mean drift rate, decision bound and nondecision time) as well as variants with additional parameters (starting-point bias, drift-rate bias or both for each condition). Model comparison selects the starting-point bias DDM variant as the winning model (median \pm 95 % CI). C) Fits of the winning DDM with starting-point bias capture choice and RT data satisfactorily. D) Fitted parameter values from the winning DDM model above (mean \pm SEM). *** indicates $p < 0.001$, with results being based on paired t-tests.

R3.P2. Related to the point above I don't quite see the theoretical argument for why a late-stage effect would not amount to true insight. If prior knowledge has the effect of changing response criteria such that my behavior is adapted in light of my prior beliefs, why wouldn't this count as a form of insight? Some more discussion of these different mechanisms (e.g., decision versus response) and their implications would be insightful.

The reviewer again raises an important consideration. We agree with the reviewer that late-stage changes at the response level *could possibly* represent a form of genuine insight where knowledge about the perceptual distortion drives the change in response criterion. However, simply observing this type of response bias does not guarantee an underlying insight-based strategy; a response bias could be at least equally consistent with a strategy in which accuracy is enhanced in the absence of explicit knowledge about the perceptual distortion (e.g., the individual knows they are making incorrect choices and adjusts their responses, but they do not know the underlying reason is a perceptual distortion). In other words, a response bias need not indicate – and cannot be taken to necessarily reflect – an insight-based strategy. Furthermore, such a response bias, even in a case driven by an insight-based strategy, would not be optimal. This is because, like a shift in the category prior (see our response to point 1 of reviewer 1, R1.P1 and S3.5 and Fig S5, pages 44-45), a response bias would not use an appropriate generative model and appropriately account for changes in encoding noise (and therefore stimulus contrast). This could lead to a scenario where no visible test stimulus is presented and a biased response is given. In contrast, the intermediate inference-level change under the perceptual-insight model represents an optimal strategy (with unbiased responses in the case of non-visible test stimuli) in that it uses the appropriate generative model and prior. Most critically, behavioral evidence for this type of intermediate inference-level change additionally guarantees an underlying insight-based strategy. We have now clarified this nuance on page 3 in the manuscript, when we first introduce the models, and elaborated further on these points in the discussion (page 20):

In the Introduction (page 3): “While an insightful agent could possibly incorporate knowledge about factor A suboptimally by adjusting its response rule, observing a response bias does not guarantee an insightful strategy (as it is also consistent with the absence of insight). In contrast, observing the abovementioned shift in d incorporating knowledge about factor A (as in Fig. 1B, right) does imply the use of an insightful strategy. “

In the Discussion (page 20): “However, model comparison favored the optimal perceptual-insight model over other potential alternatives, including suboptimal compensation strategies at intermediate (e.g., category-prior model) or late (e.g., late compensation) stages which could still reflect some form of insight – understood broadly in the sense of having knowledge of the illusion and attempting to use that knowledge to adjust decisions. “

That said, model comparison favored the optimal perceptual-insight model over other potential alternatives, including suboptimal compensation strategies at intermediate (e.g., category-prior model) or late (e.g., late compensation) stages which could still reflect some form of insight – understood broadly in the sense of having knowledge of the illusion and attempting to use that knowledge to adjust decisions.

R3.P3. Lastly, I think more could be said in the primary manuscript about the nature of the confidence report. Typically, confidence is construed as the probability that a decision was correct and subjects are instructed to rate higher confidence when they felt their choice was more likely to be correct. However, this only applies when the first-order choice is about a

stimulus property (as in the 'believe' condition here). In the 'see' condition, participants know their choices are often incorrect but are responded to provide a confidence judgment anyway. My first comment is just for clarification – how were confidence reports construed to the subjects in both of the tasks? My second comment is whether this inherent difference in what confidence is referring to in the two tasks could partly drive differences in confidence between the see and believe conditions.

This is an important point we are happy to elaborate upon. In response to the first comment about how confidence reports were presented and construed by subjects, we asked participants to report, after choosing, “how sure are you about your response?” They specifically were asked to press the up key if “very sure” and down key if “not very sure”. This was true for both Adapt-See and Adapt-Believe conditions, as well as the baseline conditions. We have clarified this further in the caption of Figure 2 (page 7) and in the Methods (page 25).

“Figure 2. A) The four task conditions differing on the presence of a rotating spiral adaptor or static control spiral (Adapt, top blue, versus No-Adapt, bottom green) and required responses (See, dark colors, versus Believe, light colors) are depicted. Details (e.g., prompt colors and response-window duration) correspond to Experiment 2 (see Methods S1.3 for minor differences in Experiment 1). Blocks start with a reminder of required responses followed by a 30-s spiral rotating counterclockwise at constant speed or a static control spiral. Trials start with a fixation screen followed by the first spiral (rotating adaptor at constant speed counterclockwise or static control) and then a second spiral (test stimulus) of variable speed. A binary left/right choice is then prompted about motion direction (for clockwise or counterclockwise motion in the test stimulus, respectively, as seen or believed depending on the required response), followed by an up/down confidence response, **provided to the question** “how sure are you of your response?”. “

With regards to the second comment, it is possible, as the reviewer suggests, that confidence reports may have been construed differently in the Adapt-See and Adapt-Believe conditions. Participants knew that the Believe conditions asked them to decide on the actual stimulus, whereas See conditions simply asked them to report their internal experience. Therefore, confidence in the Adapt-See condition referred to a subjective experience with no correct answer; confidence in the Adapt-Believe condition referred to an actual external process and had a correct answer. Participants were instructed to respond accurately and consistently, but it may be reasonable to expect that there were some qualitative differences in their confidence construal across the two critical conditions. That said, we think that these differences are well captured by the perceptual-insight model. The model takes confidence reports to be a binarized version of the probability of the choice reflecting the correct category given the internal measurement, $p(C = \text{Chat}|x)$. This is directly what subjects should be doing in the Believe conditions: report confidence about their choice about the external category. In the See conditions, they do not report this exactly. However, if during the Adapt-See condition subjects are not using the known structure of the distortion factor on the internal measurement x , reporting confidence about their choice about the external stimulus category may be nonetheless indistinguishable from reporting their internal measurement x (more similar to a heuristic model that would be hard to disambiguate in the See conditions in this task). Furthermore, it is plausible that subjects are simply choosing and reporting confidence as if they always refer to the external stimulus category, since these two strategies only differ meaningfully when one assumes that the internal measurement x depends on more than just the external stimulus s (i.e., when subjects take into account that x depends both on s and on

the distortion factor A – that is, specifically in Adapt-Believe). Therefore, the flexibility of the model to use a more complex inferential structure in Adapt-Believe compared to Adapt-See, accounts for quantitative and likely for qualitative changes in the decision process and confidence reports that go at least some way towards addressing this point.

One critical addition that should further allay remaining concerns in this regard is our new section on DDM analyses described in response to point 1. We opted to use a standard DDM approach with joint modeling of choice and RTs partly to address this point on potential differences in confidence reports (and partly because the debate on extended DDMs accounting for confidence is ongoing and no consensus seems to exist on a preferred architecture/approach, based on our discussions with colleagues working in this area). Critically, our DDM analyses which do not use confidence reports support our main conclusion that perceptual insight involves intermediate-level changes in the perceptual decision variable (i.e., in the inferential process), indicating that our results are robust to different interpretations or construal of the confidence reports across task conditions. Given these converging results, and the pupillometry results (not using confidence or RT data), we believe it is fair to say that our conclusions are supported by independent types of data and are robust to specific assumptions about confidence.

Reviewers' Comments:

Reviewer #1:

Remarks to the Author:

This is an important paper that introduces an elegant and technically advanced approach to characterize the under-investigated phenomenon of perceptual insight. The reviewers have done an admirable and thorough job at addressing my concerns. I only have a few minor issues left.

With respect to the difference between the likelihood and the prior model, I really appreciate the additional simulation and explanation that the authors have added. It might be good to explicitly mention the predictions for future experiments based on these new simulations in the manuscript. As far as I understand it, these simulations show that the prior model would be bad at compensating for the illusion if the stimulus had a very low contrast. This is something that could be directly tested in future research.

I appreciate the added explanation of how the current results connect to the notion of 'higher-level processing' for reality testing, however, I am not entirely sure how the imagery research the authors refer to is relevant here. For clarity, I would remove this reference (e.g. like "(particularly imagery-based paradigms that may require higher-order processes for imagery generation and reality monitoring [11]).").

I would like to apologize for my incorrect reference to Phil Corlett for research on insight - I vaguely remembered he did something in this area, but that was incorrect (as the authors found out after their thorough search!), so, my apologies for not double checking before making this comment.

Reviewer #2:

Remarks to the Author:

The authors have been highly responsive to the reviews and addressed all major concerns.

Reviewer #3:

Remarks to the Author:

The authors have nicely responded to my comments. The addition of extra models and a whole new modelling framework that incorporates RT has made the paper much more comprehensive. Thank you.

REVIEWERS' COMMENTS

Reviewer #1 (Remarks to the Author):

This is an important paper that introduces an elegant and technically advanced approach to characterize the under-investigated phenomenon of perceptual insight. The reviewers have done an admirable and thorough job at addressing my concerns. I only have a few minor issues left.

With respect to the difference between the likelihood and the prior model, I really appreciate the additional simulation and explanation that the authors have added. It might be good to explicitly mention the predictions for future experiments based on these new simulations in the manuscript. As far as I understand it, these simulations show that the prior model would be bad at compensating for the illusion if the stimulus had a very low contrast. This is something that could be directly tested in future research.

Response: “We discussed that the category prior model differs from the perceptual insight model in that the former incorrectly assumes a change in the category prior that produces fixed shifts in d regardless of encoding noise (Supplementary Fig. S5), and this may be tested in future experiments through manipulations of encoding noise (for instance via changes in stimulus contrast).”

I appreciate the added explanation of how the current results connect to the notion of ‘higher-level processing’ for reality testing, however, I am not entirely sure how the imagery research the authors refer to is relevant here. For clarity, I would remove this reference (e.g. like “(particularly imagery-based paradigms that may require higher-order processes for imagery generation and reality monitoring [11]).”).

Response: Removed.

I would like to apologize for my incorrect reference to Phil Corlett for research on insight - I vaguely remembered he did something in this area, but that was incorrect (as the authors found out after their thorough search!), so, my apologies for not double checking before making this comment.

Reviewer #2 (Remarks to the Author):

The authors have been highly responsive to the reviews and addressed all major concerns.

Reviewer #3 (Remarks to the Author):

The authors have nicely responded to my comments. The addition of extra models and a whole new modelling framework that incorporates RT has made the paper much more comprehensive. Thank you.